# The positive–negative–competence (PNC) model of psychological responses to representations of robots

Dario Krpan ®[1] ✉, Jonathan E. Booth ®[2] & Andreea Damien ®[1]

Robots are becoming an increasingly prominent part of society. Despite their growing importance, there exists no overarching model that synthesizes people's psychological reactions to robots and identifies what factors shape them. To address this, we created a taxonomy of affective, cognitive and behavioural processes in response to a comprehensive stimulus sample depicting robots from 28 domains of human activity (for example, education, hospitality and industry) and examined its individual difference predictors. Across seven studies that tested 9,274 UK and US participants recruited via online panels, we used a data-driven approach combining qualitative and quantitative techniques to develop the positive–negative–competence model, which categorizes all psychological processes in response to the stimulus sample into three dimensions: positive, negative and competence-related. We also established the main individual difference predictors of these dimensions and examined the mechanisms for each predictor. Overall, this research provides an in-depth understanding of psychological functioning regarding representations of robots.

Various projections indicate that robots will soon become a constituent part of society and will need to be increasingly integrated into it[1–5]. This trend highlights the importance of understanding people's psychological processes (for example, feelings, thoughts and actions) towards robots. Indeed, these processes form the basis of human–robot relationships and are therefore likely to shape the dynamics of the new world permeated by robots[6–10]. In this respect, although various processes have been investigated[6], this research area is still in its infancy for several reasons.

First, scholars have not synthesized psychological processes towards robots into an overarching framework that clarifies how they function as a whole and allows for building theories that would explain them. Second, it is unclear whether, and how many, important psychological processes remain hidden due to the lack of systematic research on this topic. Third, previous studies have mainly focused on specific robot types (for example, social[6]) rather than examining the full content space of robots across all domains of human activity (for example,

education, hospitality and industry). Finally, most research has been conducted outside of psychology (for example, healthcare and robotics[6,11,12]). Consequently, there has been little effort to integrate people's responses to robots with important constructs from psychology in a way that would allow the field to study this topic more systematically and establish a coherent research stream around it.

To address this, the present research has two objectives: (1) to develop an integrative and comprehensive taxonomy of psychological processes in response to robots from all domains of human activity that organizes these processes into dimensions; and (2) to establish which individual differences widely studied in psychology are the most important predictors of these dimensions and to understand the mechanisms behind their relationships.

In this context we use the term 'psychological processes' in reference to people's affective (that is, feelings towards robots), cognitive (that is, thoughts about robots) and behavioural responses (that is, actions towards them). This rule-of-thumb classification is

[1]Department of Psychological and Behavioural Science, London School of Economics and Political Science, London, UK. [2]Department of Management, London School of Economics and Political Science, London, UK. ✉e-mail: d.krpan@lse.ac.uk

often used to summarize and investigate psychological processes in an all-encompassing way[13–15], because an official taxonomy does not exist. We adopt it because it is useful as a guiding principle when (1) eliciting diverse psychological processes and (2) identifying and organizing previous literature, considering that psychological functioning involving robots is typically not studied as a uniform construct and comprises studies from numerous areas.

Next, we briefly review previous research on psychological processes regarding robots in terms of affective, cognitive and behavioural responses (for a detailed review see Supplementary Notes). Before this review, we first clarify how we define robots because their definition is often confined to various specific types (for example, autonomous and social[6,16–20]), and describing them as an overarching category can be less straightforward[19–21].

We adopt a general definition proposed by the Institute of Electrical and Electronics Engineers (IEEE[22]), according to which robots are devices that can act in the physical world to accomplish different tasks and are made of mechanical and electronic parts. These devices can be autonomous or subordinated to humans or software agents acting on behalf of humans. Robots can also form groups (that is, robotic systems) in which they cooperate to accomplish collective goals (for example, car manufacturing).

Previous research has documented diverse affective responses to robots, which can be classified as negative or positive[6]. Regarding negative feelings, fear and anxiety are typically experienced concerning robots taking people's jobs[23–27]. Individuals can also find robots creepy if they are designed to be human-like but look unnatural and inconsistent with human appearance[28]. Regarding positive feelings, individuals can experience happiness, amazement, amusement, enjoyment, pleasure, warmth and empathy towards robots[6,10,25,26,29–38]. Interestingly, people can also become emotionally attached, feel attracted and be in love with robots[39–44]. Whereas these romantic feelings are perceived by many as taboo, they are becoming increasingly frequent nowadays[42].

In terms of cognitive responses, people's thoughts about robots can be organized into several themes. A key theme is the level of competence displayed by robots concerning tasks in which they are specialized[19,30,45,46]. For example, robots are often seen as efficient and accurate in what they do and as more physically endurant than humans[19,47,48]. Individuals can also consider robots helpful and appreciate their effectiveness in accomplishing various tasks, from household chores to carrying heavy loads[49–52]. Another important theme is anthropomorphism (that is, ascribing human characteristics to non-living entities[7]). For instance, people may perceive robots as sentient beings that have feelings[53–59] but they may also see them as distinct from humans (for example, cold or soulless[19,60–62]) and question whether robots can be trusted in their capacities as companions, coworkers and other roles they assume[63–65].

In terms of behavioural responses, actions towards robots can be classified as either approach (for example, engaging with them) or avoidance (for example, evading them)[10,66–68]. Common approach behaviours involve communicating, cooperating, playing and requesting information[10,26,69,70]. More negative approach behaviours have also been documented, including several instances of robot abuse[71–73]. In contrast to approach, avoidance behaviours (for example, hiding from robots) are infrequently mentioned in the literature and may typically occur in environments where robots could potentially injure humans[74,75].

Overall, the reviewed literature indicates that various psychological responses to robots have been observed. However, because this topic is not studied under a common umbrella of psychological processes but in relation to diverse topics (for example, anthropomorphism, robotic job replacement or robot acceptance[7,49,76]), it is unclear how these processes are interlinked, what shapes them and whether all important processes have been discovered.

For these reasons, our research adopted a data-driven rather than a theory-driven approach[77–79]. Contrary to theory-driven studies that are inherently deductive because they test hypotheses deduced from general principles (that is, theory), data-driven research is inductive because it starts with empirical observations that are not guided by hypotheses and can progressively evolve into theory[77–82].

A data-driven approach is recommended if (1) a construct is in its early stages of development and/or (2) its theoretical foundations have not been established[77–80,82,83]. Based on this, a data-driven approach is optimal for our research for both reasons. First, as previously indicated, the conceptual bases of our topic are at an early stage because different affective, cognitive and behavioural responses to robots have not been studied under an all-encompassing construct (that is, psychological processes). Second, theoretical foundations have not yet been developed, because encapsulating the entirety of psychological functioning regarding robots by identifying, organizing and predicting the psychological processes triggered by robots is beyond the scope of existing models of human–technology relationships. To illustrate this, the technology acceptance model[84–86] and its extensions—the unified theory of acceptance and use of technology[87–89] and the Almere model[90]—examine the factors that make people accept technology (for example, perceived usefulness, ease of use or social influence) whereas the media equation[91–93] examines whether people interact with media (for example, computers) similarly to how they interact with other humans.

Data-driven approaches have three main benefits. First, they allow the study of novel topics without engaging in premature theorizing that can lead to post hoc hypothesizing and false-positive findings[77,78,94–97]. Second, because the emphasis is on inferences from data that are not constrained by previous theories and findings, these approaches can diversify knowledge of human psychology and spark unexpected insights[79,81,98]. Third, they can be more beneficial to previous research on the topic than deductive approaches directly informed by this research. In behavioural sciences, failed replications are common and researchers examining the same research questions and hypotheses, even with identical data, can often obtain different findings[99–103]. Therefore, if a data-driven study produces a finding consistent with previous research and theorizing, despite using a methodological approach that is solely guided by data and not constrained by their assumptions, this is a compelling case of support for the previous work. It is thus important to emphasize that using a data-driven approach does not imply conducting a research project that disregards previous literature. Quite the contrary, it is essential to comprehensively evaluate and discuss how the findings are linked to previous work to illuminate how the present research has extended this work and moved the field forward—a process labelled inductive integration[77].

Drawing on data-driven approaches, our research objectives—(1) establishing a taxonomy of psychological processes involving robots and (2) examining its individual difference predictors—are achieved in three phases comprising seven studies (Fig. 1; for participant information see Table 1).

Phase 1 consisted of two studies that undertook an in-depth examination of the construct of robots that was necessary to build the taxonomy. In Study 1 we developed an all-encompassing general definition of robots. In Study 2 we used this definition to identify all domains of human activity in which robots operate.

Phase 2 consisted of three studies aimed at creating the taxonomy. In Study 3 we sampled a comprehensive content space of people's psychological processes involving robots across the domains identified in Phase 1 to develop items assessing each process. In Study 4 we determined the main dimensions of these processes using exploratory factor analyses (EFAs[104–106]). In Study 5 we further confirmed these dimensions using exploratory structural equation modelling (ESEM[107,108]) and developed the psychological responses to robots (PRR) scale that can assess psychological processes towards any robot.

Phase 3 consisted of two studies that focused on determining the most important individual difference predictors of the psychological responses and testing the mechanisms behind these relationships.

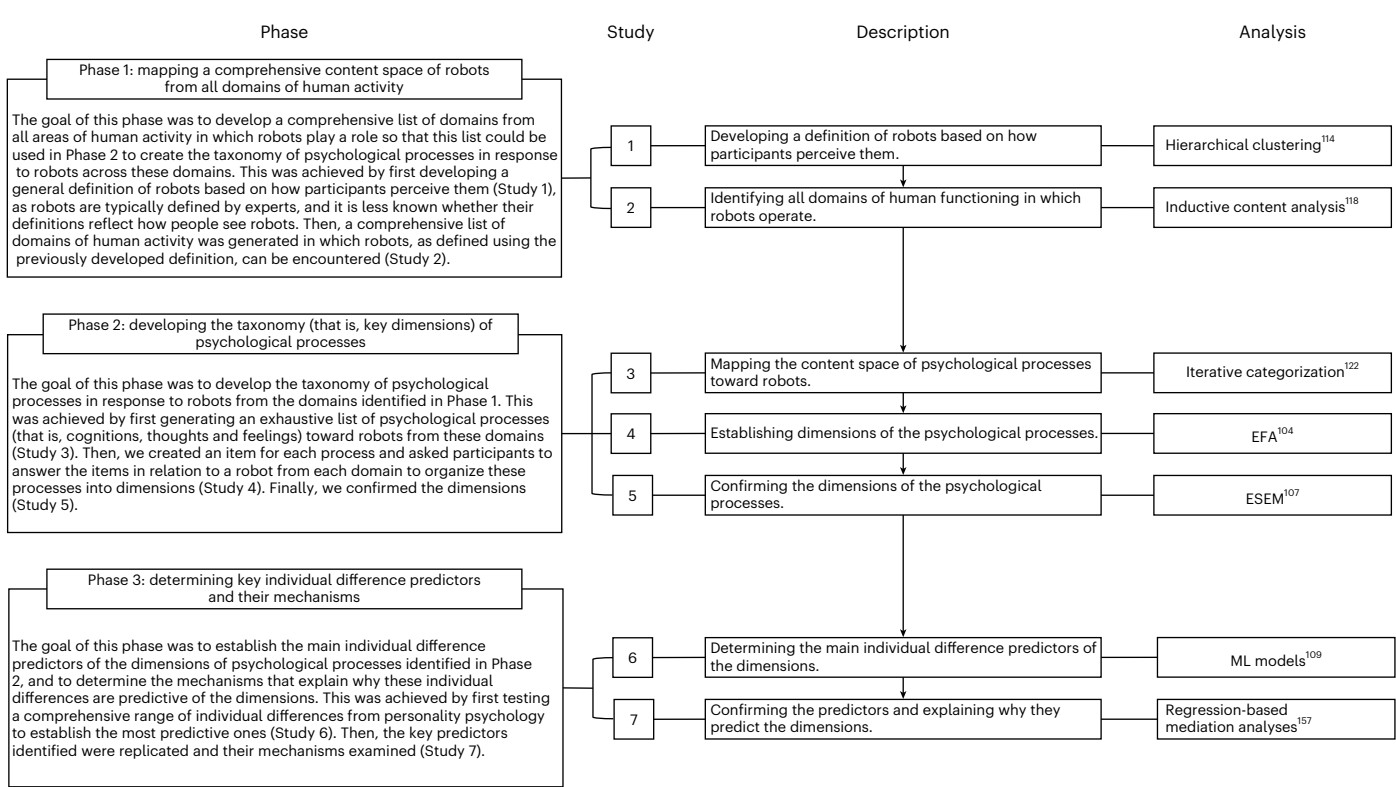

**Fig. 1 | Overview of the present research.** 'Phase' outlines the goals of each research phase, how these goals were achieved and the link between successive phases. 'Study' and 'description' indicate the number of each study and its goal while 'analysis' specifies the statistical analyses that were used in each study.

In Study 6 we used machine learning[109,110] to identify the key predictors of the main dimensions of the PRR scale. In Study 7 we probed the mechanisms behind these predictors.

All in all, to achieve our research objectives, as stimuli we used representations (that is, images and descriptions) of robots (Supplementary Table 7) from 28 exhaustive domains of human activity in which robots operate (Table 2). This comprehensive approach allowed us to minimize the chance that our findings are driven by idiosyncrasies of a sample that is small in size and/or variety of robot types, which could compromise replicability[111,112]. Despite the wide variety of our stimulus sample, it is unclear to what degree this sample is representative of the general population of robots because (1) there are no established recommendations on what variables would need to be measured to accurately define this population, (2) the type of data used to quantify general characteristics of human populations is not available for robots and (3) the field of robotics is rapidly evolving. Therefore, in the context of our research we use the term 'robot/s' in reference to our specific stimulus sample and we do not imply that our insights extend to the general population (that is, all physical robots).

## Results

In this section we briefly present the results (for a detailed description see Supplementary Results).

### Phase 1: mapping a comprehensive content space of robots

Phase 1 aimed to establish a comprehensive content space that encompasses a wide range of robots by identifying all domains of human activity in which robots operate, to ensure that our taxonomy developed in Phase 2 is not biased towards only a few robot types[111,112].

The first step in this endeavour was to devise a general definition of robots in Study 1, because robot definitions are typically proposed by experts[6,21,22,113] and it is less well known whether these reflect how people more broadly perceive robots. Because any robot definition is essentially a set of characteristics that describe robots (for example, made of mechanical parts, autonomous[6,22,113]), to develop a general definition we first recruited Sample 1 and asked them to generate robot characteristics. Using this approach, 277 characteristics were identified (Supplementary Table 3). We then recruited Sample 2 and asked them to group these characteristics into common categories. Using hierarchical cluster analysis[114–116], the following main clusters of robot characteristics were identified: (1) characteristics conveying the degree of robot–human similarity; (2) positive characteristics; (3) characteristics conveying robots' composition; (4) negative characteristics; and (5) characteristics conveying robots' ability to perform various tasks (Supplementary Table 3).

The general definition of robots that we subsequently developed by linking the themes of each cluster is available in Table 2. It is important to emphasize that we did not form the definition by always translating an individual cluster theme into a separate part, because the definition was more succinct and coherent if certain themes were combined in the same parts.

In Study 2 we used this robot definition to identify a comprehensive list of domains in which robots operate. Participants were presented with the definition and asked to generate all such domains they could think of. To develop an extensive inventory of domains, we analysed their responses using inductive content analysis[117–121]. Additionally, to ensure we did not miss any domains that participants were unable to identify, we consulted various other resources (for example, articles from the literature review of this paper and classifications detailed in Methods). The final list of domains, accompanied by the example items generated by participants, is available in Table 2.

### Phase 2: creating the taxonomy of psychological processes

To develop the taxonomy, it was first necessary to identify a comprehensive range of psychological processes involving robots in Study 3. We instructed participants to write about any feelings, thoughts and

**Table 1 | Sample size and background information for all participants who completed a study, and for those participants included in analyses (Studies 1–7)**

| Study no. | Sample no. | Sample size | Country | Age (years) | | Gender | | | | Employment status | | | Use of robots at work | | | |
|---|---|---|---|---|---|---|---|---|---|---|---|---|---|---|---|---|
| | | | | Mean | s.d. | Female | Male | Other | UD[a] | Employed | Unemployed | UD[a] | Don't know | No | Yes | UD[a] |
| **All participants** | | | | | | | | | | | | | | | | |
| 1 | 1 | 266 | UK | 49.496 | 13.598 | 132 | 133 | 1 | 0 | 175 | 91 | 0 | 3 | 161 | 11 | 0 |
| | | | | | | 49.62% | 50.00% | 0.38% | 0% | 65.79% | 34.21% | 0% | 1.71% | 92.00% | 6.29% | 0% |
| 1 | 2 | 100 | US | 36.510 | 10.566 | 42 | 58 | 0 | 0 | 94 | 6 | 0 | 2 | 90 | 2 | 0 |
| | | | | | | 42.00% | 58.00% | 0% | 0% | 94.00% | 6.00% | 0% | 2.13% | 95.74% | 2.13% | 0% |
| 2 | – | 70 | US | 36.257 | 10.270 | 31 | 39 | 0 | 0 | 64 | 6 | 0 | 1 | 55 | 8 | 0 |
| | | | | | | 44.29% | 55.71% | 0% | 0% | 91.43% | 8.57% | 0% | 1.56% | 85.94% | 12.50% | 0% |
| 3 | – | 350 | US | 40.693 | 12.194 | 193 | 153 | 1 | 3 | 325 | 24 | 1 | 5 | 279 | 41 | 0 |
| | | | | | | 55.14% | 43.71% | 0.29% | 0.86% | 92.86% | 6.86% | 0.29% | 1.54% | 85.85% | 12.62% | 0% |
| 4 | 1 | 1,668 | UK | 47.932 | 16.611 | 852 | 812 | 4 | 0 | 1,043 | 624 | 1 | 13 | 955 | 75 | 0 |
| | | | | | | 51.08% | 48.68% | 0.24% | 0% | 62.53% | 37.41% | 0.06% | 1.25% | 91.56% | 7.19% | 0% |
| 4 | 2 | 1,808 | US | 48.004 | 16.772 | 976 | 830 | 2 | 0 | 1,053 | 754 | 1 | 14 | 871 | 168 | 0 |
| | | | | | | 53.98% | 45.91% | 0.11% | 0% | 58.24% | 41.70% | 0.06% | 1.33% | 82.72% | 15.95% | 0% |
| 5 | 1 | 1,200 | UK | 46.648 | 16.616 | 590 | 601 | 6 | 3 | 753 | 447 | 0 | 14 | 690 | 49 | 0 |
| | | | | | | 49.17% | 50.08% | 0.50% | 0.25% | 62.75% | 37.25% | 0% | 1.86% | 91.63% | 6.51% | 0% |
| 5 | 2 | 1,219 | US | 46.656 | 16.914 | 616 | 598 | 5 | 0 | 712 | 506 | 1 | 12 | 639 | 61 | 0 |
| | | | | | | 50.53% | 49.06% | 0.41% | 0% | 58.41% | 41.51% | 0.08% | 1.69% | 89.75% | 8.57% | 0% |
| 6 | – | 2,505 | US | 47.405 | 17.262 | 1,299 | 1,186 | 15 | 5 | 1,537 | 964 | 4 | 19 | 1,210 | 307 | 1 |
| | | | | | | 51.86% | 47.35% | 0.60% | 0.20% | 61.36% | 38.48% | 0.16% | 1.24% | 78.72% | 19.97% | 0.07% |
| 7 | – | 1,116 | US | 42.910 | 13.535 | 552 | 555 | 9 | 0 | 843 | 273 | 0 | 22 | 754 | 66 | 1 |
| | | | | | | 49.46% | 49.73% | 0.81% | 0% | 75.54% | 24.46% | 0% | 2.61% | 89.44% | 7.83% | 0.12% |
| **Participants included in analyses** | | | | | | | | | | | | | | | | |
| 1 | 1 | 224 | UK | 50.344 | 13.262 | 121 | 102 | 1 | 0 | 145 | 79 | 0 | 3 | 136 | 6 | 0 |
| | | | | | | 54.02% | 45.54% | 0.45% | 0% | 64.73% | 35.27% | 0% | 2.07% | 93.79% | 4.14% | 0% |
| 1 | 2 | 95 | US | 36.621 | 10.729 | 39 | 56 | 0 | 0 | 91 | 4 | 0 | 2 | 88 | 1 | 0 |
| | | | | | | 41.05% | 58.95% | 0% | 0% | 95.79% | 4.21% | 0% | 2.20% | 96.70% | 1.10% | 0% |
| 2 | – | 67 | US | 35.657 | 9.634 | 31 | 36 | 0 | 0 | 61 | 6 | 0 | 1 | 52 | 8 | 0 |
| | | | | | | 46.27% | 53.73% | 0% | 0% | 91.04% | 8.96% | 0% | 1.64% | 85.25% | 13.11% | 0% |
| 3 | – | 334 | US | 40.826 | 12.154 | 184 | 147 | 1 | 2 | 311 | 22 | 1 | 5 | 270 | 36 | 0 |
| | | | | | | 55.09% | 44.01% | 0.30% | 0.60% | 93.11% | 6.59% | 0.30% | 1.61% | 86.82% | 11.58% | 0% |
| 4 | 1 | 1,528 | UK | 48.328 | 16.515 | 790 | 734 | 4 | 0 | 944 | 583 | 1 | 12 | 874 | 58 | 0 |
| | | | | | | 51.70% | 48.04% | 0.26% | 0% | 61.78% | 38.15% | 0.07% | 1.27% | 92.58% | 6.14% | 0% |
| 4 | 2 | 1,537 | US | 49.465 | 16.563 | 861 | 674 | 2 | 0 | 870 | 667 | 0 | 11 | 745 | 114 | 0 |
| | | | | | | 56.02% | 43.85% | 0.13% | 0% | 56.60% | 43.40% | 0% | 1.26% | 85.63% | 13.10% | 0% |
| 5 | 1 | 1,107 | UK | 47.112 | 16.583 | 544 | 555 | 6 | 2 | 691 | 416 | 0 | 10 | 639 | 42 | 0 |
| | | | | | | 49.14% | 50.14% | 0.54% | 0.18% | 62.42% | 37.58% | 0% | 1.45% | 92.47% | 6.08% | 0% |
| 5 | 2 | 1,108 | US | 47.100 | 16.947 | 563 | 540 | 5 | 0 | 651 | 456 | 1 | 12 | 591 | 48 | 0 |
| | | | | | | 50.81% | 48.74% | 0.45% | 0% | 58.75% | 41.16% | 0.09% | 1.84% | 90.78% | 7.37% | 0% |
| 6 | – | 2,203 | US | 47.947 | 17.493 | 1,164 | 1,021 | 14 | 4 | 1,316 | 883 | 4 | 16 | 1,064 | 235 | 1 |
| | | | | | | 52.84% | 46.35% | 0.64% | 0.18% | 59.74% | 40.08% | 0.18% | 1.22% | 80.85% | 17.86% | 0.08% |
| 7 | – | 1,071 | US | 42.846 | 13.450 | 535 | 527 | 9 | 0 | 808 | 263 | 0 | 22 | 721 | 64 | 1 |
| | | | | | | 49.95% | 49.21% | 0.84% | 0% | 75.44% | 24.56% | 0% | 2.72% | 89.23% | 7.92% | 0.12% |

All studies were administered via Qualtrics. In Studies 1 (Sample 1), 4 (Samples 1 and 2), 5 (Samples 1 and 2) and 6, participants were recruited via Pureprofile; in Studies 1 (Sample 2), 2 and 3, participants were recruited via Amazon Mechanical Turk; in Study 7, participants were recruited via Prolific. Samples for Studies 4–6 were recruited to be reasonably representative of the UK/US populations in terms of age, gender and geographical region, whereas for Study 1 (Sample 1) the focus was on gender only. Supplementary Tables 1 and 2 contain more comprehensive breakdowns of these variables, the criteria that were used to guide representative sampling and additional demographic characteristics. [a]UD, undisclosed: for gender, this category comprises participants who either selected the option 'choose not to disclose' or whose data were missing; for other variables, this category comprises participants whose data were missing. For employment status, the category 'employed' comprises participants who were either self-employed or working for an employer whereas 'unemployed' refers to participants who were not working for themselves or someone else.

**Table 2 | Definition of robots developed from the clusters of their characteristics generated by participants (Study 1), and robot domains grounded in this definition (Study 2), with example participant items analysed using indictive content analysis to develop the domains**

| Definition part | Definition | Robot clusters that informed the definition[a] |
|---|---|---|
| 1 | A robot is a non-living entity that primarily functions as helping and/or substituting humans in some capacity by performing physical and/or intellectual tasks that range from simple, routine to complex ones. | 1, 4, 5 |
| 2 | Robots are characterized by different degrees of autonomy: sometimes they only follow commands that have been pre-programmed, but sometimes they are artificially intelligent and thus are able to learn from the environment and adapt to it. | 1 |
| 3 | Although robots at times require maintenance and repair, they have potential to work tirelessly over long periods of time as they do not have life commitments (e.g., family) and/or wellbeing considerations as humans do (e.g., time off, sick leave). | 1 |
| 4 | Humans can perceive robots as having positive attributes (e.g., clever, consistent, cute, efficient, flexible, friendly, reliable, robust, safe, supportive). However, humans also can have negative perceptions about robots, such as seeing them as cold, creepy, emotionless, lacking conscience, soulless, threatening, etc., generally attributing negative qualities to robots as a result of their nonhuman nature. | 2, 4 |
| 5 | A robot typically consists of software (i.e., the code or programme on which it runs) and different materials and components used to produce it (e.g., metal, wires, sensors, microchips, etc.). Although robots can take the form and/or have characteristics of humans, they can appear as an animal or any non-living object. | 1, 3 |
| Domain no. | Domain | Example items |
| 1 | Health and human care and wellbeing (e.g., medical, surgical, fitness, lab diagnostics, elderly, disability, infant/child, and personal care) | Elder care in home; at the doctor or hospital; exercise |
| 2 | Social and companionship | Companionship; social life |
| 3 | Sex (this domain was added based on ref. 17 and was not generated from participants' responses) | – |
| 4 | Animal care (e.g., walking pets) | Animal care; walking pets |
| 5 | Security and surveillance | Public safety; security |
| 6 | Policing and military | Policing robots; warfare |
| 7 | Education, libraries, and knowledge/information management and gathering | Library; studying |
| 8 | Research and exploration within science, technology, engineering, and mathematics (STEM) (e.g., ocean exploration, supercomputing, IT innovation, space discovery) | Research/exploration |
| 9 | Communication tools and channels | Chatbots; communications |
| 10 | Leisure, recreation, and travel | Travel; watercraft |
| 11 | Culture/entertainment, gaming, toys, and other amusement | Concerts; entertainment |
| 12 | Workplace domain (i.e., to aid or replace human effort) | Work product generation |
| 13 | Dangerous and/or risky work | Do dangerous work |
| 14 | Inspection, repair and/or improvement of products, engines, equipment, technology, and/or infrastructure (e.g., buildings, bridges, roads, power supplies, nuclear reactors, pipes, gas mains) | Repairs; auto maintenance |
| 15 | Agriculture (e.g., harvesting, farms) | Agriculture; farms |
| 16 | Household chores/tasks and domestic help/assistance (i.e., inside and outside of the home) | Home; house chores |
| 17 | Industry | Factory/factories |
| 18 | Hospitality and food service (i.e., hotels, conventions, restaurants, bars, and other lodging, space, food and/or drink provider) and related customer service and support | Eating out; hospitality |
| 19 | Banking/financial services and related customer service and support | ATM; bank; business |
| 20 | Retail and commerce and related customer service and support | Retail; self-checkout |
| 21 | Construction | Construction; demolition |
| 22 | Manufacturing | Manufacturing facilities |
| 23 | Mining | Mining |
| 24 | Warehouses and fulfilment centres | Warehouse work |
| 25 | Public services (e.g., road work and other shared public good) | Roadwork |
| 26 | Transportation (i.e., land, water, and/or sky) of goods, people, and other living entities, transport equipment, and delivery/courier/shipping services | Drone; transport |
| 27 | Airports | Airports |
| 28 | Art (this domain was added based on ref. 18 and was not generated from participants' responses) | – |

Our aim was to develop domains that are narrow rather than broad, which means that some overlap between them may be present. This approach was aligned with our objective to establish a comprehensive content space of all robots to decrease the probability of using a biased stimulus sample[111,112] when developing the taxonomy of psychological responses to representations of robots. For that reason, it was more optimal to lean towards having too many rather than too few domains, to reduce the chance of failing to cover the content space of all robots in detail and omitting important types of robots. [a]Cluster 1, characteristics conveying the degree of robot–human similarity; Cluster 2, positive characteristics; Cluster 3, characteristics conveying robots' composition; Cluster 4, negative characteristics; and Cluster 5, characteristics conveying robots' ability to perform various tasks (Supplementary Table 3).

behaviours they could think of concerning robots from the domains developed in Study 2 (Table 2)—each participant was randomly allocated to one of five domains. Participants were not provided with specific robot examples for a given domain, because we expected that reliance on their own reflections and experiences would cover a broader spectrum of robots and therefore increase the diversity of psychological processes reported (for a similar methodological approach see ref. 82). Table 3 contains the final list of psychological processes derived from participants' responses using iterative categorization[122].

In Study 4 we then created items for each of these processes (Table 3) and asked participants to answer the items about an example of a robot (Supplementary Table 7) from one of the 28 domains (Table 2) to which they were randomly allocated. To develop the taxonomy from participants' responses we used maximum-likelihood EFAs[104] with Kaiser[123] normalized promax rotation[106,124]. EFAs were appropriate because the Kaiser–Meyer–Olkin measure of sampling adequacy was 0.983 and 0.984 for Samples 1 and 2, respectively, and Bartlett's test of sphericity was significant (for both samples, $P < 0.001$)[125].

To select the most appropriate factor solution we used the following procedure. We first consulted parallel analysis[126,127], very simple structure[128], Velicer map[129], optimal coordinates[130], acceleration factor[130], Kaiser rule[131] and visual inspection of scree plots[132], which indicated that extraction of between one and 19 factors (Sample 1) and between two and 18 factors (Sample 2) could be optimal. Next, we evaluated the largest factor solutions (that is, 19 factors for Sample 1 and 18 for Sample 2) against several statistical and semantic benchmarks. If the benchmarks were not met we decreased the number of factors by one and evaluated these new solutions. This procedure was continued until the benchmarks were met. Concerning statistical benchmarks, a factor solution was required to produce only valid factors—those that have at least three items with standardized loadings ≥0.5 and cross-loadings of <0.32 (refs. 105,125,133,134). Semantically, a solution was required to make sense conceptually by having factors that are coherent and easy to interpret[135,136].

For Samples 1 and 2, three-factor solutions emerged as the most optimal. These met the statistical criteria and had semantically coherent factors that denoted positive, negative and competence-related psychological processes (Table 3). Therefore, the taxonomy was labelled the positive–negative–competence (PNC) model of psychological processes regarding robots. None of the larger factor solutions met the statistical criteria.

We aimed to further validate the PNC model by confirming its dimensions and thereby developing the PRR scale that measures them. To do this, in Study 4 we selected a representative subset of PNC items (bold items in Table 3) and subjected them to ESEM[107] using the maximum-likelihood with robust standard errors (MLR) estimator[137,138] and target rotation with all cross-loadings as targets of zero[139,140]. For both samples, fit indices showed good to excellent fit (that is, SRMR < 0.05, CFI > 0.90, RMSEA < 0.06 (refs. 141–143)): Sample 1: $\chi^2(558) = 1,953.820$, $P < 0.001$, SRMR = 0.026, CFI = 0.939, RMSEA = 0.041, 90% confidence interval (CI) [0.039, 0.043]; Sample 2: $\chi^2(558) = 1,850.880$, $P < 0.001$, SRMR = 0.025, CFI = 0.944, RMSEA = 0.039, 90% CI [0.037, 0.041].

Subsequently, in Study 5 we recruited two additional samples and asked participants to answer these items about one of the two robot examples (Supplementary Table 7) from one of the 28 domains to which participants were randomly allocated. The ESEM models for both samples had a good to excellent fit (Table 4). Moreover, items previously classified under a specific dimension (that is, positive, negative or competence) by EFAs in Study 4 (Table 3) had the highest loadings for this dimension whereas the cross-loadings were <0.32. To ensure that the model comprising the three dimensions was the most appropriate we tested several alternative models, which were all rejected due to poor fit (Supplementary Results).

To show that our model has equivalent factor structure, loadings and intercepts regardless of participants' country, robot examples used and several key demographic characteristics, we tested configural, metric and scalar measurement invariance[144–146]. As shown in Table 5, measurement invariance was demonstrated in all cases given that the configural model demonstrated good to excellent fit (SRMR < 0.05, CFI > 0.90, RMSEA < 0.06 (refs. 141–143); changes in SRMR, CFI and RMSEA were, respectively, ≤0.030, 0.010 and 0.015 for the metric model and ≤0.015, 0.010 and 0.015 for the scalar model[144]. Since we could not analyse measurement invariance for participants who did versus did not use robots at work in Study 5 because the number of those who did was insufficient (Table 1), we tested this in Study 6 where sample sizes were larger. In Study 6 we also computed measurement invariance for additional participant characteristics assessed in that study (educational attainment, income, being liberal versus conservative, ethnic identity and relationship status). Measurement invariance was demonstrated in all cases (Supplementary Table 10).

Overall, the structure of the PNC model and its validity across different subgroups of participants were confirmed.

### Phase 3: examining individual difference predictors

In Study 6, to identify the main predictors of the PNC model we followed the analytic strategy described in Methods. We first computed 11 common machine learning models (for example, linear least squares, lasso[109,110]) for the positive, negative and competence dimensions separately. The key predictors in each model were 79 personality measures that were found to be conceptually or theoretically relevant to the PNC dimensions. We selected these measures by examining several comprehensive psychological scale databases (for example, Database of Individual Differences Survey Tools[147]). All measures and their justifications are available in Supplementary Table 11.

We then identified the most predictive models, which were the same across all PNC dimensions: conditional random forest ($r.m.s.e._{Positive} = 0.919$; $r.m.s.e._{Negative} = 0.988$; $r.m.s.e._{Competence} = 0.778$), linear least squares ($r.m.s.e._{Positive} = 0.929$; $r.m.s.e._{Negative} = 1.000$; $r.m.s.e._{Competence} = 0.795$), ridge ($r.m.s.e._{Positive} = 0.921$; $r.m.s.e._{Negative} = 0.994$; $r.m.s.e._{Competence} = 0.787$), lasso ($r.m.s.e._{Positive} = 0.921$; $r.m.s.e._{Negative} = 0.993$; $r.m.s.e._{Competence} = 0.784$), elastic net ($r.m.s.e._{Positive} = 0.921$; $r.m.s.e._{Negative} = 0.993$; $r.m.s.e._{Competence} = 0.784$) and random forest ($r.m.s.e._{Positive} = 0.925$; $r.m.s.e._{Negative} = 0.995$; $r.m.s.e._{Competence} = 0.781$).

Subsequently we determined all individual differences that were among the top 30 predictors across these six models and that were also statistically significant in the linear least-squares model after applying the false-discovery rate[148] correction (Supplementary Tables 12–15). Several variables met these criteria and were therefore deemed the main individual difference predictors of PNC dimensions. For the positive dimension these were general risk propensity (GRP[149]), anthropomorphism (IDAQ[150]) and parental expectations (FMPS_PE[151]); for the negative dimension these were trait negative affect (PANAS_TNA[152]), psychopathy (SD3_P[153]), anthropomorphism (IDAQ[150]) and expressive suppression (ERQ_ES[154]); and for the competence dimension these were approach temperament (ATQ_AP[155]) and security-societal (PVQ5X_SS[156]). According to the most interpretable model (that is, the linear least squares) these most predictive individual differences were positively associated with the corresponding PNC dimensions.

In Study 7, to replicate the findings we measured the most predictive individual differences in wave 1 and used linear regressions to show that they significantly predicted PNC dimensions in wave 2 (Table 6), consistent with Study 6. Furthermore, we examined various potential mediators of the relationship between each predictor and a PNC dimension using parallel mediation analyses[157] percentile-bootstrapped with 10,000 samples (for mediators and mediated effects see Table 7; the rationale behind each mediator and detailed mediation analyses are available in Supplementary Table 17 and Supplementary Results, respectively).

**Table 3 | Summary of key findings (Studies 3 and 4): psychological processes, items corresponding to each process and the output of EFAs performed on the items across two participant samples**

| Item no. | Psychological process | Item | Sample 1 (UK) | | | Sample 2 (US) | | |
|---|---|---|---|---|---|---|---|---|
| | | | P | N | C | P | N | C |
| 61 | Companionship | This robot would make a good companion. | 0.853 | | | 0.864 | | |
| 84 | Enjoyment | **I associate this robot with enjoyment**. | 0.819 | | | 0.815 | | |
| 85 | Humour | This robot is humorous. | 0.806 | | | 0.711 | | |
| 35 | Attachment | **I would feel attached to this robot**. | 0.793 | | | 0.864 | | |
| 26 | Treating the robot like a human | I would treat this robot as if it were a human. | 0.779 | | | 0.814 | | |
| 34 | Intimacy | I would be able to connect on an intimate level with this robot (e.g., share feelings, be in close contact, hug or hold, etc.). | 0.775 | | | 0.842 | | |
| 60 | Comfort | This robot is comforting. | 0.772 | | | 0.792 | | |
| 65 | Friendliness | This robot is friendly. | 0.763 | | | 0.764 | | |
| 116 | Empathy | **This robot is empathetic**. | 0.753 | | | 0.751 | | |
| 88 | Thoughtfulness | **This robot is thoughtful**. | 0.745 | | | 0.793 | | |
| 122 | Entertainment | This robot is entertaining. | 0.724 | | | 0.700 | | |
| 123 | Play | I would like to play with this robot. | 0.715 | | | 0.742 | | |
| 62 | Interaction | I would want to interact with this robot. | 0.693 | | | 0.663 | | |
| 40 | Engagement | **I would like to engage with this robot**. | 0.692 | | | 0.689 | | |
| 32 | Happiness | **This robot makes me feel happy**. | 0.681 | | | 0.806 | | |
| 41 | Motivation | This robot motivates me. | 0.681 | | | 0.796 | | |
| 27 | Perceiving the robot as human-like | **This robot is like a human**. | 0.675 | | | 0.743 | | |
| 63 | Communication | **I would find it easy to communicate with this robot**. | 0.642 | | | 0.685 | | |
| 67 | Wellbeing | This robot promotes wellbeing. | 0.635 | | | 0.685 | | |
| 24 | Anthropomorphism | I can see human traits in this robot. | 0.625 | | | 0.681 | | |
| 77 | Pleasantness | I find this robot pleasant. | 0.623 | | | 0.689 | | |
| 149 | Robot rights | **I think this robot should have rights**. | 0.609 | | | 0.711 | | |
| 95 | Relaxation | This robot makes me feel relaxed. | 0.608 | | | 0.742 | | |
| 66 | Care | **This robot provides care**. | 0.600 | | | 0.675 | | |
| 132 | Self-improvement | **This robot helps me to improve myself**. | 0.597 | | | 0.773 | | |
| 14 | Excitement | This robot makes me feel excited. | 0.590 | | | 0.734 | | |
| 125 | Creativity | **This robot is creative**. | 0.573 | | | 0.615 | | |
| 89 | Learning from robots | I could learn from this robot. | 0.567 | | | 0.648 | | |
| 107 | Empowerment | **I feel empowered by this robot**. | 0.545 | | | 0.691 | | |
| 47 | Pride | I feel proud about this robot. | 0.541 | | 0.343 | 0.691 | | |
| 64 | Social support | This robot provides support to me. | 0.534 | | | 0.633 | | |
| 129 | Competition | I would want to compete with this robot. | 0.532 | 0.420 | | 0.496 | 0.471 | |
| 96 | Gratitude | **This robot makes me feel grateful**. | 0.526 | | | 0.683 | | |
| 99 | Gaining knowledge about robot | I would want to learn more about this robot. | 0.504 | | | 0.552 | | |
| 46 | Hope | This robot makes me feel hopeful. | 0.501 | | 0.371 | 0.669 | | |
| 57 | Safety | This robot makes me feel safe. | 0.499 | | | 0.641 | | |
| 42 | Admiration | I admire this robot. | 0.495 | | 0.333 | 0.704 | | |
| 83 | Surprise | This robot surprises me. | 0.495 | | | 0.531 | | |
| 39 | Interest | I would be interested in this robot. | 0.486 | | | 0.572 | | |
| 17 | Awareness | **This robot has awareness**. | 0.482 | | | 0.576 | | |
| 143 | Liberation | This robot makes me feel liberated or free. | 0.481 | | 0.324 | 0.686 | | |
| 68 | Cooperation | This robot and I could cooperate. | 0.457 | | | 0.490 | | 0.330 |
| 98 | Information search | This robot provides me information. | 0.456 | | | 0.479 | | |

**Table 3 (continued) | Summary of key findings (Studies 3 and 4): psychological processes, items corresponding to each process and the output of EFAs performed on the items across two participant samples**

| Item no. | Psychological process | Item | Sample 1 (UK) | | | Sample 2 (US) | | |
|---|---|---|---|---|---|---|---|---|
| | | | P | N | C | P | N | C |
| 86 | Artificial intelligence | This robot is intelligent. | 0.421 | | | 0.511 | | |
| 9 | Need fulfilment | This robot fulfils my needs. | 0.409 | | 0.383 | 0.659 | | |
| 146 | Testing the robot | I would experiment with or test this robot to see what it can do. | 0.402 | | | 0.399 | | |
| 71 | Openness | I would be open to this robot. | 0.390 | −0.348 | | 0.450 | | 0.323 |
| 87 | Learning | This robot can learn. | 0.382 | | | 0.437 | | |
| 130 | Winning | I want to beat or outperform this robot. | 0.378 | 0.481 | | | 0.590 | |
| 100 | Being knowledgeable about robot | I am knowledgeable about this robot. | 0.364 | | | 0.483 | | |
| 82 | Positive affect | I feel positive about this robot. | 0.359 | −0.424 | 0.366 | 0.519 | | 0.342 |
| 45 | Uniqueness | This robot is unique. | 0.354 | | | 0.335 | | |
| 18 | Ignoring | I would ignore this robot. | −0.324 | 0.352 | | −0.379 | 0.435 | |
| 19 | Monotony | This robot deals with monotonous and repetitive tasks. | −0.362 | | 0.623 | | | 0.559 |
| 38 | Indifference | I am indifferent toward this robot. | −0.388 | | | −0.344 | | |
| 29 | Instrumentality | This robot is just a means to an end. | −0.435 | | 0.329 | −0.412 | | 0.374 |
| 28 | Objectification | This robot is merely an object. | −0.464 | | | −0.554 | | |
| 37 | Emotionless (human) | I feel no emotions toward this robot. | −0.533 | | | −0.573 | | |
| 23 | Not human | This robot does not feel or respond like humans. | −0.547 | | | −0.614 | | 0.402 |
| 36 | Emotionless (robot) | This robot is emotionless. | −0.715 | | 0.384 | −0.662 | | 0.473 |
| 13 | Anxiety | **This robot makes me feel anxious.** | | 0.806 | | | 0.693 | |
| 59 | Threat | **I feel threatened by this robot.** | | 0.804 | | | 0.755 | |
| 119 | Self-doubt | This robot makes me feel insecure (or doubt myself). | | 0.793 | | | 0.722 | |
| 49 | Being upset | This robot upsets me. | | 0.772 | | | 0.746 | |
| 12 | Stress | This robot makes me feel stressed. | | 0.769 | | | 0.731 | |
| 102 | Fear | I am afraid of this robot. | | 0.763 | | | 0.723 | |
| 78 | Unpleasantness | This robot makes me feel unpleasant. | | 0.745 | | | 0.749 | |
| 25 | Dehumanization | **I would feel dehumanized when interacting with this robot.** | | 0.742 | | | 0.708 | |
| 50 | Anger | This robot angers me. | | 0.727 | | | 0.753 | |
| 103 | Creepiness | This robot is creepy. | | 0.720 | | | 0.690 | |
| 114 | Freedom restriction | **This robot restricts or limits me.** | | 0.718 | | | 0.698 | |
| 30 | Sadness | This robot makes me feel sad. | | 0.714 | | | 0.717 | |
| 58 | Danger | This robot is dangerous. | | 0.710 | | | 0.672 | |
| 111 | Boycott | I would ban the use of this robot. | | 0.701 | | | 0.722 | |
| 72 | Negative affect | **I feel negative toward this robot.** | | 0.686 | | | 0.699 | |
| 117 | Insignificance | This robot can make humans feel insignificant or not needed. | | 0.685 | 0.441 | | 0.589 | 0.354 |
| 140 | Disconnection | This robot disconnects humans from one another. | | 0.675 | | | 0.630 | |
| 135 | Confusion | **This robot makes me feel confused.** | | 0.671 | | | 0.710 | |
| 136 | People judging the use of robots | I think using this robot is wrong. | | 0.670 | | | 0.727 | |
| 33 | Loneliness | This robot makes me feel lonely. | | 0.669 | | | 0.671 | |
| 115 | Societal issues | **This robot can have negative social implications.** | | 0.661 | | | 0.614 | |
| 81 | Dissatisfaction | This robot brings me dissatisfaction. | | 0.660 | | | 0.697 | |
| 92 | Being gross | This robot is gross. | | 0.660 | | | 0.682 | |
| 148 | Existential questioning | This robot makes me question life and existence. | | 0.659 | 0.352 | | 0.631 | |
| 113 | Immorality | **This robot is immoral.** | | 0.657 | | | 0.657 | |
| 126 | Privacy | **This robot violates privacy (e.g., is too intrusive or invasive).** | | 0.656 | | | 0.693 | |

**Table 3 (continued) | Summary of key findings (Studies 3 and 4): psychological processes, items corresponding to each process and the output of EFAs performed on the items across two participant samples**

| Item no. | Psychological process | Item | Sample 1 (UK) | | | Sample 2 (US) | | |
|---|---|---|---|---|---|---|---|---|
| | | | P | N | C | P | N | C |
| 104 | Humans lacking control | **I would lack or lose control when using or interacting with this robot.** | 0.655 | | | | 0.650 | |
| 76 | Abnormal | **This robot is abnormal.** | 0.645 | | | | 0.610 | |
| 91 | Disgust | **This robot is disgusting.** | 0.641 | | | | 0.695 | |
| 48 | Damage reputation | **This robot could damage my reputation.** | 0.637 | | | | 0.671 | |
| 79 | Hate | I would hate dealing with this robot. | 0.636 | | | | 0.651 | |
| 147 | Human being tired of robot | This robot makes me feel tired or exhausted. | 0.631 | | | | 0.688 | |
| 11 | Avoidance | I would avoid this robot. | 0.630 | | | | 0.604 | |
| 52 | Protection of self | **I would want to protect myself when interacting with this robot.** | 0.612 | | | | 0.585 | |
| 134 | Mixed feelings | I would have mixed feelings toward this robot. | 0.607 | | | | 0.628 | |
| 118 | Replacement | This robot can make humans feel replaced. | 0.600 | 0.501 | | | 0.533 | 0.418 |
| 74 | Shyness | I would feel shy around this robot. | 0.599 | | | | 0.643 | |
| 31 | Guilt | This robot makes me feel guilty. | 0.594 | | | | 0.638 | |
| 145 | Embarrassment | **I would feel embarrassed or ashamed if I had to interact with this robot.** | 0.579 | | | | 0.719 | |
| 142 | Robots contribute to human degeneration | **This robot can contribute to human degeneration (e.g., make people become lazy, use less of their mental and physical capacity, etc.).** | 0.570 | | | | 0.559 | |
| 1 | Impatience | I would feel impatient when interacting with this robot. | 0.559 | | | | 0.647 | |
| 105 | Unpredictability | This robot is unpredictable. | 0.558 | | | | 0.549 | |
| 51 | Redundancy | This robot will make human jobs redundant. | 0.536 | 0.535 | | | 0.490 | 0.380 |
| 21 | Robot damage | I would be inclined to harm or damage this robot. | 0.535 | | 0.320 | | 0.633 | |
| 112 | Unethical activities | This robot could be used for unethical activities. | 0.528 | | | | 0.478 | |
| 73 | Disappointment | This robot is disappointing. | 0.525 | −0.342 | | | 0.663 | |
| 16 | Caution | I would be cautious or careful with this robot. | 0.499 | | | | 0.381 | |
| 22 | Verbal abuse of robots | I would likely be verbally abusive toward this robot. | 0.483 | | | | 0.632 | |
| 139 | Dependence (on robots or technology) | This robot creates dependence in humans. | 0.447 | 0.388 | | | 0.416 | |
| 120 | Human interaction substitute | This robot substitutes human interaction. | 0.429 | | | | 0.336 | |
| 138 | Uselessness | This robot is useless. | 0.405 | −0.564 | | | 0.600 | −0.348 |
| 131 | Social comparison | I compare whether this robot is better than humans. | 0.345 | 0.400 | | | 0.368 | |
| 133 | Boredom | This robot is boring. | 0.343 | | | | 0.478 | |
| 3 | Inefficiency | This robot is inefficient in what it does. | 0.330 | −0.410 | | | 0.484 | |
| 144 | Authentic self | I can be my authentic self around this robot. | −0.373 | | | | | 0.374 |
| 10 | Confidence | I would feel confident in this robot. | −0.375 | 0.466 | | 0.412 | | 0.426 |
| 80 | Satisfaction | I am satisfied with this robot. | −0.379 | 0.403 | | 0.483 | | 0.371 |
| 69 | Coexistence | I could coexist with this robot. | −0.381 | | | | | |
| 121 | Trust | I would trust this robot. | −0.393 | 0.342 | | 0.523 | | |
| 70 | Acceptance | I would be accepting of this robot. | −0.491 | 0.333 | | 0.429 | −0.343 | 0.372 |
| 4 | Performance | **This robot can effectively achieve a certain result or a specified outcome.** | | 0.734 | | | | 0.669 |
| 5 | Usefulness | This robot is useful. | | 0.703 | | | | 0.603 |
| 6 | Help | This robot is helpful. | | 0.662 | | | | 0.592 |
| 7 | Accuracy | **This robot is accurate in what it does.** | | 0.642 | | | | 0.570 |
| 2 | Complexity | **This robot can do complex tasks.** | | 0.641 | | | | 0.484 |
| 8 | Financial costs | **This robot reduces costs.** | | 0.632 | | | | 0.545 |
| 54 | Speed | **This robot is fast at what it does.** | | 0.605 | | | | 0.527 |

**Table 3 (continued) | Summary of key findings (Studies 3 and 4): psychological processes, items corresponding to each process and the output of EFAs performed on the items across two participant samples**

| Item no. | Psychological process | Item | Sample 1 (UK) | | | Sample 2 (US) | | |
|---|---|---|---|---|---|---|---|---|
| | | | P | N | C | P | N | C |
| 93 | Future orientation | I think this robot is the future. | | | 0.603 | | | 0.488 |
| 94 | Progress | I associate this robot with progress. | | | 0.583 | | | 0.517 |
| 109 | Social good | This robot is a benefit to society. | | | 0.545 | 0.424 | | 0.400 |
| 128 | Time freedom | This robot frees up my time to do other things. | | | 0.531 | 0.375 | | 0.417 |
| 55 | Level of advancement | **This robot is advanced**. | | | 0.516 | | | 0.526 |
| 127 | Easier life | This robot makes my life easier. | | | 0.516 | 0.478 | | 0.392 |
| 43 | Being impressed | This robot impresses me. | | | 0.510 | 0.412 | | 0.409 |
| 110 | Corporate social responsibility (CSR) | Any benefits gained from this robot should be shared with or passed onto society. | | | 0.504 | | | 0.428 |
| 141 | Robots augment human capabilities | This robot can augment human capabilities. | | | 0.471 | | | 0.424 |
| 20 | Endurance | This robot has endurance (e.g., never tires, runs nonstop, etc.). | | | 0.462 | | | 0.580 |
| 101 | Monitoring | I would monitor this robot to make sure it functions properly. | | | 0.426 | | | 0.389 |
| 15 | Human alertness | I would feel alert with this robot. | | | 0.398 | | | 0.397 |
| 97 | Bias | This robot is not biased. | | | 0.387 | | | 0.440 |
| 53 | Robot superiority | This robot is superior to humans. | | | 0.363 | 0.353 | | |
| 44 | Novelty | This robot is novel. | | | | | | |
| 56 | Human superiority | Whenever I am given a choice, I will choose a human over this robot. | | | | | | |
| 75 | Unusualness | This robot is unusual. | | | | | | |
| 90 | Cleanliness | I would find this robot sanitary. | | | | | | 0.360 |
| 106 | Dominance | I am dominant over this robot. | | | | | | |
| 108 | Humans having control | I would have control over this robot. | | | | | | |
| 124 | Autonomy | This robot is autonomous. | | | | | | |
| 137 | Objectivity | This robot is objective. | | | | | | |
| | | Variance explained (%) | 15.995 | 17.891 | 10.422 | 20.736 | 17.224 | 8.586 |
| | | Eigenvalues | 23.832 | 26.658 | 15.529 | 30.897 | 25.663 | 12.793 |
| | | P | – | | | – | | |
| | | N | −0.192 | – | | −0.192 | – | |
| | | C | 0.471 | −0.417 | – | 0.470 | −0.245 | – |

P, N and C refer to the dimensions (that is, factors) that comprise positive, negative and competence-related psychological processes, respectively, regarding robots. Values under each factor correspond to standardized factor loadings; only loadings with absolute values ≥0.320 are reported for clarity. The psychological processes and corresponding items are ordered according to item loadings on the three factors, while item no. corresponds to the number they were assigned when they were created. Items in bold were those selected for the PRR scale tested in Study 5 (Table 4). Coefficients for factors P, N and C at the bottom of the table denote correlations between factors. All items were scored on a seven-point Likert scale (1, strongly disagree; 7, strongly agree).

To aid the interpretation of the mechanisms, below we summarize the mediated effects from Table 7 that successfully explain a portion of the relationship between the key individual differences and PNC dimensions.

For the positive dimension, GRP[149] was a positive predictor because people scoring higher on this trait valued the risks associated with robot adoption (GRP_M3) and were curious to see how robots would change the world (GRP_M4). Moreover, IDAQ[150] was a positive predictor because people scoring higher on this trait generally felt positive towards inanimate entities with human features (IDAQ_M3), and because interaction with such entities helped them fulfil the need to experience strong emotions regularly (IDAQ_M2). FMPS_PE[151] was also a positive predictor due its association with valuing robots because they were closer to perfection than humans (FMPS_PE_M1), and also because they could help humans fulfil their own high expectations (FMPS_PE_M2) and could help humans cope with their own high expectations of themselves (FMPS_PE_M6).

For the negative dimension, PANAS_TNA[152] was a positive predictor because people scoring high on this trait were more likely to be in a state of activated displeasure (for example, feeling scared and upset; 12-PAC_AD[158]). Furthermore, SD3_P[153] was a positive predictor because people scoring high on this trait were also more likely to be in the state of activated displeasure (12-PAC_AD[158]), had negative feelings towards other people's inventions (SD3_P_M2) and felt inferior towards technologies in which they were not proficient (SD3_P_M3). For ERQ_ES[154] and IDAQ[150], we did not manage to explain the mechanism behind their relationship with the negative dimension.

For the competence dimension, ATQ_AP[155] was a positive predictor because people scoring high on this trait were more likely to value exceptional skills and competencies (ATQ_AP_M5). PVQ5X_SS[156] was also a positive predictor because it was associated with people linking advanced technology (for example, robots and machines) with how powerful society is (PVQ5X_SS_M4).

## Table 4 | ESEMs of the PRR scale (Study 5)

| Item no. | Sample 1 (UK) | | | Sample 2 (US) | | |
|---|---|---|---|---|---|---|
| | P | N | C | P | N | C |
| 116 | 0.801 | | | 0.756 | | |
| 132 | 0.760 | | | 0.746 | | |
| 88 | 0.758 | | | 0.769 | | |
| 35 | 0.742 | | | 0.756 | | |
| 84 | 0.740 | | | 0.789 | | |
| 32 | 0.682 | | | 0.729 | | |
| 40 | 0.674 | | | 0.588 | | |
| 149 | 0.672 | | | 0.654 | | |
| 27 | 0.658 | | | 0.662 | | |
| 96 | 0.655 | | | 0.626 | | |
| 107 | 0.644 | | | 0.638 | | |
| 66 | 0.587 | | | 0.604 | | |
| 63 | 0.561 | | | 0.536 | | |
| 125 | 0.557 | | | 0.523 | | |
| 17 | 0.546 | | | 0.496 | | |
| 59 | | 0.799 | | | 0.816 | |
| 113 | | 0.756 | | | 0.656 | |
| 25 | | 0.750 | | | 0.787 | |
| 114 | | 0.746 | | | 0.724 | |
| 13 | | 0.733 | | | 0.785 | |
| 48 | | 0.723 | | | 0.709 | |
| 145 | | 0.719 | | | 0.695 | |
| 135 | | 0.715 | | | 0.713 | |
| 126 | | 0.709 | | | 0.698 | |
| 76 | | 0.694 | | | 0.658 | |
| 115 | | 0.685 | | | 0.682 | |
| 72 | | 0.680 | | | 0.760 | |
| 104 | | 0.675 | | | 0.708 | |
| 91 | | 0.673 | | | 0.701 | |
| 52 | | 0.653 | | | 0.687 | |
| 142 | | 0.595 | | | 0.622 | |
| 7 | | | 0.762 | | | 0.644 |
| 4 | | | 0.701 | | | 0.606 |
| 54 | | | 0.681 | | | 0.660 |
| 2 | | | 0.643 | | | 0.659 |
| 55 | | | 0.617 | | | 0.619 |
| 8 | | | 0.483 | | | 0.514 |
| Factor | | | | | | |
| P | – | | | – | | |
| N | −0.239 | – | | −0.183 | – | |
| C | 0.386 | −0.431 | – | 0.423 | −0.352 | – |

**Model fit**

Sample 1, $\chi^2(558)=1{,}918.764$, $P<0.001$, SRMR=0.028, CFI=0.927, RMSEA=0.047, 90% CI [0.045, 0.049]

Sample 2, $\chi^2(558)=1{,}839.997$, $P<0.001$, SRMR=0.029, CFI=0.927, RMSEA=0.046, 90% CI [0.043, 0.048]

Values under each factor correspond to standardized factor loadings; only loadings ≥0.32 are reported for clarity. The items to which the numbers (no.) correspond can be seen in Table 3. Coefficients for factors P, N and C at the bottom of the table (above model fit) denote the standardized loadings of the factors on each other. All factors also yielded good to excellent Cronbach's α-values (Sample1, positive, α=0.927; negative, α=0.943; competence, α=0.818; Sample2, positive, α=0.923; negative, α=0.943; competence, α=0.802).

## Table 5 | Measurement invariance tests of the PRR scale for country: United Kingdom versus United States; robot example: A versus B; gender: female versus male; age: below median versus median and above; and employment status: employed versus unemployed (Study 5)

| Invariance model | SRMR | ΔSRMR | CFI | ΔCFI | RMSEA | ΔRMSEA |
|---|---|---|---|---|---|---|
| **Sample 1 (UK) versus Sample 2 (US)** | | | | | | |
| Configural | 0.028 | – | 0.927 | – | 0.046 | – |
| Metric | 0.033 | 0.005 | 0.926 | 0.001 | 0.045 | 0.001 |
| Scalar | 0.035 | 0.002 | 0.923 | 0.003 | 0.045 | <0.001 |
| **Robot example A versus B (Sample 1)** | | | | | | |
| Configural | 0.031 | – | 0.924 | – | 0.048 | – |
| Metric | 0.038 | 0.007 | 0.925 | 0.001 | 0.046 | 0.002 |
| Scalar | 0.038 | <0.001 | 0.924 | 0.001 | 0.046 | <0.001 |
| **Robot example A versus B (Sample 2)** | | | | | | |
| Configural | 0.032 | – | 0.927 | – | 0.046 | – |
| Metric | 0.040 | 0.008 | 0.925 | 0.002 | 0.045 | 0.001 |
| Scalar | 0.040 | <0.001 | 0.924 | 0.001 | 0.044 | 0.001 |
| **Gender: female versus male (Sample 1)** | | | | | | |
| Configural | 0.031 | – | 0.926 | – | 0.048 | – |
| Metric | 0.041 | 0.010 | 0.924 | 0.002 | 0.047 | 0.001 |
| Scalar | 0.042 | 0.001 | 0.920 | 0.004 | 0.047 | <0.001 |
| **Gender: female versus male (Sample 2)** | | | | | | |
| Configural | 0.032 | – | 0.927 | – | 0.046 | – |
| Metric | 0.041 | 0.009 | 0.926 | 0.001 | 0.044 | 0.002 |
| Scalar | 0.041 | <0.001 | 0.923 | 0.003 | 0.045 | 0.001 |
| **Age: <48 (median) versus ≥48 years (Sample 1)** | | | | | | |
| Configural | 0.031 | – | 0.925 | – | 0.048 | – |
| Metric | 0.039 | 0.008 | 0.924 | 0.001 | 0.046 | 0.002 |
| Scalar | 0.040 | 0.001 | 0.919 | 0.005 | 0.047 | 0.001 |
| **Age: <45 (median) versus ≥45 years (Sample 2)** | | | | | | |
| Configural | 0.033 | – | 0.921 | – | 0.048 | – |
| Metric | 0.041 | 0.008 | 0.919 | 0.002 | 0.047 | 0.001 |
| Scalar | 0.043 | 0.002 | 0.915 | 0.004 | 0.047 | <0.001 |
| **Employment status: employed versus unemployed (Sample 1)** | | | | | | |
| Configural | 0.031 | – | 0.926 | – | 0.048 | – |
| Metric | 0.039 | 0.008 | 0.924 | 0.002 | 0.047 | 0.001 |
| Scalar | 0.039 | <0.001 | 0.923 | 0.001 | 0.046 | 0.001 |
| **Employment status: employed versus unemployed (Sample 2)** | | | | | | |
| Configural | 0.032 | – | 0.925 | – | 0.047 | – |
| Metric | 0.039 | 0.007 | 0.924 | 0.001 | 0.045 | 0.002 |
| Scalar | 0.039 | <0.001 | 0.923 | 0.001 | 0.045 | <0.001 |

The symbol Δ refers to the absolute value of a change in fit indices for an invariance model relative to the previous (that is, metric minus configural; scalar minus metric). For robot example, 'A' indicates that the robot example for the domain to which participants from Study 5 were randomly allocated belonged to one of the two stimulus sets used in the present research, while 'B' indicates that the robot example belonged to the other stimulus set (Supplementary Table 7). For gender, very few participants identified themselves as 'other' or did not disclose any information (Table 1), and they were therefore randomly classified as either 'female' or 'male' so they could be used in invariance testing. For employment status the category 'employed' includes those participants who were self-employed or working for an employer. For use of robots at work we could not analyse measurement invariance because of the insufficient number of participants who used robots at work (Table 1). However, for Study 6, in which sample sizes were larger (Table 1), we tested measurement invariance for this variable and for additional participant characteristics assessed in that study (educational attainment, income, political orientation: liberal versus conservative, ethnic identity and relationship status). Measurement invariance was met in all cases (Supplementary Table 10).

**Table 6 | Main individual difference predictors of the positive, negative and competence dimensions (Study 7)**

| Variable | b | s.e. b | 99% CI | t | P | $f^2$ |
|---|---|---|---|---|---|---|
| DV, positive dimension | | | | | | |
| Model 1: GRP positively predicts DV | | | | | | |
| (constant) | 2.938 | 0.086 | 2.717–3.159 | 34.329 | <0.001 | 1.102 |
| GRP | 0.145 | 0.035 | 0.054–0.236 | 4.099 | <0.001 | 0.016 |
| Model 2: IDAQ positively predicts DV | | | | | | |
| (constant) | 2.791 | 0.068 | 2.615–2.967 | 40.888 | <0.001 | 1.564 |
| IDAQ | 0.178 | 0.023 | 0.120–0.236 | 7.868 | <0.001 | 0.058 |
| Model 3: FMPS_PE positively predicts DV | | | | | | |
| (constant) | 2.786 | 0.107 | 2.511–3.061 | 26.137 | <0.001 | 0.639 |
| FMPS_PE | 0.158 | 0.034 | 0.071–0.245 | 4.684 | <0.001 | 0.021 |
| DV, negative dimension | | | | | | |
| Model 4: PANAS_TNA positively predicts DV | | | | | | |
| (constant) | 2.090 | 0.078 | 1.888–2.292 | 26.734 | <0.001 | 0.669 |
| PANAS_TNA | 0.124 | 0.047 | 0.003–0.245 | 2.634 | 0.009 | 0.006 |
| Model 5: IDAQ positively predicts DV | | | | | | |
| (constant) | 2.133 | 0.060 | 1.979–2.287 | 35.714 | <0.001 | 1.193 |
| IDAQ | 0.056 | 0.020 | 0.005–0.107 | 2.840 | 0.005 | 0.008 |
| Model 6: SD3_P positively predicts DV | | | | | | |
| (constant) | 1.707 | 0.098 | 1.456–1.959 | 17.496 | <0.001 | 0.286 |
| SD3_P | 0.296 | 0.048 | 0.172–0.420 | 6.152 | <0.001 | 0.035 |
| Model 7: ERQ_ES positively predicts DV | | | | | | |
| (constant) | 2.041 | 0.083 | 1.827–2.255 | 24.586 | <0.001 | 0.565 |
| ERQ_ES | 0.064 | 0.021 | 0.010–0.117 | 3.085 | 0.002 | 0.009 |
| DV, competence dimension | | | | | | |
| Model 8: ATQ_AP positively predicts DV | | | | | | |
| (constant) | 4.470 | 0.151 | 4.079–4.860 | 29.527 | <0.001 | 0.816 |
| ATQ_AP | 0.157 | 0.029 | 0.081–0.233 | 5.345 | <0.001 | 0.027 |
| Model 9: PVQ5X_SS positively predicts DV | | | | | | |
| (constant) | 4.859 | 0.106 | 4.586–5.133 | 45.797 | <0.001 | 1.962 |
| PVQ5X_SS | 0.097 | 0.024 | 0.034–0.160 | 3.962 | <0.001 | 0.015 |

DV, dependent variable. Model 1, $R^2$=0.015; Model 2, $R^2$=0.055; Model 3, $R^2$=0.020; Model 4, $R^2$=0.006; Model 5, $R^2$=0.007; Model 6, $R^2$=0.034; Model 7, $R^2$=0.009; Model 8, $R^2$=0.026; and Model 9, $R^2$=0.014. All models had 1,069 residual degrees of freedom. In all models we used t-tests (two-sided) to assess the significance of the coefficients, with the significance criterion being $P < 0.010$ based on the Benjamini–Yekutieli correction[216,217] for multiple comparisons. The table contains raw P values that are statistically significant if they meet this benchmark; therefore, all nine predictors reached statistical significance. $f^2$ denotes Cohen's $f^2$ effect size[218]. GRP and FMPS_PE were measured on a 1–5 scale (1, strongly disagree; 5, strongly agree); IDAQ was measured on a 0–10 scale (0, not at all; 10, very much); PANAS_TNA was measured on a 1–5 scale (1, strongly disagree; 5, strongly agree); SD3_P was measured on a 1–5 scale (1, disagree strongly; 5, agree strongly); ERQ_ES and ATQ_AP were measured on a 1–7 scale (1, strongly disagree; 7, strongly agree); and PVQ5X_SS was measured on a 1–6 scale (1, not like me at all; 6, very much like me).

## Discussion

In this section we first discuss (1) our findings and their contributions in relation to previous research to achieve inductive integration[77] and then (2) the main limitations (for a detailed discussion see Supplementary Discussion).

Starting with Phase 1, we first discuss the robot definition (Table 2) and then the domains (Table 2). Regarding the definition, ours and that of IEEE[22] both conceptualize robots as devices or entities that can perform different tasks (Part 1, Table 2), emphasize that robots can have different degrees of autonomy (Part 2, Table 2) and include robots' composition (Part 5, Table 2). However, the two definitions also have unique elements: ours includes robots' durability (Part 3, Table 2) and positive/negative attributes (Part 4, Table 2) whereas the IEEE definition includes robots' capability to form robotic systems. Overall, although our definition is somewhat more nuanced, both definitions are remarkably aligned, which indicates that experts and lay individuals perceive robots similarly.

Regarding the domains in which robots operate we have identified 28 (Table 2), which is more than professional organizations usually propose (for example, the IEEE lists 18 domains on their website, https://robots.ieee.org/learn/types-of-robots/). However, this is not surprising because our list was intentionally nuanced to enable the identification of a comprehensive sample of robots, and we hope that other scholars will adopt it in their research for this purpose. It is important to emphasize that, despite the meticulous procedure used to develop the list, it is possible that (1) we failed to identify more niche domains and (2) the number of domains might increase as technology advances.

Continuing with Phase 2, we first compare the psychological processes of the PNC model (Table 3) against those reported in previous research and then discuss the model (Tables 3 and 4) more specifically. In general, participants evoked the processes identified in the literature reviewed in the Introduction, including positive feelings such as happiness[10,33] (Item 32); negative feelings such as anxiety[27] (Item 13); performance[48] and usefulness[19] (Items 4 and 5); anthropomorphism[36,159] (Items 24 and 27); and various approach[66] (Items 22 and 40) or avoidance[26] (Items 11 and 52) behaviours (Table 3). Importantly, participants also described many infrequent or previously unidentified processes. For example, they indicated that robots contribute to human degeneration (Item 142); lead to existential questioning (Item 148); make people feel dehumanized (Item 25); help humans self-improve (Item 132); and restrict freedom (Item 114).

One of the main contributions of our research is showing that these seemingly highly diverse psychological processes fall under three dimensions: positive (P), negative (N) and competence (C) (Tables 3 and 4). In general, previous research on human–robot relationships and interactions has focused on studying and measuring specific psychological reactions to robots (for example, safety, anthropomorphism, animacy, intelligence, likeability and various social attributes[159,160]) but did not attempt to identify all these reactions and investigate them under an all-encompassing construct of psychological processes. In that regard, the PNC model can be seen as an integrative framework that links and organizes an exhaustive list of psychological processes, both those that researchers have already studied separately and the less common ones generated by our participants. We believe that our model moves the field forward, not only through this integration but also by enabling researchers to systematically study psychological processes regarding robots by (1) using the PNC as a guide to inform the design of future research on these processes and (2) employing the PRR scale to measure them.

One of the most interesting insights spawned by the PNC model stems from comparing it with the stereotype content model (SCM[161,162]). According to the SCM, people form impressions of other humans along two dimensions: warmth (that is, positive and negative social characteristics) and competence (that is, a person's ability to successfully accomplish tasks). Although our model is broader than the SCM because it comprises all psychological processes rather than only social and intellectual characteristics, the competence dimensions from the two models are thematically comparable whereas the positive and negative attributes from the SCM's warmth dimension are broadly aligned with our positive and negative dimensions. These comparisons suggest that (1) people use similar criteria when forming impressions of robots and humans and (2) robots' similarity to humans does not play a role in this regard, because many of our stimuli depicted non-humanoid robots (Supplementary Table 7).

**Table 7 | All variables tested as mediators and their mediated effects (in parentheses), listed under the relevant individual difference predictors of the positive, negative and competence dimensions (Study 7)**

**DV, positive dimension**

Mediators tested for predictor GRP

GRP_M1 (ab=0.001, 99% CI$_{bootstrapped}$ [−0.006, 0.010], ab$_%$=0.007). I think robots do not pose a risk to me.

GRP_M2 (ab<0.001, 99% CI$_{bootstrapped}$ [−0.016, 0.016], ab$_%$=0.002). I think robots do not pose a risk to society.

**GRP_M3 (ab=0.057, 99% CI$_{bootstrapped}$ [0.029, 0.093], ab$_%$=0.393).** I think robot adoption has its risks, but these risks are what make robots appealing.

**GRP_M4 (ab=0.013, 99% CI$_{bootstrapped}$ [0.001, 0.032], ab$_%$=0.090).** I am curious to see how robots will change the world.

GRP_M5 (ab=−0.003, 99% CI$_{bootstrapped}$ [−0.018, 0.009], ab$_%$=−0.021). I think robot adoption has its risks, but the potential rewards are high.

GRP_M6 (ab=−0.001, 99% CI$_{bootstrapped}$ [−0.010, 0.006], ab$_%$=−0.007). The benefits of robots outweigh their risks.

GRP_M7 (ab=0.020, 99% CI$_{bootstrapped}$ [−0.004, 0.048], ab$_%$=0.138). (1) I feel that technology helps me to align with my ideal self; (2) I feel that technology helps me to succeed in my endeavours.[a]

Mediators tested for predictor IDAQ[b]

IDAQ_M1 (ab=0.004, 99% CI$_{bootstrapped}$ [−0.009, 0.019], ab$_%$=0.022). When I interact with a non-human entity (e.g., robots, machines, nature, animals), I can experience strong emotions that I would normally experience toward human beings.

**IDAQ_M2 (ab=0.021, 99% CI$_{bootstrapped}$ [0.008, 0.039], ab$_%$=0.118).** Interacting with non-human entities (e.g., robots, machines, nature, animals) helps me fulfil the need to experience strong emotions regularly.

**IDAQ_M3 (ab=0.035, 99% CI$_{bootstrapped}$ [0.016, 0.060], ab$_%$=0.197).** When I see a non-human entity (e.g., robots, machines, nature, animals) that has human characteristics, I experience positive feelings.

IDAQ_M4 (ab=−0.001, 99% CI$_{bootstrapped}$ [−0.007, 0.004], ab$_%$=−0.006). When I see a non-human entity (e.g., robots, machines, nature, animals) that has human characteristics, I experience negative feelings.

Mediators tested for predictor FMPS_PE

**FMPS_PE_M1 (ab=0.031, 99% CI$_{bootstrapped}$ [0.009, 0.058], ab$_%$=0.196).** I value robots because they are closer to perfection than humans.

**FMPS_PE_M2 (ab=0.027, 99% CI$_{bootstrapped}$ [0.006, 0.055], ab$_%$=0.171).** I value robots because I believe they can help me fulfil my own high expectations.

FMPS_PE_M3 (ab=0.011, 99% CI$_{bootstrapped}$ [−0.024, 0.048], ab$_%$=0.070). I value robots because I believe they can help me fulfil my parents' high expectations.

FMPS_PE_M4 (ab=0.009, 99% CI$_{bootstrapped}$ [−0.013, 0.032], ab$_%$=0.057). I value robots because their superiority over humans allows me to become superior over others.

FMPS_PE_M5 (ab=0.026, 99% CI$_{bootstrapped}$ [−0.012, 0.065], ab$_%$=0.165). I value robots because I believe they help me better cope with my parents' high expectations of me.

**FMPS_PE_M6 (ab=0.033, 99% CI$_{bootstrapped}$ [0.011, 0.062], ab$_%$=0.209).** I value robots because I believe they help me better cope with my own high expectations of myself.

**DV, negative dimension**

Mediators tested for predictor PANAS_TNA

**Activated Displeasure—12-PAC_AD (ab=0.231, 99% CI$_{bootstrapped}$ [0.029, 0.450], ab$_%$=1.863)**; Deactivated Displeasure—12-PAC_DD (ab=−0.019, 99% CI$_{bootstrapped}$ [−0.180, 0.147], ab$_%$=−0.153); Displeasure—12-PAC_D (ab=−0.030, 99% CI$_{bootstrapped}$ [−0.228, 0.173], ab$_%$=−0.242); Unpleasant Activation—12-PAC_UA (ab=−0.073, 99% CI$_{bootstrapped}$ [−0.231, 0.081], ab$_%$=−0.589); Unpleasant Deactivation—12-PAC_UD (ab=0.033, 99% CI$_{bootstrapped}$ [−0.054, 0.130], ab$_%$=0.266)[c]. Measured using 12-point affect circumplex (12-PAC)[158].

Mediators tested for predictor IDAQ[b]

IDAQ_M1 (ab=0.001, 99% CI$_{bootstrapped}$ [−0.011, 0.014], ab$_%$=0.018); IDAQ_M2 (ab=0.008, 99% CI$_{bootstrapped}$ [−0.002, 0.022], ab$_%$=0.143); IDAQ_M3 (ab=−0.011, 99% CI [−0.026, −0.001], ab$_%$=−0.196); IDAQ_M4 (ab=−0.005, 99% CI$_{bootstrapped}$ [−0.021, 0.011], ab$_%$=−0.089)[d]. Same as for 'DV, positive dimension: IDAQ'.

Mediators tested for predictor SD3_P

SD3_P_M1 (ab=−0.011, 99% CI$_{bootstrapped}$ [−0.075, 0.050], ab$_%$=−0.037). I tend to have negative feelings toward other people.

**SD3_P_M2 (ab=0.122, 99% CI$_{bootstrapped}$ [0.070, 0.183], ab$_%$=0.412).** I tend to have negative feelings toward other people's creations and inventions.

**SD3_P_M3 (ab=0.022, 99% CI$_{bootstrapped}$ [0.005, 0.050], ab$_%$=0.074).** Using technologies that I am not proficient in makes me feel inferior.

SD3_P_M4 (ab=−0.008, 99% CI$_{bootstrapped}$ [−0.036, 0.020], ab$_%$=−0.027). Technology can expose me for who I am.

**12-PAC_AD (ab=0.064, 99% CI$_{bootstrapped}$ [0.010, 0.136], ab$_%$=0.216)**; 12-PAC_DD (ab=−0.005, 99% CI$_{bootstrapped}$ [−0.061, 0.051], ab$_%$=−0.017); 12-PAC_D (ab=−0.028, 99% CI$_{bootstrapped}$ [−0.098, 0.037], ab$_%$=−0.095); 12-PAC_UA (ab=−0.017, 99% CI$_{bootstrapped}$ [−0.067, 0.022], ab$_%$=−0.057); 12-PAC_UD (ab=−0.003, 99% CI$_{bootstrapped}$ [−0.037, 0.029], ab$_%$=−0.010).[c] Same as for 'DV, negative dimension: PANAS_TNA' (ref. 158).

Mediators tested for predictor ERQ_ES

ERQ_ES_M1 (ab=0.007, 99% CI$_{bootstrapped}$ [−0.010, 0.029], ab$_%$=0.109). At the moment, I feel mentally exhausted.

ERQ_ES_M2 (ab=0.007, 99% CI$_{bootstrapped}$ [−0.012, 0.027], ab$_%$=0.109). At the moment, I feel emotionally exhausted.

12-PAC_AD (ab=0.017, 99% CI$_{bootstrapped}$ [−0.002, 0.043], ab$_%$=0.266); 12-PAC_DD (ab=−0.005, 99% CI$_{bootstrapped}$ [−0.026, 0.013], ab$_%$=−0.078); 12-PAC_D (ab=−0.008, 99% CI$_{bootstrapped}$ [−0.037, 0.014], ab$_%$=−0.125); 12-PAC_UA (ab=−0.007, 99% CI$_{bootstrapped}$ [−0.023, 0.003], ab$_%$=−0.109); 12-PAC_UD (ab=−0.001, 99% CI$_{bootstrapped}$ [−0.013, 0.009], ab$_%$=−0.016).[c] Same as for 'DV, negative dimension: PANAS_TNA' (ref. 158).

**Table 7 (continued) | All variables tested as mediators and their mediated effects (in parentheses), listed under the relevant individual difference predictors of the positive, negative and competence dimensions (Study 7)**

| DV, competence dimension |
|---|
| Mediators tested for predictor ATQ_AP |
| ATQ_AP_M1 (ab=0.008, 99% CI$_{bootstrapped}$ [−0.004, 0.026], ab$_\%$=0.051). I value robots that can help me perform better than others. |
| ATQ_AP_M2 (ab=0.015, 99% CI$_{bootstrapped}$ [−0.007, 0.042], ab$_\%$=0.096). I value robots that can help me become better at a task, goal, or skill that I want to accomplish or master. |
| ATQ_AP_M3 (ab=−0.010, 99% CI$_{bootstrapped}$ [−0.029, 0.005], ab$_\%$=−0.064). When evaluating other people, it is important to me how good they are at what they do. |
| ATQ_AP_M4 (ab=0.012, 99% CI$_{bootstrapped}$ [−0.001, 0.030], ab$_\%$=0.076). When evaluating robots, it is important to me how good they are at what they do. |
| **ATQ_AP_M5 (ab=0.020, 99% CI$_{bootstrapped}$ [0.002, 0.042], ab$_\%$=0.127).** I highly value exceptional skills and competencies. |
| ATQ_AP_M6 (ab=0.003, 99% CI$_{bootstrapped}$ [−0.021, 0.027], ab$_\%$=0.019). When I see a human that can accomplish something challenging, I react strongly to it. |
| ATQ_AP_M7 (ab=−0.006, 99% CI$_{bootstrapped}$ [−0.032, 0.020], ab$_\%$=−0.038). When I see a robot that can accomplish something challenging, I react strongly to it. |
| ATQ_AP_M8 (ab=0.011, 99% CI$_{bootstrapped}$ [−0.012, 0.037], ab$_\%$=0.070). When I see the potential for robots to improve human life, I get excited. |
| ATQ_AP_M9 (ab=0.017, 99% CI$_{bootstrapped}$ [−0.010, 0.047], ab$_\%$=0.108). When I encounter robots or other inventions that can better my life, I react strongly to it. |
| ATQ_AP_M10 (ab=0.018, 99% CI$_{bootstrapped}$ [−0.002, 0.043], ab$_\%$=0.115). I am thrilled when seeing robots helping society to achieve tasks that are often difficult to accomplish. |
| Mediators tested for predictor PVQ5X_SS |
| PVQ5X_SS_M1 (ab=0.009, 99% CI$_{bootstrapped}$ [−0.005, 0.027], ab$_\%$=0.093). I think advanced technology (e.g., robots, machines, devices) can make the country more powerful. |
| PVQ5X_SS_M2 (ab=0.016, 99% CI$_{bootstrapped}$ [−0.008, 0.042], ab$_\%$=0.165). With effective use of advanced technology (e.g., robots, machines, devices), the country maintains its strength to defend its citizens. |
| PVQ5X_SS_M3 (ab=0.004, 99% CI$_{bootstrapped}$ [−0.003, 0.015], ab$_\%$=0.041). I think advanced technology (e.g., robots, machines, devices) can create order and stability. |
| **PVQ5X_SS_M4 (ab=0.015, 99% CI$_{bootstrapped}$ [<0.001, 0.036], ab$_\%$=0.155).** Advanced technology (e.g., robots, machines, devices) is a reflection of how powerful our society is. |
| PVQ5X_SS_M5 (ab=0.007, 99% CI$_{bootstrapped}$ [−0.001, 0.019], ab$_\%$=0.072). Being surrounded by advanced technology (e.g., robots, machines, devices) that is effective at what it does makes me feel safe. |

For each mediator we first present its name, followed by its mediated effect (ab) in parentheses. Mediated effects are presented in raw units. For example, for GRP_M3 (ab=0.057, 99% CI [0.029, 0.093], ab$_\%$=0.393) the mediated effect ab indicates that for one unit increase in GRP as a predictor, the positive dimension increased by 0.057 units, which is the effect that can be accounted for by the mediator (GRP_M3). For an easier understanding of the magnitude of each mediated effect, ab$_\%$ is also reported and indicates the percentage of the total effect between a predictor and DV (that is, coefficients b in Table 6) explained by the mediator. In some cases ab$_\%$ can exceed 1 (that is, 100%), which means that the effect travelling through the mediator is larger than the total effect itself. A mediated effect is significant only if its 99% CI$_{bootstrapped}$ does not contain 0 (ref. 157). Some mediated effects (ab and ab$_\%$) are negative; this means they are in the opposite direction to the effect between a predictor and DV, and therefore do not explain their relationship. All mediators that successfully explained a portion of the relationship between a predictor and DV (that is, mediated effects that are positive and whose 99% CI$_{bootstrapped}$ does not contain 0) are presented in bold typeface. [a]The two items for GRP_M7 were averaged into a composite score. [b]For IDAQ as a predictor we used the same mediators for the positive and negative dimensions, considering that we wanted to ensure that any potential differences between the mechanisms for these two dimensions are not a consequence of different mediators being used in the mediation models. [c]The five 12-PAC mediators capture state affect because they were assessed in relation to how people currently felt. [d]Although the mediated effect of IDAQ_M3 was significant, the direction of this effect was negative (ab=−0.011) and thus opposite to the positive direction of the relationship between IDAQ and the negative domain (Table 6). Therefore, the mediator failed to explain this relationship.

Ending with Phase 3, we discuss our findings on individual difference predictors (Tables 6 and 7) in relation to the previous relevant literature. In this respect, researchers found that extraversion, openness and anthropomorphism predicted positive responses to robots[163–166]; the need for cognition predicted lower negative attitudes towards robots[167]; and animal reminder disgust, neuroticism and religiosity predicted experiencing robots as eerie[168]. Among these, our research corroborated only the positive relationship between anthropomorphism and positive responses (Table 6).

We also went beyond previous research by discovering many relationships not easily anticipated by theory. For example, although we had a sound rationale behind each predictor (Supplementary Table 11) it would have been difficult to foresee psychopathy as the most robust predictor of the negative dimension (Table 6)[153]. We also did not expect that one of the main mechanisms behind negative robot perceptions would be negative feelings towards other people's creations and the state of activated displeasure, which mediated the relationship between psychopathy and the negative PNC dimension (Table 7). Therefore, using a data-driven approach allowed us to generate unexpected insights, thus diversifying the body of knowledge on psychological reactions to robots[79,81,98].

There are several limitations to this research. First, the stimuli were not physical robots but their depictions. These stimuli hold ecological validity because people often interact with robots indirectly (for example, via social media or various websites), and many psychological processes may therefore be shaped in this manner. Nonetheless, previous research showed that direct interaction with robots impacts people's experiences[27,169,170]. Therefore, based on the present findings it is not known whether our taxonomy applies to the physical counterparts of the robots depicted by our stimuli, and investigating this is currently unachievable because many of these robots are inaccessible for in-person research due to their size, cost, limited production or potential use as weapons (for example, industrial and military robots). However, this research may be possible in the future if such robots become more accessible.

Second, participants were from Western, educated, industrialized, rich and democratic[171] countries (United Kingdom and United States). Because our research proposed and investigated a construct (that is, psychological processes regarding robots) from scratch, our priority was to establish its foundations. Combining the investigation of cultural differences with this agenda using equally meticulous methods would have exceeded the scope of a single article. Nevertheless, because measurement invariance analyses showed that the PNC model applies to individuals regardless of their income, age, education, use of robots at work, political orientation, ethnic identity and relationship status, it is plausible that the model would generalize to countries that differ from the United Kingdom or United States on these population characteristics. Conducting an in-depth examination of this question will be a crucial step as this research topic progresses.

Third, we recruited online participants who are inherently more confident with technology. Whereas this might have influenced the findings, alternative modes of recruitment (for example, laboratory) would have yielded smaller and less representative participant samples[172–176]. Furthermore, to reduce the chance of technological proficiency biasing the findings, all machine learning models controlled for a variable indicative of technological proficiency (that is, previous frequency of interaction with robots; Supplementary Tables 11 and 12).

Finally, rapid technological development might make robots with an embodiment similar to humans able to perform and simulate all human activities, thereby substantially changing how people perceive robots. However, since our comparison of the PNC model and SCM[161,162] indicates that people form impressions of robots and humans in a similar manner, it is unlikely that robots becoming more like humans will have a notable impact on the structure of our model. Even if it does, the PNC can be updated via the same methodological procedures we used.

## Methods

This research complies with the ethics policy and procedures of the London School of Economics and Political Science (LSE), and has also been approved by its Research Ethics Committee (no. 20810). Informed consent was obtained from all participants and they were compensated for their participation. Table 1 summarizes key participant information. In Studies 4–6, participants were recruited to be reasonably representative of the UK/US populations for age, gender and geographical region, and in Study 1 (Sample 1) for gender only. More comprehensive breakdowns of participant information and the criteria used for representative sampling are available in Supplementary Tables 1 and 2.

To be included in analyses, participants had to pass seriousness checks[177], instructed-response items (for example, please respond with 'somewhat disagree')[178–180], understanding checks in which they identified the main research topic (that is, robots) amongst dummy topics (for example, animals or art) and completely automated public Turing tests to tell computers and humans apart (CAPTCHAs), used to safeguard against bots[181]. The number of these quality checks varied per study. For seriousness checks: Study 1, two (one per sample); Study 2, one; Study 3, one; Study 4, two (one per sample); Study 5, two (one per sample); Study 6, one; and Study 7, two (one per wave). For instructed-response items: Study 1, six (two in Sample 1 and four in Sample 2); Studies 2 and 3, none; Study 4, eight (four per sample); Study 5, four (two per sample); Study 6, three; and Study 7, three (two in wave 1 and one in wave 2). For understanding checks: Study 1, two (one per sample); Study 2, one; Study 3, one; Study 4, two (one per sample); Study 5, two (one per sample); Study 6, one; and Study 7, none. For CAPTCHA: Study 1, one (in Sample 2); Study 2, one; Study 3, one; Study 4, two (one per sample); Study 5, two (one per sample); Study 6, one; and Study 7, two (one per wave).

In Studies 4–7, which were quantitative, we employed pairwise deletion for missing data because various simulations showed that this does not bias the type of analyses we used when missing data are infrequent (≤5%)—even in smaller participant samples (for example, 240)—and larger samples are generally more robust to missing data[182,183]. In our analyses, the percentage of participants with missing data never exceeded 1.95.

The analyses using machine learning models (Study 6) did not rely on distributional assumptions due to cross-validation[184], and neither did the mediation analyses (Study 7) due to bootstrapped confidence intervals used to test mediated effects[157]. All other quantitative analyses assumed a normal distribution of data. Because formal normality tests are sensitive to small deviations that do not bias findings[134], we assumed variables to be normal if they had skewness between −2 and 2 and kurtosis between −7 and 7 (refs. [185–187]). All the required variables met these criteria (Supplementary Tables 18–23). Given the large sample sizes we used, even severe deviations from normality would not compromise the validity of statistical inferences[157,188,189].

Next, we succinctly describe the methods of the studies in each phase (for a more comprehensive description, see Supplementary Methods). Study 7 was preregistered on 12 December 2021 via the Open Science Framework (OSF) and can be accessed using this link: https://osf.io/nejvm?view_only = 79b6eeee42e24cb2a977927712b-dcdd2. There were no deviations from the preregistered protocol. Data and analysis codes for all studies are also publicly available via the OSF using the following link: https://osf.io/2ntdy/?view_only = 2cacc7b1cf2141cf8c343f3ee28dab1d

### Phase 1: mapping a comprehensive content space of robots

**Study 1.** Sample size. To determine Sample 1 size we relied on previous work showing that, in qualitative research, samples having 30–50 participants tend to reach the point of data saturation, which means that the addition of further participants produces little new information[190–195]. We recruited a considerably larger sample (266; Table 1) to ensure that the study detected all important robot characteristics because the robot definition we wanted to develop was essential for all subsequent studies. For Sample 2 we recruited 100 participants (Table 1), which is comparable to other studies using hierarchical clustering[196,197] given the lack of guidelines on optimal sample sizes for this technique (for additional insights based on simulations, see Supplementary Methods).

Procedure. In Sample 1, participants first answered the consent form after which they were presented with three items that elicited robot characteristics. In the following order, they were asked to: (1) state the first thing that comes to mind when they think about a robot; (2) define in their own words what a robot is; and (3) list as many characteristics they associate with robots that they could think of. At the end we assessed participant information, including gender, age, employment status and use of robots at work (Table 1). In Sample 2, after answering the consent form, participants were exposed to 277 robot characteristics produced by Sample 1 (Supplementary Table 3) and were asked to sort them into groups based on similarity. In this regard, participants were provided with up to 60 empty boxes representing different groups into which they could drag the characteristics they perceived as being similar. At the end, participant information was assessed as for Sample 1.

Analytic approach. We first extracted robot characteristics generated by Sample 1 participants for the three questions described in the Study 1 procedure and then rephrased those that were stated vaguely (for example, 'appearance of thought') into a more precise formulation (for example, 'appears to think on its own'). Next, we deleted all characteristics that were identical and therefore redundant. However, we included many items that were overlapping or similar (for example, 'performs actions' and 'performs certain actions') to ensure that the potential content space of robot characteristics was sampled in detail (for the final list of 277 characteristics see Supplementary Table 3). The characteristics, as sorted into categories by Sample 2 participants, were subjected to hierarchical cluster analysis for categorical data[114–116]: a dissimilarity matrix was computed using Gower's distance[198,199], clusters were produced using Ward's linkage method[200,201] and the optimal number of clusters was determined via the mean silhouette width approach using the partitioning around medoids algorithm[114,202,203]. The five clusters that emerged were then arranged into the robot definition (Table 2).

**Study 2.** Sample size. To determine sample size we followed the same guidelines as for Study 1 (Sample 1) that considered the point of data saturation in qualitative research.

Procedure. After completing the consent form, participants were presented with the robot definition developed in Study 1. They were then asked to think about and list any domains that came to mind in which humans can encounter and/or interact with robots. It was explained that, by 'domains', we mean any area of human life and human

activity in which people encounter, interact with, use, are helped by and/or are substituted by robots. At the end, participant information was assessed as in Study 1 (Table 1).

Analytic approach. To identify the domains we performed an inductive qualitative content analysis on participants' responses[117–121]: we first created a list of all domain items identified by participants (see Supplementary Results, subsection 'Additional analysis output') and then arranged these items into common categories that correspond to the domains of robot use. The first author created the initial list of categories from the domain items. The list was revised by the remaining authors and, eventually, it was consolidated by all three authors. To ensure that no important domains had been omitted we also consulted the classification of robots proposed by IEEE (https://robots.ieee.org/learn/types-of-robots/), the list of industries and sectors endorsed by the International Labor Organization (https://www.ilo.org/global/industries-and-sectors/lang--en/index.htm) and the articles from our literature review.

## Phase 2: creating the taxonomy of psychological processes

**Study 3.** Sample size. To determine sample size we followed the same guidelines as for Study 1 (Sample 1) and Study 2. Because Study 3 aimed to identify a comprehensive range of psychological processes towards robots, which was a crucial step of our research, we recruited a substantially larger sample than required (350; Table 1) to ensure that even highly infrequent processes were detected.

Procedure. Participants first completed the consent form and were then randomly allocated to five out of the 28 domains we developed (Table 2). After reading the definition of robots (Table 2), we prompted them to think about robots from the allocated domains by writing about interactions they had with such robots, or else about interactions they could imagine or were exposed to via media. To assess participants' psychological processes, we then asked them to list and describe feelings they had experienced (for affective responses), thoughts they had (for cognitive responses) and actions they engaged in (for behavioural responses) when they interacted with any robots they could think of from each domain, or to write about feelings, thoughts and actions they could conceive in case they had never interacted with these robots. At the end, participant information was assessed as in Studies 1 and 2 (Table 1).

Analytic approach. We implemented iterative categorization[122]. This qualitative analysis involved first splitting participants' responses to questions assessing their psychological processes into key points (that is, separate issues or thoughts—for example, 'I think this will be the future')—and then grouping these points into themes based on similarity. Out of 334 participants who were included in analyses (Table 1), only four produced merely meaningless responses that could not be analysed and the remaining 330 generated 10,332 valid key points (approximately 31 per participant) that were analysed.

**Study 4.** Sample size. Because power analyses are difficult to implement for EFAs before any parameters are known, to determine the sizes of Samples 1 and 2 we consulted various resources that estimated the optimal sample size for EFAs (for a more comprehensive description see Supplementary Methods). Because the size of 1,500 met all the estimates, we recruited the samples required to reach this number after accounting for exclusions (Table 1).

Procedure. The procedure for both samples was identical. After answering the consent form, participants were randomly allocated to a domain (Table 2) and received a specific example of a robot from that domain (Supplementary Table 7) that included an image and description approximately eight lines long. For the sex domain, two robot examples were created (one male and one female) and participants assigned to this domain were randomly allocated to one. Participants were then asked to answer 149 items (Table 3), presented in a randomized order, about the robot in question. At the end, participant information was assessed as in Studies 1, 2 and 3 (Table 1).

Analytic approach. For both samples we planned several steps to determine the optimal factor structure. First, the Kaiser–Meyer–Olkin measure of sampling adequacy and Bartlett's test of sphericity were required to show that our data are suitable for EFAs[125]. Second, to determine the preliminary number of factors for examining in EFAs, we used parallel analysis[126,127,204], very simple structure[128], Velicer map[129], optimal coordinates[130], acceleration factor[130], Kaiser rule[131] and visual inspection of scree plots[132]. This was advisable because consulting several criteria allows understanding of the range within which lies the optimal number of factors potentially[82,135,136,205,206]. Next, we aimed to evaluate the largest factor solution identified in the previous step against several statistical benchmarks using maximum-likelihood EFAs[104,105] with Kaiser-normalized[123] promax rotation[106,124]. Namely, the factor solution was required to produce only valid factors (that is, those that have at least three items with loadings ≥0.5 and cross-loadings <0.32) to be accepted[105,125,133,134]. If these criteria were not met, we aimed to decrease the number of factors by one and evaluate the new solution—this procedure would continue until a satisfying solution was identified. Finally, the accepted factor structure also had to have factors that are coherent and easy to interpret[135,136]. Importantly, this approach to selecting the best structure is not only statistically and semantically viable but has precedent in previous taxonomic research[82,205].

**Study 5.** Sample size. To determine sample size we used Monte Carlo simulations[207] based on the data from Samples 1 and 2 (Study 4). Details are available in Supplementary Methods.

Procedure. The procedure for both samples was identical. After answering the consent form, participants were randomly allocated to one robot example. The randomization procedure was the same as in Study 4 except that there were two (rather than one) possible robot examples per domain (Supplementary Table 7). The sex domain had four examples—two male and two female robots. The descriptions of robots were also consistent with Study 4. Participants were then asked to answer the 37 selected items (Table 4), presented in a randomized order, about the robot in question. At the end, participant information was assessed as in Studies 1, 2, 3 and 4 (Table 1).

Analytic approach. The maximum-likelihood with robust standard errors estimator[137,138] was implemented using ESEM[82,107,108,208] to test model fit. Target rotation with all cross-loadings specified as targets of zero was chosen[139,140]. The following fit criteria were used[141–143]: SRMR < 0.05, excellent fit; SRMR = 0.05–0.08, good fit; SRMR > 0.08, poor fit; CFI > 0.95, excellent fit; CFI = 0.90–0.95, good fit; CFI < 0.90, poor fit; RMSEA < 0.06, excellent fit; RMSEA = 0.06–0.10, good fit; and RMSEA > 0.10, poor fit. For testing of configural measurement invariance the same fit criteria were used. For metric invariance, changes in SRMR, CFI and RMSEA were required to be ≤0.030, 0.010 and 0.015, respectively, and, for scalar invariance, ≤0.015, 0.010 and 0.015, respectively[144,146].

## Phase 3: examining individual difference predictors

**Study 6.** Sample size. For machine learning algorithms combined with cross-validation there are no straightforward guidelines for compution of power analyses. Simulations showed that, for the tenfold cross-validations we were planning to use, a sample of 2,000 leads to high generalizability (that is, a likelihood that the results will apply to other samples from the same population) without inflating the time taken to run the models[209]. Therefore, we aimed to recruit a sample that would have approximately 2,200 participants after accounting for exclusions, in case of any additional missing data.

Procedure. After answering the consent form, participants were randomly allocated to one robot example as in Study 5 and asked to answer the PRR scale items (Table 4) presented in a randomized order. They then completed measures that assessed the 79 individual differences we tested as predictors (Supplementary Table 11), ranging from general personality traits, such as BIG 5 (ref. 210) or approach

temperament[155], to more specific ones, such as psychopathy[153]. We also measured covariates for inclusion in the models alongside the individual differences (that is, familiarity with the robot, frequency of interaction, descriptive norms, injunctive norms, age, income and political orientation; Supplementary Table 11). Finally participant information was assessed as in the previous studies, with the addition of education level, ethnic identity and relationship status (Supplementary Table 1).

Analytic approach. We implemented a rigorous multistep procedure to select the most predictive individual differences. Using the caret package[109,110] in R, we computed the following 11 machine learning models for each PNC dimension separately: linear least squares, ridge, lasso, elastic net, *k*-nearest neighbours, regression trees, conditional inference trees, random forest, conditional random forest, neural networks and neural networks with a principal component step. For each model, tenfold cross-validation[184,211–214] was implemented and all 79 individual differences plus covariates were used as predictors.

The most predictive models were selected using root-mean-square error (r.m.s.e.)[109,110,184]. For each PNC dimension, the model with the highest r.m.s.e. was identified and the remaining models were compared with it using paired-samples *t*-tests (Bonferroni corrected *α* of 0.00167 was used as the significance criterion). Ultimately, the model with the highest r.m.s.e. and those not significantly different from it were identified as the most predictive models. For each of these models we first identified the 30 most important predictors using the VarImp function in R[110] and then identified individual differences that appeared in the top 30 across all models.

Based on the linear least-squares model—which is in essence a linear regression algorithm combined with cross-validation and thus outputs *P* values—we retained only the most important individual differences identified in the previous step that were also statistically significant after applying false-discovery rate correction[148]. We used this approach because in Study 7 we aimed to replicate the selected predictors using linear regressions; therefore, we wanted to further minimize the likelihood that these predictors are false positives.

**Study 7.** Sample size. Because this study tested the key predictors identified in Study 6, sample size was estimated using power analyses[215] based on the parameters from that study (Supplementary Methods).

Procedure. The study consisted of two waves. In wave 1, participants first completed the consent form and were then presented with, in a randomized order, the measures assessing the most predictive individual differences identified in Study 7 (Table 6). Finally participant information was assessed as in Studies 1–5. Approximately 4 days after completing wave 1, participants were invited to participate in wave 2. They first completed the consent form and were then presented with the items measuring the mediators (Table 7) in a randomized order. Subsequently, they were randomly allocated to a robot example as in Studies 5 and 6 and asked to answer the PRR scale items (Table 4), presented in a randomized order.

Analytic approach. To test whether the key individual differences predicted the relevant PNC dimensions we used linear regressions, one per predictor (Table 6). Furthermore, to identify the most important mediators we used the Process package (Model 4 (ref. 157)) to perform parallel mediation analyses (that is, with all potential mediators analysed together for the relevant predictor; Table 7), percentile-bootstrapped with 10,000 samples. In line with the Benjamini–Yekutieli correction[216,217], the significance criterion was 0.01 for the regression analyses whereas for the mediated effects we used 99% confidence intervals that are the equivalent of this criterion.

**Reporting summary**

Further information on research design is available in the Nature Portfolio Reporting Summary linked to this article.

## Data availability

The data that support the findings from all studies, as well as the materials used, are publicly available via the OSF: https://osf.io/2ntdy/?view_only = 2cacc7b1cf2141cf8c343f3ee28dab1d), except for the stimuli used in Studies 4–7, which can be obtained from the corresponding author on request.

## Code availability

The codes for all the analyses for the studies conducted are publicly available via the OSF using the following link: https://osf.io/2ntdy/?view_only = 2cacc7b1cf2141cf8c343f3ee28dab1d.

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

## Acknowledgements

This research was supported by the LSE Research Support Fund awarded to J.E.B. and D.K. It was also supported by internal LSE departmental funding awarded by the Department of Management to J.E.B. and by the Department of Psychological and Behavioural Science to D.K. The funders had no role in study design, data collection and analysis, decision to publish or preparation of the manuscript.

## Author contributions

D.K. was responsible for conceptualization (lead), data curation (lead), formal analysis (lead), funding acquisition (lead), investigation (lead), methodology (lead), project administration (lead), validation (lead), visualization (lead), writing of the original draft (lead) and writing review and editing (lead). J.E.B. was responsible for conceptualization (supporting), formal analysis (supporting), funding acquisition (lead), investigation (supporting), methodology (lead), validation (lead) and writing review and editing (supporting). A.D. was responsible for conceptualization (lead), formal analysis (supporting), funding acquisition (supporting), investigation (lead), methodology (lead), validation (lead), visualization (lead), writing of the original draft (supporting) and writing review and editing (lead).

## Competing interests

The authors declare no competing interests.

## Additional information

**Correspondence and requests for materials** should be addressed to Dario Krpan.

# Reporting Summary

## Statistics

For all statistical analyses, confirm that the following items are present in the figure legend, table legend, main text, or Methods section.

| n/a | Confirmed | |
|---|---|---|
| ☐ | ☒ | The exact sample size (*n*) for each experimental group/condition, given as a discrete number and unit of measurement |
| ☐ | ☒ | A statement on whether measurements were taken from distinct samples or whether the same sample was measured repeatedly |
| ☐ | ☒ | The statistical test(s) used AND whether they are one- or two-sided<br>*Only common tests should be described solely by name; describe more complex techniques in the Methods section.* |
| ☐ | ☒ | A description of all covariates tested |
| ☐ | ☒ | A description of any assumptions or corrections, such as tests of normality and adjustment for multiple comparisons |
| ☐ | ☒ | A full description of the statistical parameters including central tendency (e.g. means) or other basic estimates (e.g. regression coefficient) AND variation (e.g. standard deviation) or associated estimates of uncertainty (e.g. confidence intervals) |
| ☐ | ☒ | For null hypothesis testing, the test statistic (e.g. *F*, *t*, *r*) with confidence intervals, effect sizes, degrees of freedom and *P* value noted<br>*Give P values as exact values whenever suitable.* |
| ☒ | ☐ | For Bayesian analysis, information on the choice of priors and Markov chain Monte Carlo settings |
| ☒ | ☐ | For hierarchical and complex designs, identification of the appropriate level for tests and full reporting of outcomes |
| ☐ | ☒ | Estimates of effect sizes (e.g. Cohen's *d*, Pearson's *r*), indicating how they were calculated |

*Our web collection on statistics for biologists contains articles on many of the points above.*

## Software and code

Policy information about availability of computer code

**Data collection**
Data for all studies were collected using Qualtrics (https://www.qualtrics.com/). The following Qualtrics versions were used: Study 1 (Sample 1) - version [February, March 2019]; Study 1 (Sample 2) - version [July 2019]; Study 2 - version [September 2019]; Study 3 - version [October 2019]; Study 4 (Sample 1) - version [March 2021]; Study 4 (Sample 2) - version [April 2021]; Study 5 (Sample 1) - version [June, July, 2021]; Study 5 (Sample 2) - version [June, July 2021]; Study 6 - version [September, October, 2021]; Study 7 - version [December 2021].
The original surveys in Qualtrics that were used for data collection are available via the following link (for each study, check the folder "Materials"): https://osf.io/2ntdy/?view_only=2cacc7b1cf2141cf8c343f3ee28dab1d
Qualtrics is a commercial survey platform, and we did not use any of our own code to collect the data.

**Data analysis**
The data were analyzed using the following software (and packages where relevant):
Study 1 (Sample 2) - R software (version 4.2.1): packages dplyr (version 1.1.1), cluster (version 2.1.3), dendextend (version 1.16.0), and ape (version 5.6-2).
Study 4 (Samples 1 and 2) - R software (version 4.2.1): packages psych (version 2.2.5), paran (version 1.5.2), nFactors (version 2.4.1.1), GPArotation (version 2023.3-1), and MVN (version 5.9).
Study 5 (Samples 1 and 2) - Mplus (version 8.6); R software (version 4.2.1): packages psych (version 2.2.5) and MVN (version 5.9); Bifactor Indices Calculator (version 10-4-2017).
Study 6 - R software (version 4.2.1): packages psych (version 2.2.5), caret (version 6.0-93), tidyverse (version 1.3.2), rsample (version 1.1.0), skimr (version 2.1.4), ggplot2 (version 3.4.2), ggthemes (version 4.2.4), ggpubr (version 0.4.0), glmnet (version 4.1-4), party (version 1.3-11), randomForest (version 4.7-1.1), forecast (version 8.18), fabletools (version 0.3.2), h2o (version 3.38.0.1), and MVN (version 5.9); Mplus (version 8.6).
Study 7 - R software (version 4.2.1): packages psych (version 2.2.5), sensemakr (version 0.1.4), sjPlot (version 2.8.14), and MVN (version 5.9); SPSS (version 23): package Process (version 3.4.1).

For more information about R software, see https://www.r-project.org/; for more information about Mplus, see https://www.statmodel.com/; for more information about SPSS, see https://www.ibm.com/products/spss-statistics; and for more information about Bifactor Indices Calculator, see https://uknowledge.uky.edu/edp_tools/1/
Studies 1 (Sample 1), 2, and 3 involved only qualitative analyses, and therefore no statistical software was used in these studies. All analyses codes are available via the following link: https://osf.io/2ntdy/?view_only=2cacc7b1cf2141cf8c343f3ee28dab1d

For manuscripts utilizing custom algorithms or software that are central to the research but not yet described in published literature, software must be made available to editors and reviewers. We strongly encourage code deposition in a community repository (e.g. GitHub). See the Nature Portfolio guidelines for submitting code & software for further information.

## Data

Policy information about availability of data

All manuscripts must include a data availability statement. This statement should provide the following information, where applicable:

- Accession codes, unique identifiers, or web links for publicly available datasets
- A description of any restrictions on data availability
- For clinical datasets or third party data, please ensure that the statement adheres to our policy

The data that support the findings from all the studies are publicly available via the Open Science Framework (OSF) using the following link: https://osf.io/2ntdy/?view_only=2cacc7b1cf2141cf8c343f3ee28dab1d

## Human research participants

Policy information about studies involving human research participants and Sex and Gender in Research.

| Reporting on sex and gender | Findings apply to both male and female gender. Participants also had the option to identify themselves using a different gender label, although few of them selected that option. Gender was assessed in each study using self-reports; Table 1 includes the breakdown of gender and other demographic information for all studies. In a nutshell, as can be seen from Table 1, the number of males and females who participated in each study was similar. Overall, 5283 female and 4965 male participants completed the studies, whereas 4832 female and 4392 male participants were included in analyses. Moreover, 43 participants who completed the studies identified themselves as "Other", whereas 42 participants who were included in analyses identified themselves as "Other". Finally, the data for 11 participants who completed the studies, and 8 participants who were included in analyses, were missing or were not disclosed. |
|---|---|
| Population characteristics | See above. |
| Recruitment | The information regarding how participants were recruited is summarized in Table 1 in the article. In Studies 1 (Sample 1), 4 (Samples 1 and 2), 5 (Samples 1 and 2), and 6, participants were recruited via Pureprofile (https://www.pureprofile.com/). In Studies 1 (Sample 2), 2, and 3, participants were recruited via Amazon Mechanical Turk (https://www.mturk.com). In Study 7, participants were recruited via Prolific (https://www.prolific.co/). Therefore, all data were collected using online participant panels. It is possible that such panels attract specific types of participants, and that certain self-selection biases might have been present (e.g., individuals who are more confident with technology being more likely to participate). We aimed to minimize any potential impact of such biases on our findings by measuring various relevant variables and using them in statistical analyses. For example, one of the covariates we used in the machine learning models (Study 6; see Supplementary Tables 11-12) was a variable indicative of technological proficiency involving robots (i.e., people's previous frequency of interaction with robots). It is also important to emphasize that these panels generally contain more diverse (Buhrmester, Kwang, & Gosling, 2011; Buhrmester, Talaifar, & Gosling, 2018; Casler, Bickel, & Hackett, 2013) and more attentive participants than typical university research pools (Hauser & Schwarz, 2016) and are widely used in psychological and behavioural sciences research. Moreover, it is not a given that other modes of recruitment (e.g., participant pools of university research labs) would avoid technological proficiency as a potential bias of online recruitment panels, considering that research participation is often advertised online and participants such as students tend to use technology for their studies. |
| Ethics oversight | This research complies with the ethics policy and procedures of the London School of Economics and Political Science and has also been approved by its Research Ethics Committee (ref. 20810). |

Note that full information on the approval of the study protocol must also be provided in the manuscript.

# Field-specific reporting

Please select the one below that is the best fit for your research. If you are not sure, read the appropriate sections before making your selection.

☐ Life sciences     ☒ Behavioural & social sciences     ☐ Ecological, evolutionary & environmental sciences

For a reference copy of the document with all sections, see nature.com/documents/nr-reporting-summary-flat.pdf

# Behavioural & social sciences study design

All studies must disclose on these points even when the disclosure is negative.

**Study description**

- Study 1: Developing a definition of robots based on how participants perceive them. This study employed mixed-methods and therefore produced both qualitative and quantitative data. Two different samples of participants were tested. In Sample 1, participants were asked to generate as many characteristics of robots as possible. Then, we recruited Sample 2 and asked them to group the characteristics identified by the previous sample into common categories. Using hierarchical cluster analysis (Kaufman & Rousseeuw, 2005; Nielsen, 2016; Šulc & Řezanková, 2019), we then identified the main clusters that comprise the robot characteristics and used them to construct the robot definition.

- Study 2: Identifying all domains of human functioning in which robots operate. This study was qualitative and therefore produced qualitative data. In the study, we used the robot definition developed in Study 1 to identify a comprehensive list of all domains of human functioning in which robots can be encountered. Participants were presented with the definition and asked to generate all such domains they could think of. To develop an extensive inventory of domains, we analyzed their responses using inductive content analysis (Elo & Kyngäs, 2008; Elo et al., 2014; Hsieh & Shannon, 2005; Mayring, 2004; Vaismoradi, Turunen, & Bondas, 2013).

- Study 3: Mapping the content space of psychological processes toward robots. This study was qualitative and therefore produced qualitative data. In this study, the aim was to identify a comprehensive range of psychological processes regarding robots. Participants were asked to write about any feelings, thoughts, and behaviors they could think of in relation to robots from the domains developed in Study 2. Their responses were analyzed using iterative categorization (Neale, 2016) to generate the final list of psychological processes.

- Study 4: Establishing dimensions of the psychological processes. This was a quantitative study (i.e., it used a cross-sectional, correlational design) and therefore produced quantitative data. Two different samples of participants were tested. We randomly allocated participants from each sample to an example of a robot from one of the 28 domains established in Study 2 and asked them to answer questions measuring each of the 149 psychological processes established in Study 3 in relation to this specific robot. To identify the dimensions, the data were analyzed using exploratory factor analyses (EFAs; Schmitt, 2011).

- Study 5: Confirming the dimensions of the psychological processes. This was a quantitative study (i.e., it used a cross-sectional, correlational design) and therefore produced quantitative data. We tested two samples to confirm the dimensions established in Study 4 using exploratory structural equation modeling (ESEM; Asparouhov & Muthén, 2009).

- Study 6: Determining the main individual difference predictors of the dimensions confirmed in Study 5. This was a quantitative study (i.e., it used a cross-sectional, correlational design) and therefore produced quantitative data. To select the most predictive individual differences from the ones we tested, we employed a range of commonly used machine learning algorithms (e.g., lasso, random forests; Helwig, 2017; Jacobucci, Brandmaier, & Kievit, 2019; Joel et al., 2020; Kuhn, 2008, 2022) in combination with k-fold cross validation (de Rooij & Weeda, 2020).

- Study 7: Confirming the predictors and establishing the mechanism. This was a quantitative study (i.e., it used a longitudinal, correlational design in 2 waves) and therefore produced quantitative data. In this study, we aimed to explain the relationship between the most predictive individual differences from Study 6 and the dimensions of psychological responses regarding robots identified in Studies 4 and 5. The study therefore consisted of two waves. In wave 1, we measured the individual differences, and in wave 2 we first assessed a range of potential mediators and then asked participants to answer the items measuring the dimensions. Linear regressions and linear regression-based mediation analyses (Hayes, 2018) were used to analyze the data.

**Research sample**

As indicated under "Recruitment" (see the "Human research participants" section), the information regarding how participants were recruited is summarized in Table 1 in the manuscript. In Studies 1 (Sample 1), 4 (Samples 1 and 2), 5 (Samples 1 and 2), and 6, participants were recruited via Pureprofile (https://www.pureprofile.com/). In Studies 1 (Sample 2), 2, and 3, participants were recruited via Amazon Mechanical Turk (https://www.mturk.com/). In Study 7, participants were recruited via Prolific (https://www.prolific.co/). Therefore, all data were collected using online participant panels. In Studies 1 (Sample 1), 4 (Sample 1), and 5 (Sample 1) participants were UK adults, and in Studies 1 (Sample 2), 2, 3, 4 (Sample 2), 5 (Sample 2), 6, and 7 participants were US adults. Participants in Studies 4-6 were recruited to be reasonably representative of the UK/US populations in terms of age, gender, and geographical region, whereas for Study 1 (Sample 1) the focus was on gender only. Supplementary Tables 1-2 contain more comprehensive breakdowns of these variables, the criteria that were used to guide representative sampling, and various demographic characteristics. We targeted specifically UK and US samples because the type of online panels we used to recruit participants are typically able to provide large and in some cases reasonably representative samples from these countries, which can be more difficult when it comes to recruiting participant from other countries. As stated in the Discussion section of the present article when discussing the limitations, since our research proposed and investigated a construct (i.e., psychological processes regarding robots) from scratch, our priority was to establish its foundations, and combining the investigation of cultural differences with this agenda using equally meticulous methods would have exceeded the scope of a single article.

Overall, Table 1 in the article provides basic demographic information for our participants, whereas Supplementary Tables 1 and 2 contain more comprehensive information in this regard. Below we present mean age, standard deviation of age, and the number of female, male, other, and undisclosed participants who completed each study (see Table 1 in the article).
- Study 1 (Sample 1): 49.496, 13.598, 132, 133, 1, 0
- Study 1 (Sample 2): 36.510, 10.566, 42, 58, 0, 0
- Study 2: 36.257, 10.270, 31, 39, 0, 0
- Study 3: 40.693, 12.194, 193, 153, 1, 3
- Study 4 (Sample 1): 47.932, 16.611, 852, 812, 4, 0

- Study 4 (Sample 2): 48.004, 16.772, 976, 830, 2, 0
- Study 5 (Sample 1): 46.648, 16.616, 590, 601, 6, 3
- Study 5 (Sample 2): 46.656, 16.914, 616, 598, 5, 0
- Study 6: 47.405, 17.262, 1299, 1186, 15, 5
- Study 7: 42.910, 13.535, 552, 555, 9, 0

| | |
|---|---|
| Sampling strategy | As indicated above, participants were recruited via online panels commonly used in psychological and behavioural research (Prolific, Pureprofile, and Amazon Mechanical Turk). These and other online participant panels generally use some form of convenience sampling (e.g., Chandler & Shapiro 2016; Armitage & Eerola, 2020; see also https://researcher-help.prolific.co/hc/en-gb/articles/360009223133-Is-online-crowdsourcing-a-legitimate-alternative-to-lab-based-research-), and the sampling strategy used in the present research was therefore convenience sampling. More information about the composition of our participant samples is provided in the section "Research sample" above.<br><br>In the Methods section for each study in the article, there is a section on "Sample size" that explains how the sample size was predetermined (see also Supplementary Methods). For studies that had qualitative elements (Study 1, Sample 1; Study 2; and Study 3), we recruited sample sizes larger than 50 participants, given that simulations have indicated that sample sizes larger than 30-50 participants (Mayring, 2019; van Rijnsoever, 2017) tend to reach the point of data saturation, which implies that adding new participants beyond this number produces very little new information (Faulkner & Trotter, 2017). For Study 1 (Sample 2; see section "Sample Size" for that study in the article), in which we used hierarchical cluster analysis, the sample size was based on recent simulations, according to which the most important determinant of power seems to be the number of observations per cluster, with 20 observations yielding sufficient power to detect a cluster (Dalmaijer et al., 2022). For Study 4 (see section "Sample Size" for that study in the article), we consulted several resources to determine the number of participants to test for each sample because there is no consensus regarding sample size requirements for EFA (Costello & Osborne, 2005; Hogarty, Hines, Kromrey, Ferron, & Mumford, 2005; Kyriazos, 2018; MacCallum, Widaman, Zhang, & Hong, 1999; Reio Jr & Shuck, 2015). First, a few resources posit that the ratio of the number of participants to the number of items should be at least 10:1 (Everitt, 1975; Gorsuch, 1983; Reio Jr & Shuck, 2015). Second, some studies estimated that, if the ratio of the number of items to the number of factors is larger than 10:3, recruiting approximately 400 participants leads to high power, even under low communalities (MacCallum et al., 1999). Third, it has been proposed that a sample size larger than 300 is sufficient for a wide range of factor solutions (Dimitrov, 2012; Guadagnoli & Velicer, 1988). Our sample sizes for Study 4 met all these criteria. For Study 5, we determined the number of participants to test using Monte Carlo simulations (Muthén & Muthén, 2002) based on the data from Samples 1 and 2 (Study 4). Concerning Study 6, there are no clear guidelines for the use of machine learning algorithms combined with cross-validation regarding sample size and power. In a series of simulations, Song, Tang, and Wee (2021) showed that, for 10-fold cross-validations that we were planning to use, a sample size of 2000 leads to high generalizability (i.e., likelihood that the results will apply to other samples from the same population) without inflating time taken to run the models. We therefore aimed to recruit a sample that would result in roughly 2200 participants after applying the exclusion criteria, in case of any additional missing data. Finally, we determined the sample size for Study 7 by computing a-priori power analyses (Faul, Erdfelder, Buchner, & Lang, 2009) based on the data from Study 6. |
| Data collection | Qualtrics (https://www.qualtrics.com/) was used to collect the data (for the versions of Qualtrics that were used, see the "Data collection" field above). This is an online survey software widely used by universities across the world. Participants were anonymous and completed the study in their own surroundings. Participation was allowed on PCs, laptops, and tablets, but not on mobile phones. The researchers (i.e., authors of this paper) were not blinded to study predictions and aims. However, since the participants were anonymous and there was no contact between the researchers and participants, it is implausible that experimenter demand effects played a role in the present research. Importantly, since the present research used a data-driven approach as described in the Introduction section of the article, the majority of studies did not have a priori predictions. Only Study 5, in which we aimed to confirm the dimensions of psychological processes established in Study 4, and Study 7, in which we aimed to corroborate the main individual difference predictors identified in Study 6, were confirmatory. This is another reason why experimenter demand effects concerning study predictions were unlikely to play a role in the present research. |
| Timing | Start and stop dates for data collection in each study:<br>- Study 1 (Sample 1): 27 February – 4 March 2019<br>- Study 1 (Sample 2): 26 July – 27 July 2019<br>- Study 2: 24 September 2019<br>- Study 3: 21 October – 24 October 2019<br>- Study 4 (Sample 1): 4 March – 11 March 2021<br>- Study 4 (Sample 2): 8 April – 24 April 2021<br>- Study 5 (Sample 1): 29 June – 3 July 2021<br>- Study 5 (Sample 2): 24 June – 2 July 2021<br>- Study 6: 23 September – 15 October 2021<br>- Study 7: 13 December 2021 |
| Data exclusions | The exclusion criteria were pre-established (e.g., see pre-registration for Study 7: https://osf.io/nejvm?view_only=79b6eeee42e24cb2a977927712bdcdd2). They are comprehensively described in the Methods section in the article and in Supplementary Methods. In general, participants were excluded from analyses if they did not correctly answer seriousness checks (Aust, Diedenhofen, Ullrich, & Musch, 2013), instructed-response items (Kung, Kwok, & Brown, 2018; Meade & Craig, 2012; Thomas & Clifford, 2017), and understanding checks in which they were asked to identify the main topic of the study amongst a range of dummy topics. Table 1 in the article summarizes participants who completed the study and who were included in analyses after the exclusion criteria were applied. From the participants who completed the study, the following number of participants were excluded from data analyses:<br>- Study 1 (Sample 1): 42<br>- Study 1 (Sample 2): 5<br>- Study 2: 3 |

- Study 3: 16
- Study 4 (Sample 1): 140
- Study 4 (Sample 2): 271
- Study 5 (Sample 1): 93
- Study 5 (Sample 2): 111
- Study 6: 302
- Study 7: 45

Non-participation

Considering that participation in the present research took place anonymously and online, we only have knowledge of participants who completed the study (see Table 1 in the article). In some cases, online participants recruited via the online panels we used (Prolific, Pureprofile, and Amazon Mechanical Turk) test the survey and answer one or few questions and then leave - these data are captured under incomplete data but we are not aware of whether and how many of these participants are unique participants. Overall, non-participation data for the present research are not available.

Randomization

As can be seen under "Study description", the present research was not experimental. Therefore, there were no different conditions to which participants could be randomized. However, it is important to emphasize that in Studies 4-7, in which participants were allocated to robot examples from 28 possible robot domains, this allocation was random.

# Reporting for specific materials, systems and methods

We require information from authors about some types of materials, experimental systems and methods used in many studies. Here, indicate whether each material, system or method listed is relevant to your study. If you are not sure if a list item applies to your research, read the appropriate section before selecting a response.

## Materials & experimental systems

| n/a | Involved in the study |
|---|---|
| ☒ | ☐ Antibodies |
| ☒ | ☐ Eukaryotic cell lines |
| ☒ | ☐ Palaeontology and archaeology |
| ☒ | ☐ Animals and other organisms |
| ☒ | ☐ Clinical data |
| ☒ | ☐ Dual use research of concern |

## Methods

| n/a | Involved in the study |
|---|---|
| ☒ | ☐ ChIP-seq |
| ☒ | ☐ Flow cytometry |
| ☒ | ☐ MRI-based neuroimaging |

