## [Peer Review File · Nature Human Behaviour]

Peer Review Information

Journal: Nature Human Behaviour

Manuscript Title: The Positive-Negative-Competence (PNC) Model of Psychological Responses to Representations of Robots

Corresponding author name(s): Dario Krpan

Reviewer Comments & Decisions:

Decision Letter, initial version:
--

10th January 2023

Dear Dr Krpan,

Thank you once again for your manuscript, entitled "The Coming of a Brave New World: The Positive-Negative-Competence (PNC) Model of Psychological Responses to Robots", and for your patience during the peer review process.

Your Article has now been evaluated by 2 referees. You will see from their comments copied below that, although they find your work of potential interest, they have raised quite substantial concerns. In light of these comments, we cannot accept the manuscript for publication, but would be interested in considering a revised version if you are willing and able to fully address reviewer and editorial concerns.

We hope you will find the referees' comments useful as you decide how to proceed. If you wish to submit a substantially revised manuscript, please bear in mind that we will be reluctant to approach the referees again in the absence of major revisions. We are committed to providing a fair and constructive peer-review process. Do not hesitate to contact us if there are specific requests from the reviewers that you believe are technically impossible or unlikely to yield a meaningful outcome.

To guide the scope of the revisions, the editors discuss the referee reports in detail within the team, including with the chief editor, with a view to (1) identifying key priorities that should be addressed in revision and (2) overruling referee requests that are deemed beyond the scope of the current study. We hope that you will find the prioritised set of referee points to be useful when revising your study. Please do not hesitate to get in touch if you would like to discuss these issues further.

1) Our reviewers raise concerns about the conceptualization of the research. In specific, they are concerned that the concept being measured is not responses to robots, but ideas about robots. We ask that you follow Reviewer 2's advice and run at least one additional experiment with a representative

sample, that explicitly examines responses to robots (vs responses to the idea of robots). Any newly run experiments should be preregistered and have at least 80% power to detect the smallest meaningful effect size (not observed effect sizes from the existing studies).

2) Both reviewers raise concerns about the representativeness of the samples, and the generalizability of the findings (Reviewer 2). In your revision, please discuss the representativeness of your samples and how the fact that all participants came from the USA and UK might affect generalizability of the findings.

3) Reviewer 1 raises a number of concerns about data quality and methodological choices. Please carefully address these concerns and revise your manuscript to provide sufficient methodological and technical details in your revised manuscript.

If you wish to submit a suitably revised manuscript we would hope to receive it within 4 months. I would be grateful if you could contact us as soon as possible if you foresee difficulties with meeting this target resubmission date.

- Include a "Response to the editors and reviewers" document detailing, point-by-point, how you addressed each editor and referee comment. If no action was taken to address a point, you must provide a compelling argument. When formatting this document, please respond to each reviewer comment individually, including the full text of the reviewer comment verbatim followed by your response to the individual point. This response will be used by the editors to evaluate your revision and sent back to the reviewers along with the revised manuscript.
- Highlight all changes made to your manuscript or provide us with a version that tracks changes.

[REDACTED]

Thank you for the opportunity to review your work. Please do not hesitate to contact me if you have any questions or would like to discuss the required revisions further.

Sincerely,

Samantha Antusch

Samantha Antusch, PhD

Senior Editor
Nature Human Behaviour

Reviewer expertise:

Reviewer #1: responses to robots ; robot-human interactions ; psychology

Reviewer #2: machine learning ; human-robot interaction

REVIEWER COMMENTS:

Reviewer #1:

Remarks to the Author:

This manuscript reported 7 studies with a data-driven approach to develop the positive-negative-competence model of psychological responses to robots. Additionally, individual differences predicting the psychological responses were also explored. I enjoyed reading the manuscript. Although the whole manuscript is quite complicated with 7 studies, the authors described and explained them clearly. I really appreciate that extremely detailed description of the methods and results are provided in the supplement materials. However, there are a number of issues that dampened my enthusiasm.

1. I applaud the data driven approach. However, I do think that the study could have been benefited with a combination of data driven approach and theoretical support. Media equation and Computers as Social Actions (CASA) is well established framework to study robot. I am surprised that the authors did not cite this.

2. Additionally, the validity of any data driven approach depends significantly on the quality of the data as well as the representativeness of the data. The authors have done many right things to ensure data quality, e.g., seriousness check, etc. However, the representativeness of the data is not well addressed. The authors only reported gender and mean age of the sample characteristics. Educational level and prior direct experience with robot (not imagined) are two important factors related to psychological responses to robots. The online participant recruitment panel used in the current study seem to lack representativeness in this regard. Many of the studies measured employment status and use of robots at work but the data were not reported.

3. Sample 2 in study 1 were asked to sort 277 different robot characteristics generated by sample. This is a very challenging task to sort that many items. I have some reservations on the quality of the data. The average age of the participants in this sample is 36, compared to about 45 for the other studies (except study 2). Is that an indicator that only relatively younger participants are able to complete such a cognitively challenging task in the study?

4. Although the authors stated that it is purposeful to have overlapping domains in study 2, I am still not fully convinced why it is better than having more mutually exclusive domains. For instance, industry and manufacturing have a lot of overlapping. Why not merge them?

5. Study 3 had a large sample size (334 of the 350 participants remained). The authors mentioned that "Irrelevant points, including those that were meaningless or that merely repeated the name of the domain in question, were labelled in red and excluded from further analyses.". I wonder how many responses were generated by the 334 participants in total and how many of these responses are such irrelevant points. How many of the 334 participants generated usable data?

6. Study 4 participants were asked to answer 149 questions. Are those questions ordered randomly?

7. Also, for study 4, with such a large number of items for EFA, why didn't the authors choose to separate cognitive, affective, and behavioral responses rather than combine them together for the

EFA? Would separately analyzing the items of cognitive, affective, and behavioral responses reduce the over-factoring issue? I am also not fully convinced why the 3-factor solution should be chosen. This is key component of the manuscript and deserves more discussion.

8. Study 4 and 5 have two samples from different online participants panels. What is the reason not combining these two samples to conduct analysis?

9. The effect sizes of the individual differences are quite small. This deserves some discussion.

Reeves, B., & Nass, C. (1996). *The media equation: How people treat computers, television, and new media like real people*. Cambridge, UK.

Reviewer #2:

Remarks to the Author:

Conceptual novelty

The authors present a huge effort in identifying a number of concepts concerning how people perceive robots. This is interesting and unprecedented. There is a general issue about this work: what is investigated is not the attitude of people towards robots, but their attitude towards their concept of robot. It is not clear how much direct experience may have affected this, or media, or memes circulating in the specific parts of the society (not well identified except from country of origin) where the samples have been drawn.

Methodological novelty

The methodological apparatus is impressive and may be a guidance for further research.

Applied/Societal-/Policy-related Advance

The findings, although limited, may support decisions in both societal and industrial policies.

Evidence-based advance

The studies are impressive and support the findings, which are not much different from what was expected.

Data & methodology.

All data are available and the methodological processes deeply described

Preregistration

In some studies preregistration was applied and the authors followed it.

Appropriate use of statistics and treatment of uncertainties

The methodological aspects are correct.

Custom code: If the work includes custom code, does the code run as intended? If you are unable to access the code, please contact us.

Running the code was not needed for this paper.

Conclusions: Do you find that the conclusions and data interpretation are robust, valid and reliable?

The limitations of the studies should be put well in evidence. For sure the paper presents an approach that could be followed to perform analogous research, and this is a value.

Suggested improvements: Please list additional analyses, experiments or data that could help strengthening the work in a revision.

See comments below. I think that o further experiments can be done given the effort and the complexity of these studies. Limitations are in the construction of the study and can only put in evidence.

References: Does this manuscript reference previous literature appropriately? If not, what references should be included or excluded?

See comments below.

Clarity and context: Is the abstract clear, accessible? Are abstract, introduction and conclusions appropriate?

The abstract is clear, as well as introduction and conclusions.

Please indicate any particular part of the manuscript, data, or analyses that you feel is outside the scope of your expertise, or that you were unable to assess fully.

Here below are detailed comments. The numbers below refer to the lines of the manuscript.

162

There are official definitions of what a robot is, issued by ISO and by IEEE. It would be better to refer to them rather than to a paper.

References

Institute of Electronics and Electrical Engineers: IEEE standard ontologies for robotics and automation. IEEE Std 1872-2015 pp. 1–60 (2015). DOI 10.1109/IEEESTD.2015.7084073

International Standards Organization: ISO 8373:2012 – Robots and robotic devices — Vocabulary. ISO (2012)

164

It would be better to focus on significant differences: sonars and infrared sensors both measure a simple distance, while a camera or a depth camera or a laser range finder would better give an idea of the range of possibilities in sensing, when compared to the simple single distance sensor such a sonar.

167

No robot with the characteristic b) (which means autonomy at least at a certain degree) are performing surgery. Surgical robots are completely tel-guidated and have no autonomy: all the responsibility is up to the surgeon. As an example of a sophisticated robot you may mention humanoids able to run and jump, or autonomous vehicles.

169

It seems that feelings investigated are not about robots, but about the idea of robot that people may have developed from media, often not from direct experience. Direct experience may change a lot these feelings, as roboticists know from exhibitions of their products to general public. This should be put in evidence.

175

As recently put in evidence, uncanny valley effect comes from bad implementations of robots more than from the concept itself, as proposed by Mori more than 50 years ago.

References

The uncanny valley is wrong - https://www.youtube.com/watch?v=LKJBND_IRdI

Bartneck, C., Kanda, T., Ishiguro, H., & Hagita, N. (2009, September). My robotic doppelgänger-A critical look at the uncanny valley. In RO-MAN 2009-The 18th IEEE international symposium on robot and human interactive communication (pp. 269-276). IEEE.

Cheetham Marcus, Suter Pascal, Jäncke Lutz (2011) The Human Likeness Dimension of the "Uncanny Valley Hypothesis": Behavioral and Functional MRI Findings, *Frontiers in Human Neuroscience*, 5, <https://www.frontiersin.org/articles/10.3389/fnhum.2011.00126>

343 and 348

Links should be mentioned in print, not just as a link.

356

See comment on line 162: this is not true, since the normative organisms have defined them. Of course, anybody may have personal opinions, and the writer is also critical about the official definitions, but they are a reference for all norms related to robots.

357

This goal has some relevance, although it is an analysis of the expectations about robots that may be influenced by personal experiences and attitudes, as seen also in the cited literature, and not an analysis of the experience with robots.

390

The sample is strongly biased on US and UK people, as from table 2. It should be mentioned from the beginning that these are results from a population of these two countries. Not much is said about confidence with technology and robots, but if they are people recruited through platform, their confidence with technology may be quite good. This is also a strong bias.

453

Possibly, it is more common to have experience with robots at home (e.g., cleaners, robotic pets) than at work.

484

Links should be mentioned in print, not just as a link.

526

This apparent contradiction hides an error of overgeneralization: a robot can have many different appearances and abilities. Possibly a robotic pet (robotic pets are the highest number of robots in our houses, increasing by millions every Xmas) might be more accepted than a robotic fighting drone, a Nao more accepted than Boston Dynamics' robots doing parkour.

References

Bonarini, A. (2016) Can my robotic home cleaner be happy? Issues about emotional expression in non-bio-inspired robots. *Adaptive Behavior*, 24(5), 335-349.

Bonarini A., Besio S. (2022), *Robot play for all: Developing Toys and Games for Disability*. Springer Cham, CH.

In general, Table 4 seems a discursive description of the ISO and IEEE definitions, so a confirmation that people may describe robots as the experts did, as then reported at the end of the paper as a feature of this research. Contrasts between "common sense" and "experts" should not be mentioned in the paper since the authors found that they aren't present. It is good that they could support that experts are not living in an ivory tower, and are not too far from laymen.

678

It is relevant to have these questions listed, since they are strongly related to the main findings of Study 3.

724

The way the robot was presented should be reported in the paper: a picture, a description, a functional/operational description, all of them? The selection and the presentation of the robots strongly influence the answers. An explicit reference to supplementary material should be put here. Alexa (one of the "robots" included in the list) is by no way a robot, but an AI application. It does not move.

1223

In the light of the mentioned references (see comments about line 175, above) this is not surprising at all. Moreover, attitudes towards humans themselves may affect opinions about humanoids, as it appears also from the findings reported later in the paper.

1250

What is the reference to (SM, pp.100-109)? Supplementary material may be mentioned explicitly.

1256 and 1284

Links should be mentioned in print, not just as a link.

1452

The authors has to add a couple of domains not mentioned, but present in their experience and possibly other domains are missing, maybe because related to a niche or not enough known by people in the sample, or not associated to robots.

1459

This is not "what robot are", but "what the idea of robot is", quite different from reality, and still it is not clear whether the expectations will be matched. For instance, most of the autonomous robots are expected to survive at least a working day without being attached to a power source, and research about this is struggling since years without any significant results. Many "intelligent" robots needs to be attached to a reliable internet link and are not autonomous.

1499

Given the time passed from the IEEE work and the significant advancement of technology, we may expect that this list will grow in the next years.

1627

Declaring a limitation does not mean that it does not exist, and it should put in evidence from the beginning and prevent to present results as general. A careful phrasing should be considered throughout the paper.

Review by Andrea Bonarini

Author Rebuttal to initial comments:

REVIEWER 1'S COMMENTS

COMMENT 1: This manuscript reported 7 studies with a data-driven approach to develop the positive-negative-competence model of psychological responses to robots. Additionally, individual differences predicting the psychological responses were also explored. I enjoyed reading the manuscript. Although the whole manuscript is quite complicated with 7 studies, the authors described and explained them clearly. I really appreciate that extremely detailed description of the methods and results are provided in the supplement materials. However, there are a number of issues that dampened my enthusiasm.

RESPONSE: We would like to thank you for taking the time to read our manuscript with attention and giving us useful feedback that challenged us to significantly improve it. Your comments showed us that we were not sufficiently clear when explaining some important methodological or analytic decisions and that, in some cases, we did not report analyses or sufficiently detailed data that would have addressed several potential limitations. Below, we answer each of your comments in detail to describe how we tackled these issues.

Before we proceed with answering the comments, we want to inform you that, although the editor initially required us to conduct an additional experiment for this revision, they have eventually clarified with us certain aspects of the manuscript and informed us that we do not need to conduct the additional experiment. In their words, as communicated to us: "We would not require you to carry the initially requested experiment out. You should note in the response to editors' and reviewers' letter that this course of action was agreed with the editors (otherwise, the reviewers may wonder why you did not carry out the experiment we requested)."

Furthermore, to help you navigate the revisions we made, we outline several formatting details. Compared to the original version of the manuscript, the revised version is formatted as required by the journal. In this new version, all revisions we made in response to the comments by you and Reviewer 2 are highlighted in yellow. Likewise, all revisions that we made in the Supplementary Information file are highlighted in yellow. Furthermore, in both the revised manuscript and Supplementary Information, references appear as superscript Arabic numerals. Therefore, whenever we copy-paste a section from the manuscript or Supplementary Information in response to your comment, the references are presented in this format and can be matched to the sources in the corresponding reference lists.

Overall, we are grateful for the effort you have invested in reading our manuscript and for all the useful suggestions you have provided.

COMMENT 2: I applaud the data driven approach. However, I do think that the study could have been benefited with a combination of data driven approach and theoretical support. Media equation and Computers as Social Actions (CASA) is well established framework to study robot. I am surprised that the authors did not cite this.

RESPONSE: Thank you for this insight. After re-reading our manuscript in response to your comment, we realized that we were too vague when justifying the use of data-driven approach, and we did not clearly convey that by using this approach we aimed to benefit previous work in ways that go beyond deductive studies rather than neglecting it. Moreover, we failed to explain why we did not combine data-driven research with frameworks such as the Media equation and Computers as Social Actions, even if we considered this before starting the project. Therefore, to address your comment, we substantially revised the sections where we justify our data-driven approach as follows (pp.4-5):

“A data-driven approach is recommended in the early stages of developing a construct, when theoretical foundations have not yet been proposed, and/or theorizing is conflicting and ambiguous^{77-80,82,83}. Considering this guiding principle, it is important to elaborate why our research lends itself to this approach. As previously indicated, different affective, cognitive, and behavioral responses to robots have not been studied under an all-encompassing construct (i.e., psychological processes), with the aim to investigate their overarching structure. In that sense, the conceptual bases of our research are at an early stage. Furthermore, although various theories and models of human-technology relationship exist, encapsulating the entirety of psychological functioning regarding robots by identifying, organizing, and predicting psychological processes that robots trigger is beyond their scope. For example, the technology acceptance model⁸⁴⁻⁸⁶ examines factors that make people accept technology, whereas the media equation⁸⁷⁻⁸⁹ proposes that people interact with technological devices like they interact with humans. Overall, we are developing an early construct for which there are no clear or coherent theoretical predictions, which makes a data-driven inductive approach optimal.

Data-driven approaches have three main benefits. First, they allow studying novel topics without engaging in premature theorizing that can lead to post-hoc hypothesizing and spawn false positive findings^{77,78,90-93}. Second, because the emphasis is on inferences from data that are not constrained by previous theories and findings, these approaches can diversify knowledge of human psychology and spark novel, unexpected insights^{79,81,94}. Third, they can, in many cases, be of greater benefit to previous research on the topic than deductive approaches directly informed by this research. In psychology and social sciences, failed replications are common, and researchers examining the same research questions and hypotheses, even with identical data, can often obtain different and even opposite findings⁸⁹⁻⁹³. Therefore, if a data-driven study produces a finding that is aligned with previous research and theorizing,

despite using a methodological approach that is not constrained by their assumptions and is solely guided by data, this is a highly compelling case of support for the previous work. It is thus important to emphasize that using a data-driven approach does not imply conducting a research project that disregards previous literature and its conclusions. Quite to the contrary, it is essential to comprehensively evaluate and discuss how the findings are linked to previous work to illuminate how the present research has corroborated and extended this work and moved the field forward—this process is labelled inductive integration⁷⁷.”

Overall, we hope that it now comes across why data-driven research was the most optimal choice for our project, and that by using this approach we wanted to contribute to and expand on previous research in ways that would not have been possible had we based our studies directly on this research. In the Discussion section (**pp.8-10**), we examine the most important insights that the present research has achieved in this regard, and in Supplementary Information (**pp.135-138**) we provide an expanded version of this discussion. Importantly, to further address your comment, we added to Supplementary Information (**pp.137-138**) the following paragraph that more comprehensively discusses the implications of our research for the media equation:

“Interestingly, this insight [*note: “this insight” refers to the conclusion that people may use similar criteria when forming impressions of robots and humans, which we formed by comparing our PNC model with the stereotype content model (SCM) in the preceding paragraph] has important implications for one of the most widely used frameworks regarding human relationship with technology—the media equation²⁰⁷⁻²⁰⁹. According to this framework, people interact with robots, computers, and other devices like they interact with humans and treat them as social actors. However, it remains unknown why exactly this is the case, and several researchers have speculated on potential explanations (see ref.²¹⁰). The alignment between our PNC model and the SCM^{205,206} indicates that people may treat robots as social actors because, on a fundamental cognitive level, they form impressions of robots and humans by considering the same criteria (i.e., by assessing whether robots/humans have any positive and/or negative qualities, and whether they are competent in what they do). In other words, upon encountering a robot, humans may form an impression of this robot that is organized along these criteria and contains various attributes that can be applied to both robots and humans (e.g., being friendly or unfriendly, being more or less thoughtful, being more or less knowledgeable, etc.). If subsequent communication with the robot is guided by these attributes, it will naturally contain elements that apply to both humans and robots, thus creating the impression that people interact with robots like they interact with humans. Investigating this in more depth may lead to interesting insights about cognitive mechanisms that underpin the media equation.”

Finally, we also want to emphasize that studies are increasingly using data-driven “bottom-up” approaches to investigate broad and complex constructs that aim to summarize human psychological responses to some object or theme, as is the case with psychological processes regarding robots. For example, Weisman, Legare, Smith, Dzokoto, Aulino, et al. (2021, p.1358) investigated concepts of mental life among adults and children by “allowing data to give rise to ontological structures, rather than

working ‘from the top down’ by using a theory to guide hypothesis-driven data collection.” Theories and models are infrequently developed with the purpose of describing the totality of human psychological experience regarding a topic (e.g., technology) and often focus on some of its aspects (e.g., why people accept technology, do they interact with it comparable to how they interact with humans). In such cases, imposing theories on phenomena that exceed their scope could prevent the discovery of important insights or lead to conclusions that are constrained by the theory and do not reflect people’s psychological experiences in their full richness.

COMMENT 3: Additionally, the validity of any data driven approach depends significantly on the quality of the data as well as the representativeness of the data. The authors have done many right things to ensure data quality, e.g., seriousness check, etc. However, the representativeness of the data is not well addressed. The authors only reported gender and mean age of the sample characteristics. Educational level and prior direct experience with robot (not imagined) are two important factors related to psychological responses to robots. The online participant recruitment panel used in the current study seem to lack representativeness in this regard. Many of the studies measured employment status and use of robots at work but the data were not reported.

RESPONSE: We are grateful for this comment because you raise an important issue that we did not sufficiently tackle in the original manuscript despite having the data to do so. To our understanding, this issue is as follows: the composition of the participant samples we were using could have biased the main conclusions about psychological responses to robots as reflected in our taxonomy. For example, maybe the taxonomy we developed works only for some demographic groups (e.g., men, or employed individuals, or people who use robots at work, or those who are more educated, etc.), but not for others (e.g., women, or unemployed individuals, or those who do not use robots at work, or those who are not well educated, etc.). If this were the case, then our taxonomy would not be an accurate representation of psychological responses to robots, especially if the data lack representativeness regarding some of these groups.

The most formal statistical way to address this issue is to test for measurement invariance, which can be probed as part of confirmatory factor analytic procedures (Chen, 2007; Putnick & Bornstein, 2016). In relation to our taxonomy, demonstrating measurement invariance would show the following: a) its factor structure is equivalent for some two groups of participants (configural invariance), such as men vs. women, employed vs. unemployed, less vs. more educated, those who do vs. do not use robots at work, etc.; b) the factor loadings are equivalent across these groups (metric invariance); and c) the item intercepts are equivalent across the groups (scalar invariance; see <https://www.psychologicalscience.org/observer/testing-for-measurement-invariance>). In sum, demonstrating measurement invariance would show that our taxonomy, as measured by the Psychological Responses to Robots (PRR) scale (Table 5, pp.39-40), works for the demographic groups

that are being compared, and that the conceptualization of psychological processes regarding robots is therefore equivalent across these groups and applies to each of them.

In that regard, demonstrating measurement invariance goes beyond sample representativeness because representativeness itself does not formally test or show if a construct applies to all demographic groups that constitute the sample or if it is mostly shaped by the predominant demographic group but differs for other groups. Therefore, the presence or absence of a representative sample cannot show or ensure that a developed construct is or is not biased. In contrast, measurement invariance analysis can directly compare different groups and demonstrate whether the construct works for them, thus showing support, or lack of support, for the validity of the construct (Arrindell, Checa, Espejo, et al., 2022; Putnick & Bornstein, 2016).

In the original version of our manuscript, in Study 5, where we confirmed the structure of the taxonomy, we tested measurement invariance only for the following variables: country (UK vs. US) and robot example (for details, see Table 6, **pp.41-42**). However, we did not do this regarding other participant information that we had. Therefore, to address your comment, we now report measurement invariance for gender (female vs. male), age (below median vs. median and above), and employment status (employed vs. unemployed): see Table 6 (**pp.41-42**) and the fourth full paragraph on **p.7** in the revised manuscript. As the number of participants who used robots at work in Study 5 was too small for computing reliable invariance analyses (42 for Sample 1 and 48 for Sample 2), we did not test invariance for this variable in Study 5. Instead, we did it in Study 6, which had a larger sample and more participants who used robots at work (i.e., 235). In that study, we also tested measurement invariance for additional participant characteristics that were assessed only in that study: educational attainment—secondary education or below vs. higher education; income—below median vs. median and above; political orientation—liberal vs. conservative; ethnic identity—white vs. another identity; and relationship status—no partner vs partner.

In the revised manuscript, we state this on **p.7** as follows (see also Table 6 note, **pp.41-42**): “Because we could not analyse measurement invariance for participants who did vs. did not use robots at work in Study 5, since the number of those who did not was insufficient (Table 1), we tested this in Study 6, where the sample sizes were larger. In Study 6, we also computed measurement invariance for additional participant characteristics assessed in that study (educational attainment, income, being liberal vs. conservative, ethnic identity, and relationship status). Measurement invariance was demonstrated in all these cases (Supplementary Table 9).” The table with these additional invariance analyses for Study 6 is included in Supplementary Information (**p.158**), given the space limitation in the main manuscript, and because the core focus of Study 6 was identifying the best predictors of psychological processes regarding robots. We think this is the most optimal way to present the analyses. However, we are happy to follow any of your recommendations regarding reporting if you have other suggestions.

Overall, the measurement invariance analyses show that psychological responses to robots as assessed via our taxonomy are equivalent across many different participant characteristics, from gender and age

to education and use of robots at work, and that the validity of the construct is therefore not compromised by differences in these characteristics. We hope that the extensive tests of invariance indicate the seriousness with which we took the potential issue you have pointed out, given that such extensive tests are rarely conducted in scale/construct development literature (see D'Urso et al., 2022, <https://psyarxiv.com/n3f5u/>). We also want to emphasize that we tested measurement invariance for the variables in question (i.e., country, robot example, gender, age, employment status, use of robots at work, educational attainment, income, being liberal vs. conservative, ethnic identity, and relationship status) because we thought this most directly addressed your comment and made sense conceptually. However, we are also happy to test measurement invariance regarding other variables if you see this as more appropriate (e.g., in Study 6, we measured a range of different variables; see Supplementary Table 10 in Supplementary Information, **pp.159-166**).

In addition to conducting new measurement invariance tests, to address your comment we improved the reporting of demographic characteristics, considering that one of your concerns was that we did not sufficiently report several demographic variables, which made it difficult to understand the composition of our samples. In this regard, we added to Table 1 (**pp.29-30**) two new variables that we measured in all studies: employment status and use of robots at work. Moreover, in Supplementary Information (**p.142**), we included more extensive Supplementary Tables 1 and 2 where we report a comprehensive breakdown of some variables (e.g., age) as well as additional participant characteristics that were measured in some studies: geographical region, educational attainment, income, political orientation, etc. In the revised manuscript, we refer to these additional tables as follows (**p.11**): “More comprehensive breakdowns of participant information and the criteria used for representative sampling are available in Supplementary Tables 1-2.”

Finally, to address your comment, we also report information about sample representativeness that we omitted from the original article, which might have contributed to certain doubts about our sample. Indeed, for our key Studies 4-6 that established the taxonomy and its most important predictors that were further replicated in Study 7, we requested from the recruiter to obtain samples that are representative in terms of age, gender, and geographical region, whereas for Study 1 (Sample 1) we aimed at representativeness in terms of gender. This is now reported in the note to Table 1 (**pp.29-30**), and on **p.11** as follows: “In Studies 4-6, participants were recruited to be reasonably representative of the UK/US populations for age, gender, and geographical region, and in Study 1 (Sample 1) for gender only.” We omitted mentioning sample representativeness in the original article to save space, since it would have required reporting additional demographic information such as geographical region and age breakdown. However, considering your comment, we agree that this demographic information is highly important, and it is now reported in Supplementary Information (**p.142**) as indicated above. Please note that we have submitted the evidence from the recruiter that they were targeting representative samples as described in this paragraph directly to the editor.

Although the arguments and reporting of data provided in response to this comment are relevant to each study in our article, they more directly apply to the quantitative studies (4-7). For that reason, it is

important to additionally clarify why the quality of the data in our qualitative studies (1-3) does not compromise the main conclusions of our article, in case some doubts about these studies are still present. The primary aim of those studies as a whole was to identify a comprehensive range of psychological processes regarding robots so these processes could be formally tested in Studies 4-7. Therefore, the main flaw that could have occurred in Studies 1-3 is a failure to identify one or more important psychological processes. We provide several reasons below to suggest that this is unlikely to have happened.

First, these studies were generally larger and more diverse than would be expected from a qualitative study (e.g., Hennink & Kaiser, 2022; Vasileiou, Barnett, Thorpe, & Young, 2018). For example, it has been demonstrated that sample sizes larger than 30-50 (and sometimes even fewer) participants tend to reach the point of data saturation, which implies that adding new participants beyond this number produces very little new information (e.g., Faulkner & Trotter, 2017; Fugard & Potts, 2015; Guest, Namey, & Chen, 2020; Hennink & Kaiser, 2022; Mayring, 2019; van Rijnsoever, 2017). Studies 1-2 had at least double this sample size, and Study 3 had 334 participants included in analyses. For this study, it took us over a year to analyze the data, and examining the coding files that total 1084 pages and include 10332 valid key points that were analyzed (see folder “Study 3 -> Iterative Categorization”: https://osf.io/2ntdy/?view_only=2cacc7b1cf2141cf8c343f3ee28dab1d) reveals the comprehensiveness of psychological processes the study has generated. Based on both reading these analyses and the aforementioned guidelines on sample size (e.g., Hennink & Kaiser, 2022; Mayring, 2019; van Rijnsoever, 2017), it would be difficult to argue that testing any new participants, regardless of their demographic characteristics, could have generated additional important psychological processes that are not covered.

Second, if the data from Studies 1-3 were somehow flawed or biased by certain sample characteristics in a way that led to erroneous conclusions about psychological processes regarding robots, the taxonomy that was subsequently developed would have been valid for only some demographic characteristics but not others, given that it was grounded in insights from Studies 1-3. However, as measurement invariance analyses showed (see the fourth full paragraph on **p.7**), the taxonomy was valid across many demographic characteristics, from age and gender to educational attainment.

Finally, examining previous literature gives no indication that the conclusions from Studies 1-3 may have been flawed. For example, the robot definition developed in Study 1 is aligned with the official Institute of Electronics and Electrical Engineers (IEEE) definition (see the first highlighted paragraph on **p.9** in the revised manuscript), as also acknowledged by Reviewer 2 (Comment 19). Furthermore, Study 2, which relied on this definition to develop a comprehensive list of domains in which robots can be encountered, combined both qualitative research and literature examination to ensure we did not miss any domains (see paragraph 3 on **p.6**). In that regard, we were not able to identify any additional important domains beyond what our list presented in Table 3 (**p.32**) includes. Finally, Study 3, which relied on the domains developed in Study 2 to generate a comprehensive list of psychological processes regarding robots, captured all such processes that were identified by the previous literature but also went beyond and

identified a wider range of psychological processes (see the third full paragraph on **p.9** in the manuscript, as well as Supplementary Information, **p.136**).

COMMENT 4: Sample 2 in study 1 were asked to sort 277 different robot characteristics generated by sample. This is a very challenging task to sort that many items. I have some reservations on the quality of the data. The average age of the participants in this sample is 36, compared to about 45 for the other studies (except study 2). Is that an indicator that only relatively younger participants are able to complete such a cognitively challenging task in the study?

RESPONSE: Thank you for flagging this potential issue. We agree that it might have been problematic if cognitive demands associated with the task made this task suitable only for some participants but prevented others from solving it. We have therefore examined this issue in depth to address your concern, and here we provide evidence that indicates this was not the case. To ensure that participants were paying attention while sorting robot characteristics, we used four instructed-response items (e.g., “Please insert this item into Group 20”; Meade & Craig, 2012). If younger participants found the task easier, it would have been expected for them to make fewer attention-related errors while solving the task, which means that fewer younger and more relatively older participants would have been excluded from analyses due to failing to accurately sort the instructed response items. However, this was not the case. Out of five participants excluded from analyses, three belonged to the age group “25-34,” one to the age group “34-44,” one to the age group “45-54,” and none of the excluded participants were older than 54. In the revised manuscript, this information can be obtained from Supplementary Table 1 (Supplementary Information, **p.142**) which provides a breakdown of different age categories for all participants who completed the study and for those who were included in analyses.

The reason why in Study 1 (Sample 2), Study 2, and Study 3 the average age is somewhat lower than in other samples is that participants were recruited using MTurk (see Table 1 on **pp.29-30**). Whereas MTurk participants are representative of the general US population across most psychological constructs (McCredie & Morey, 2019), they are on average somewhat younger than the general population (Difallah, Filatova, & Ipeirotis, 2018). For example, several large studies have found that the average age of MTurk participants is 35.5 years (McCredie & Morey, 2019; Michel, O’Neill, Hartman, & Lorys, 2017), which is comparable to the average age of our participants in the studies that used MTurk (Study 1 Sample 2 = 36.510; Study 2 = 36.257; Study 3 = 40.693).

In sum, based on the evidence we examined, there are no indications that age compromised the quality of data for Study 1 (Sample 2). Importantly, as already indicated, out of 100 participants who took part in the study, only 5 were excluded from analyses due to not passing various attentiveness checks, which indicates that participants generally did not find it problematic to sort the robot characteristics attentively. Moreover, if one examines the clusters of robot characteristics that were generated from participants’ responses (a comprehensive presentation of the clusters is available in Supplementary

Information, **pp.84-87** and **pp.143-145**), it is plausible to conclude that these clusters generally make sense and that robot characteristics are grouped based on thematic similarity rather than randomly. Importantly, the robot definition developed from the clusters is aligned with the official IEEE definition, as indicated in the revised manuscript (see the first highlighted paragraph on **p.9**).

Finally, to address your comment in depth, we want to more broadly demonstrate that age is highly unlikely to have compromised the validity of the present findings. The most direct statistical evidence that supports this claim is that measurement invariance analyses showed that relatively younger vs. older participants (Table 6, **pp.41-42**) produced equivalent factor structure, loadings, and item intercepts regarding psychological processes in response to robots, which means that they experience these processes in an equivalent way. Second, as already indicated in response to your previous comments, our key studies 4-6 were broadly representative of different age groups, and all our quantitative studies (4-7) generally had large samples and included participants from all age groups. Finally, in our qualitative studies 1-3 that were used to ultimately produce a comprehensive list of psychological responses to robots used for the taxonomy (Table 4, **pp.33-38**), sample sizes were generally larger than would be expected in this type of research (for a detailed argument in this respect, please see the final 3 paragraphs in our response to your Comment 3). We appreciate your comment and hope our response assuages your concern.

COMMENT 5: Although the authors stated that it is purposeful to have overlapping domains in study 2, I am still not fully convinced why it is better than having more mutually exclusive domains. For instance, industry and manufacturing have a lot of overlapping. Why not merge them?

RESPONSE: Thank you for this comment. We acknowledge your perspective in this regard, and we want to point out that before we made the final decision about the domains, we comprehensively examined several different ways of organizing them, including the one you bring up (i.e., merging some domains that may be overlapping). Considering that qualitative approaches are inevitably more flexible and informed by the researcher's perspective or approach to interpretation (Willig, 2019), we made the final decision by considering one of the main objectives of Study 2. This objective was to establish a representative content space of all robots, given that researching a construct (e.g., psychological responses regarding robots) by relying on a limited set of stimuli (e.g., specific types of robots) that are not representative of the entire population of these stimuli (e.g., all robots) can bias the findings and decrease their replicability (Westfall, Kenny, & Judd, 2014; see also Westfall, Judd, & Kenny, 2015). In that regard, we concluded it is more optimal to lean toward having too many domains (rather than too few), because the more domains there are, the more diverse the sample of robots is, and the less likely it becomes that important types of robots will be omitted. To address your comment in the revised manuscript, we have included this explanation in the note to Table 3 that presents all domains so the readers can understand why potential overlap may exist (**p.32**): "Our aim was to develop domains that are narrow rather than broad, which means that some overlap between them may be present. This

approach was aligned with our objective to establish a representative content space of all robots to avoid using a biased stimulus sample^{108,109} when developing the taxonomy of psychological responses to robots. For that reason, it was more optimal to lean toward having too many rather than too few domains to reduce the chance of failing to cover the content space of all robots in detail and omitting important types of robots.”

COMMENT 6: Study 3 had a large sample size (334 of the 350 participants remained). The authors mentioned that “Irrelevant points, including those that were meaningless or that merely repeated the name of the domain in question, were labelled in red and excluded from further analyses.”. I wonder how many responses were generated by the 334 participants in total and how many of these responses are such irrelevant points. How many of the 334 participants generated usable data?

RESPONSE: Thank you for this suggestion. We agree that we should have more clearly reported how many participants in Study 3 generated usable data, given the importance this study has for the remaining studies. To address your comment, we have included the following text in the revised manuscript (**p.13**): “Out of 334 participants who were included in analyses (Table 1), only four produced merely meaningless responses that could not be analyzed, and the remaining 330 participants generated 10332 valid key points (approximately 31 per participant) that were analyzed.”

COMMENT 7: Study 4 participants were asked to answer 149 questions. Are those questions ordered randomly?

RESPONSE: Thank you for pointing this out. We confirm that the 149 questions assessing psychological processes regarding robots in Study 4 (Samples 1 and 2) were presented in a randomized order to each participant. This also applies to the order of presentation of items measuring psychological processes regarding robots across all studies. The information was missing from the original version of the manuscript, and we report it in the revised manuscript by including the phrase “presented in a randomized order” when describing how the items for Studies 4-7 were presented (**pp.13-15**).

COMMENT 8: Also, for study 4, with such a large number of items for EFA, why didn’t the authors choose to separate cognitive, affective, and behavioral responses rather than combine them together for the EFA? Would separately analyzing the items of cognitive, affective, and behavioral responses reduce the over-factoring issue? I am also not fully convinced why the 3-factor solution should be chosen. This is key component of the manuscript and deserves more discussion.

RESPONSE: We are grateful for these suggestions. When we wrote the original version of the manuscript, we aimed to be succinct when describing the rationale behind the selected 3-factor solution in Study 4. However, as a consequence, we failed to adequately explain why this factor solution is the most optimal one. In response to this comment, we therefore comprehensively explain that our choice of the factor solution was based on a) statistical criteria; b) semantic criteria (i.e., whether the factors are easy to interpret conceptually); and c) precedents in previous taxonomic research. Moreover, we address your suggestion about separately analyzing the items of cognitive, affective, and behavioral responses. Finally, we conclude that the three-factor solution is the only plausible one that does not have any serious limitations and can be defended from multiple perspectives. Importantly, we outline the changes that we made in the manuscript to clarify our rationale behind selecting the factor structure.

In terms of statistical criteria, the first step in selecting the best factor structure was to show that our data for Samples 1 and 2 are suitable for exploratory factor analyses (EFAs) by computing the Kaiser-Meyer-Olkin (KMO) measure of sampling adequacy and Bartlett's test of sphericity (Beavers et al., 2013). In this regard, to further proceed with EFAs, the KMO values had to be higher than 0.9, and Bartlett's test had to be statistically significant at $p < .05$ (Beavers et al., 2013).

The second step was to use the following analyses/criteria to determine the preliminary number of factors to examine in EFAs: parallel analysis (PA; Dinno, 2009; Horn, 1965), Very Simple Structure (Revelle & Rocklin, 1979), Velicer Map (Zwick & Velicer, 1986), Optimal Coordinates (Raïche, Walls, Magis, Riopel, & Blais, 2013), Acceleration Factor (Raïche et al., 2013), Kaiser Rule (Luo, Arizmendi, & Gates, 2019), and visual inspection of scree plots (Cattell, 1966). This is advisable because no single statistical criterion is best for selecting the most optimal factor solution (Gorsuch, 2003), and consulting many different criteria allows understanding the range within which this solution potentially lies. In practice, especially when it comes to analyzing complex taxonomies, some criteria (e.g., parallel analysis) can greatly overestimate the optimal number of factors, whereas some criteria (e.g., Very Simple Structure) may underestimate it, and the factor solutions that researchers eventually select lie somewhere in between (e.g., Parrigon, Woo, Tay, & Wang, 2017; Rauthmann, Gallardo-Pujol, Guillaume, Todd et al., 2014).

The goal of the third step was to more stringently evaluate a range of different factor solutions to identify the one that is statistically most optimal. Statistical techniques used for this purpose were maximum likelihood (ML) EFAs (Goretzko, Pham, & Bühner, 2021) with an oblique rotation—promax with Kaiser (1958) normalization. This rotation was selected because it tends to produce a clean factor structure and is thus compatible with taxonomic research that involves many items and can suffer from interpretability issues (Parrigon et al., 2017; Schmitt & Sass, 2011). For an optimal factor solution, we expected all factors to be valid (i.e., clearly defined and have little noise), as reflected in a sufficient number of items with high loadings and low cross-loadings (Beavers et al., 2013; Costello & Osborne, 2005). More specifically, we expected all factors to meet the following benchmarks: 1) they should have at least 3 items with standardized loadings of .5 or higher; and 2) these items should not have cross-loadings of 0.32 or higher (Beavers et al., 2013; Costello & Osborne, 2005; Schmitt, Sass, Chappelle, & Thompson, 2018; Tabachnick & Fidell, 2001). If a factor solution does not meet these

benchmarks, it likely has too many factors, and fewer factors are therefore typically more appropriate (e.g., Beavers et al., 2013; Costello & Osborne, 2005). In line with this rationale, our plan was to first evaluate, against the benchmarks, the factor solution estimated by the analysis from step 2 that indicated retaining the largest number of factors (in taxonomic research, this is typically parallel analysis; Parrigon et al., 2017). If this solution was not satisfactory, we aimed to decrease the number of factors by one and evaluate this new factor solution. We planned to continue this procedure until we identified a factor solution that met the benchmarks. This solution was deemed the statistically most optimal one.

In terms of semantic criteria, it was important that the statistically most optimal solution also makes sense on a conceptual level by having factors that are easy to interpret and are coherent. Indeed, the number of factors to determine is not only a statistical but also a subjective decision based on factor interpretability, and it is recommended that researchers avoid selecting factor structures that are flawed semantically, even if they are statistically sound (Gorsuch, 2003).

Finally, the statistical and semantic criteria that we used to identify the best factor solution were also guided by precedents in previous taxonomic research. Although we could not identify previous research on robots that is methodologically similar to the present research when it comes to determining the best factor solution, we did identify conceptually comparable studies from a different domain. These studies focused on developing two different taxonomies of psychological situation characteristics: CAPTION (Parrigon, Woo, Tay, & Wang, 2017) and DIAMONDS (Rauthmann, Gallardo-Pujol, Guillaume, Todd et al., 2014). In each case, participants were asked to rate a range of different situations on many characteristics (e.g., malicious, sentimental, enjoyable, humorous), similar to how our participants were asked to rate a range of different robots in relation to various psychological processes (e.g., empathy, creativity, anxiety). Importantly, the procedure to select the best factor structure (see Study 2 for CAPTION and Study 1 for DIAMONDS) was similar to our Study 4. For example, the researchers first computed several analyses to determine the optimal number of factors (e.g., parallel analysis, optimal coordinates), which resulted in a wide range of factors being recommended, comparable to our results: 5-22 factors for CAPTION, and 3-17 for DIAMONDS. However, when EFAs with promax rotation were conducted to explore these factor structures, factors with only weak loadings and many high cross-loadings were identified. The researchers therefore started analyzing other structures with fewer factors until they identified the ones that did not have any weak factors and that also made sense semantically: 7 factors were retained for CAPTION and 8 for DIAMONDS.

Overall, this explanation clarifies our reasoning behind selecting the 3-factor solution in Study 4 and shows that we are aligned with comparable high-quality taxonomic research that we could identify. To address your comment, we have included an abbreviated version of this explanation in the revised manuscript (see the “Analytic Approach” section for Study 4 on **p.13**), whereas the comprehensive explanation is available in Supplementary Information (see the “Analytic Approach” section on **pp.12-14**). We also now report the results in a way that is aligned with the explanation for easier interpretation (see the yellow highlighted text at the bottom of **p.6** in the manuscript, and the yellow highlighted text on **p.80** in Supplementary Information under Study 4).

In addition to comprehensively explaining our logic concerning the factor structure selection, to address your comment, it is also important to explain why, as you state, “for study 4, with such a large number of items for EFA, we didn’t choose to separate cognitive, affective, and behavioral responses rather than combine them together for the EFA”. Before going into a detailed explanation, it is important to emphasize that, although 149 items may seem like a large number, this is not unusual for taxonomic research, and there are studies that used a considerably larger number of items. For example, in CAPTION (Parrigon, Woo, Tay, & Wang, 2017) that we previously introduced, the number of items was 851.

The main reason why we did not perform separate EFAs for affect, cognition, and behavior is that these categories do not constitute a formal classification of psychological processes based on some theoretical or empirical differences between these processes. Rather, this is a “rule-of-thumb” classification that researchers frequently use to summarize various psychological processes and investigate them in humans in an all-encompassing way (e.g., Cacioppo & Decety, 2009; Chen, Lam, Hui, Ng et al., 2016), considering that a more formal, widely accepted classification or taxonomy does not exist. In our research, we used it for two main reasons: a) it allowed us to elicit a wide range of psychological processes regarding robots to understand human psychological functioning on this topic in its full richness; and b) it helped us to identify and organize previous literature on psychological responses to robots, which is rarely studied as a uniform topic under the common umbrella of psychological processes and may contain studies that belong to various more specific topics and themes not linked to each other. Although we attempted to convey this rationale in the original manuscript, it is our fault that we did not express clearly enough that the division of psychological processes into affective, cognitive, and behavioral is not a formal classification, which might have created the impression that it could have been used to inform EFAs. To address your comment, we now clearly explain this on **p.3** as follows:

“In this case, we use the term “psychological processes” concerning people’s affective (i.e., feelings toward robots), cognitive (i.e., thoughts about robots), and behavioral responses (i.e., actions toward them). This “rule-of-thumb” classification is often used to summarize various psychological processes and investigate them in an all-encompassing way^{13–15}, since an official taxonomy does not exist. We adopt it because it is useful as a guiding principle when eliciting diverse psychological processes involving robots, and when identifying and organizing previous literature on this topic, considering that psychological functioning in relation to robots is typically not studied as a uniform construct and comprises studies from numerous areas.”

Importantly, our approach in this respect has precedent in previous literature on which we relied to inform our research design. Chen, Lam, Hui, Ng et al. (2016) studied psychological processes in response to globalization. Comparable to our research, item development involved asking participants “to list feelings, thoughts, and behaviors related to contact with other cultures under the influence of globalization” (Chen et al., 2016, p. 308). Moreover, their analytic approach did not involve splitting items based on these categories. Instead, all items were included in analyses together. Similar to our

research, their factor solution did not correspond to feelings, thoughts, and behaviors but to themes more directly linked to the topic of investigation—multicultural acquisition and ethnic protection. In this regard, it is important to emphasize that even when formal theories and classifications are studied psychometrically for the first time, it is advised to use exploratory factor analytic approaches on all items to estimate whether the structure proposed by the theory will naturally emerge or a different structure is advisable (e.g., Caviola, Everett, & Faber, 2019; Haig, 2005; Reio Jr & Shuck, 2015; Wood & Boyce, 2017).

Overall, to sum up, when deciding on the best factor structure in Study 4, it was important to us that the solution could be defended from multiple perspectives: a) that it does not have serious statistical limitations; b) that it makes sense semantically; and c) that it is aligned with the relevant literature and follows best practices. In that regard, we performed EFAs on all items without separating them, in line with previous statistical advice and research conventions, and we identified that all factor solutions that exceeded 3 factors could be criticized on statistical grounds because of having too many noisy factors that do not have an optimal number of items required to form a factor. Importantly, in addition to Study 4, which established the 3-factor solution, Study 5 convincingly confirmed it using a series of ESEM models (see paragraphs 3-5 on **p.7**).

COMMENT 9: Study 4 and 5 have two samples from different online participants panels. What is the reason not combining these two samples to conduct analysis?

RESPONSE: We appreciate this suggestion. Based on our investigation of other articles, it seems there is no clear norm on what to do in this case. It appears more articles tend to report separate samples, but there are also some articles that combine them. The main reason why we decided not to combine the samples is that they come from two separate countries (UK and US). Whereas these are both developed countries, there are differences between them in population density, geography, quality of life, economy, and other factors that may have important implications for psychological functioning (for a summary of differences, see <https://www.worlddata.info/country-comparison.php?country1=GBR&country2=USA>). For that reason, we did not find it appropriate to combine them, and we found it more compelling to show that the results are consistent across these two samples regardless of the differences we mention above. From a statistical and methodological perspective, we also think it is more rigorous to treat the samples separately because, when we started the research, we did not assume it is a given they must yield similar findings. Rather, we put the burden of proof on the data and accumulated evidence regarding the similarity across different studies. We hope this response clarifies why we did not combine the samples.

COMMENT 10: The effect sizes of the individual differences are quite small. This deserves some discussion.

RESPONSE: Thank you for this suggestion. To answer your comment, we inserted the following text in the Discussion section when addressing limitations (**p.10**): “Fourth, effect sizes for the key individual difference predictors from Study 7 ranged from small to medium (Table 7). This absence of large effects indicates that personality may shape psychological responses to robots through a confluence of various individual differences, including the ones we uncovered, rather than via one or a few dominant traits.” Moreover, in Supplementary Information (**p.140**), we included an expanded discussion of this insight, which explains that these effect sizes are aligned with the effect sizes typically obtained in personality psychology:

“Fifth, effect sizes for some of the key individual difference predictors we established were small (Supplementary Table 15). Considering that these predictors were initially identified amongst numerous individual differences using a meticulous procedure that relied on various machine learning models (for the description of the analytic approach, see the “Supplementary Methods” section for Study 6), this finding indicates that effect sizes for the relationships between individual differences and psychological processes regarding robots are small to medium at best. Therefore, no single individual difference may be dominant in shaping these processes, and various individual differences, including the most predictive ones we detected, may operate together in shaping them. Nevertheless, it is important to emphasize that recent research has found that traditional effect size classifications (in case of Cohen’s f^2 , .02 = small, .15 = medium, and .35 = large²²⁵) are inflated, considering that average effect sizes in personality psychology range from small to medium²²⁶, and that it would be more appropriate to refer to traditionally medium effects as large²²⁷. In that respect, our effect sizes are aligned with a large body of personality psychology and comparable to the predictive power of individual differences in other contexts.”

Thank you once again for all the comments and advice you have provided us with. We have found them invaluable in improving the manuscript.

REVIEWER 2’S COMMENTS

COMMENT 1: The authors present a huge effort in identifying a number of concepts concerning how people perceive robots. This is interesting and unprecedented.

RESPONSE: We would like to thank you for reading our manuscript thoroughly and giving us insightful comments that helped us improve it. Our approach to the topic of how humans perceive robots was shaped by our psychology background, and we found your insights from the perspective that combines

computer engineering with several other areas of expertise extremely valuable. We carefully considered each of your comments, and below we explain the changes we made to address them.

Before we proceed with answering the comments, we want to inform you that, although the editor initially required us to conduct an additional experiment for this revision, they have eventually clarified with us certain aspects of the manuscript and informed us that we do not need to conduct the additional experiment. In their words, as communicated to us: “We would not require you to carry the initially requested experiment out. You should note in the response to editors’ and reviewers’ letter that this course of action was agreed with the editors (otherwise, the reviewers may wonder why you did not carry out the experiment we requested).”

Furthermore, to help you navigate the revisions we made, we outline several formatting details. Compared to the original version of the manuscript, the revised version is formatted as required by the journal. In this new version, all revisions we made in response to the comments by you and Reviewer 1 are highlighted in yellow. Likewise, all revisions that we made in the Supplementary Information file are highlighted in yellow. Furthermore, in both the revised manuscript and Supplementary Information, references appear as superscript Arabic numerals. Therefore, whenever we copy-paste a section from the manuscript or Supplementary Information in response to your comment, the references are presented in this format and can be matched to the sources in the corresponding reference lists.

Overall, we appreciate the time and effort you dedicated to reading our manuscript and the valuable insights you shared.

COMMENT 2: There is a general issue about this work: what s investigated is not the attitude of people towards robots, but their attitude towards their concept of robot. It is not clear how much direct experience may have affected this, or media, or memes circulating in the specific parts of the society (not well identified except from country of origin) where the samples have been drawn.

RESPONSE: Thank you for this comment. To address it, we first explain why the stimuli toward which we measured people’s psychological reactions were images and descriptions of robots rather than actual physical robots with whom participants could directly interact, and we then outline the changes we made in response to the comment.

Considering that our goal was to develop a taxonomy that summarizes a comprehensive range of people’s psychological processes regarding robots as an overarching category, our key objective was to use as stimuli a wide range of robots spanning all domains of human functioning, from healthcare to industry. This was necessary to avoid one of the most serious methodological limitations in psychology that can occur when researchers want to produce insights that apply to a stimulus category: focusing on only a few examples (i.e., individual robot types) of a stimulus category (i.e., robots) prevents the

generalizability of findings beyond these few examples and can also lead to various replication failures (see Westfall, Judd, & Kenny, 2015). Therefore, because many robots that we planned to use as stimuli are not accessible for in-person research due to their size, limited production, cost, or potential use as weapons (e.g., industrial robots, military robots, etc.), the only plausible option we had was to focus on visualizations and descriptions of robots as stimuli.

This type of approach, in which visualizations or descriptions rather than actual objects are used as stimuli, is common in our discipline. For example, in many studies from cognitive and consumer psychology that use complex designs, participants are presented with computerized tasks that contain visual representations of various consumer products, objects, or monetary options rather than their actual physical versions (e.g., Amasino, Sullivan, Kranton, & Huettel, 2019; Rinck & Becker, 2007; Sullivan & Huettel, 2021; Sullivan, Hutcherson, Harris, & Rangel, 2015; Thomas, Molter, Krajbich, Heekeren, & Mohr, 2019; Tucker & Ellis, 1998). Also, such research designs, or designs where stimuli are merely imagined or described, have been applied in various studies on robots that focus on robots as a general category (e.g., Jackson, Castelo, & Gray, 2020; Mathur & Reichling, 2016).

In the light of these observations, to address your comment and several related comments you provide below, our approach was to more clearly express from the start that we are not using actual robots as stimuli, and also more comprehensively explain our rationale.

First, in the abstract, we specified the type of stimuli we use when introducing the topic of our research (**p.2**): “To address this, we created a taxonomy of affective, cognitive, and behavioral processes in response to a representative set of stimuli depicting robots from all domains of human activity (e.g., education, hospitality, industry), and examined its individual difference predictors.”

Second, when overviewing our research in the Introduction section, we included the following paragraph (**p.5**): “Across the three phases, we used robots that were either imagined (Phase One) or presented digitally on the screen and described (Phases Two and Three) as stimuli to avoid nongeneralizable findings that apply to few robots^{108,109}. Indeed, this approach allowed us to sample robots across all areas of human activity, which would not have been possible had we focused on physical robots, many of which are inaccessible for in-person research due to their size, cost, limited production, or potential use as weapons (e.g., industrial robots, military robots). Nonetheless, this approach has ecological validity because people’s views on robots are often based on indirect interaction via social media and other websites rather than direct interaction.”

Moreover, when examining limitations in the Discussion section, we added the following text (**p.10**): “Finally, to ensure our taxonomy is representative across diverse robots^{108,109}, our stimuli were not physical robots but their depictions. Such stimuli are widely used⁶ and hold ecological validity, considering that people often interact with robots indirectly, through social media or various websites, and many psychological processes may therefore be shaped via this route. Nonetheless, previous research showed that direct interaction with robots impacts people’s experiences^{27,171,172}. Investigating

whether this would change the structure of our taxonomy may be possible in the future when and if diverse robots become more accessible.”

We also included a more comprehensive version of this text in Supplementary Information (**pp.140-141**):

“Finally, as stimuli, we used robots that were either imagined (Phase One) or presented digitally on the screen and described (Phases Two and Three), rather than actual physical robots. This approach ensured our findings generalize to robots across all domains of human activity^{182,183} (Supplementary Table 5) since many of these robots are inaccessible for in-person research due to their size, limited production, cost, or potential use as weapons (e.g., industrial robots, military robots). There are benefits to this approach beyond stimulus generalizability. First, it does not restrict research to a specific location and enables testing many participants, which was crucial in our studies as developing a taxonomy requires large samples^{94,139}. Second, it holds ecological validity in relation to how people typically encounter robots. For instance, among the 28 domains of human activity examined in this research (Supplementary Table 5), individuals may have direct exposure to robots in only a few of these domains (e.g., household, education, public services). Even within these domains, people may own only one or a few robots and directly interact with them (e.g., robotic pets, social robots, robotic vacuum cleaners). Conversely, it is plausible that people encounter most other robots and form opinions about them via social media or news websites in the form of visuals and descriptions. This reflects the ecological validity of the stimuli we used. However, various studies have shown that direct contact with robots can influence people’s psychological responses^{13,228,229}. In this respect, direct contact with the robots from our stimuli could potentially change the structure of our taxonomy (e.g., resulting in a model that has fewer or more factors than the PNC and includes different psychological processes). Alternatively, direct contact may not alter the taxonomy’s structure but simply change how robots are evaluated on the PNC dimensions (e.g., as more or less positive, negative, or competent). These questions could be resolved by measuring psychological processes regarding all physical counterparts of the robots from our stimuli (Supplementary Figures 1-58) and comparing them to the psychological processes uncovered in our research. Whereas this is currently not possible because of the practical reasons we described (e.g., accessibility of the robots), it may change in the future as various robots, including the more specialized ones, become more widely available.”

Finally, we want to explain why, in addition to the revisions we made, we did not change the terminology we use in the manuscript from “psychological processes in response to robots” into “psychological processes in response to the concept (or idea) of robots.” Although we initially considered making this change, we decided against it because we concluded that factually describing what type of stimuli we use in our research, as indicated above, is the most straightforward way to address your comment while making the article approachable to different types of potential target audiences. In that regard, we assumed that referring to “the concept of robots” may be potentially confusing to a portion of our target audience, such as the researchers who use similar research paradigms and who may therefore be likely to draw on our research and implement insights from it.

For example, when these researchers employ visuals or descriptions of certain objects as stimuli (e.g., foods, robots), they use expressions such as “deciding between an apple and an orange” (Thomas, Molter, Krajbich, Heekeren, & Mohr, 2019, p.625) or “rising robot workforce” (Jackson, Castelo, & Gray, 2020, p.969) or “robot partners” (Mathur & Reichling, 2016, p.22) rather than “deciding between the concept of an apple and an orange” or “the concept of rising robot workforce” or “the concept of robot partners.” Therefore, by encountering expressions such as “the concept of robots,” they may intuitively assume we are studying something that is conceptually different from their research and potentially neglect or overlook our article.

Moreover, we may draw criticism from scholars who study or in some way use mental representations—a core construct that permeates many theories and ideas from cognitive psychology, social psychology, and neuroscience and is therefore explicitly or implicitly embedded in the work of numerous researchers from these fields (Smith & Queller, 2002; Smortchkova, Dołrega, & Schlicht, 2020). They may be critical of us using the term “the concept of robots” in reference to images or descriptions rather than actual physical robots. From their perspective, it could be argued that many psychological processes regarding robots are mental representations that constitute the concept of robots—these representations can be formed through various experiences with robots (including images and physical robots) and can also go beyond these experiences (Smortchkova et al., 2020). In other words, the term “the concept (or idea) of robots” goes way beyond images of robots and can be formed through experiences with physical robots, their descriptions and images, and any other information about them. In that regard, regardless of whether we employed images or physical robots as stimuli in our research, it would have been impossible to disentangle to what extent direct experiences, media, memes, and other sources of information shaped the formation of the psychological processes we measured because we would need to know all possible sources of information people were exposed to regarding robots throughout their lives. However, knowing this was not necessary to accomplish our research goals, since we were not studying the origins of the psychological processes we captured (given the difficulty of investigating this topic scientifically), but how these processes are organized and what personality factors predict them.

Overall, we think that describing and discussing what type of stimuli we use to study robots is the most optimal way to address your comment because this approach is factual while avoiding terminology that has potential to confuse or lead to scholarly disagreement, such as “the concept of robots”. However, we are fully open to your suggestions, and we are willing to reconsider our approach and implement any terminology you find suitable.

COMMENT 3: Methodological novelty

The methodological apparatus is impressive and may be a guidance for further research.

Applied/Societal-/Policy-related Advance

The findings, although limited, may support decisions in both societal and industrial policies.

Evidence-based advance

The studies are impressive and support the findings, which are not much different from what was expected.

Data & methodology.

All data are available and the methodological processes deeply described

Preregistration

In some studies preregistration was applied and the authors followed it.

Appropriate use of statistics and treatment of uncertainties

The methodological aspects are correct.

Custom code: If the work includes custom code, does the code run as intended? If you are unable to access the code, please contact us.

Running the code was not needed for this paper.

RESPONSE: We would like to thank you for this comment and for recognizing the contributions and implications of our research. Considering that the comment contains observations about different aspects of our research and does not recommend any changes, we did not undertake any revisions in response to it. We thank you again for your encouraging words.

COMMENT 4: Conclusions: Do you find that the conclusions and data interpretation are robust, valid and reliable?

The limitations of the studies should be put well in evidence. For sure the paper presents an approach that could be followed to perform analogous research, and this is a value.

RESPONSE: We agree with your suggestion. Based on our understanding, to put the limitations of the studies well in evidence means to a) emphasize the limitations from the start (e.g., in the abstract and/or introduction) rather than only at the end in the Discussion section, and b) include in the Discussion section those limitations that we failed to mention in the previous manuscript. We confirm that we did this for the potential limitations you mention across different comments.

First, in relation to your Comments 2 and 10, which posit that our research examines people's attitude toward the concept of robots rather than the attitude toward robots because our stimuli are images and descriptions, we made several changes that are comprehensively described in response to Comment 2, so we will not list them here to avoid repetition.

Second, in response to your Comments 15 and 28 that refer to using UK and US samples as a limitation, we acknowledged that the present research is based on these samples already in the abstract (p.2): “Across seven studies that tested 9274 UK and US participants recruited via online panels...” Furthermore, compared to the original manuscript, we more comprehensively discussed this limitation in the Discussion section of the revised manuscript (p.10): “There are several limitations to this research. First, our participants were from Western countries (US and UK). Since our research proposed and investigated a construct (i.e., psychological processes regarding robots) from scratch, our priority was to establish its foundations. Combining the investigation of cultural differences with this agenda using equally meticulous methods would have exceeded the scope of a single article. Nevertheless, as measurement invariance analyses showed that the PNC model applies to individuals regardless of their income, age, education, use of robots at work, political orientation, ethnic identity, and relationship status, it is plausible that the model would generalize to countries different from the UK or US on these population characteristics. Conducting an in-depth examination of this question will be a crucial step as the research topic progresses.”

Finally, in response to your Comment 15 about our participants being recruited through online panels and therefore having potentially high confidence with technology, we made several changes. First, we stated already in the abstract that we recruited participants via online panels (p.2): “...9274 UK and US participants recruited via online panels...” Second, we discussed and justified this potential limitation in the Discussion section (p.10): “Third, we recruited online participants who are inherently more confident with technology. Whereas this might have influenced the findings, alternative modes of recruitment (e.g., lab) would have yielded smaller and less representative samples^{166–170}. Furthermore, because measurement invariance analysis showed that our taxonomy was equivalent for people who do (vs. do not) use robots at work and all machine learning models we fit controlled for previous frequency of interaction with robots (Supplementary Tables 9-11), it is less plausible that technological proficiency biased the findings.”

More detailed explanations are available in our specific responses below to the above comments in question.

COMMENT 5: Suggested improvements: Please list additional analyses, experiments or data that could help strengthening the work in a revision.

See comments below. I think that o further experiments can be done given the effort and the complexity of these studies. Limitations are in the construction of the study and can only put in evidence.

RESPONSE: Thank you for this comment and for recognizing the complexity and robustness of our studies. Our response to Comment 4 describes the changes we made to put the limitations of our research in evidence, so we refrain from listing these changes here to avoid repetition.

COMMENT 6: References: Does this manuscript reference previous literature appropriately? If not, what references should be included or excluded?

See comments below.

Clarity and context: Is the abstract clear, accessible? Are abstract, introduction and conclusions appropriate?

The abstract is clear, as well as introduction and conclusions.

Please indicate any particular part of the manuscript, data, or analyses that you feel is outside the scope of your expertise, or that you were unable to assess fully.

Here below are detailed comments. The numbers below refer to the lines of the manuscript.

RESPONSE: We would like to thank you for the detailed comments. Below we describe the changes we made to address each of the comments.

COMMENT 7:

162

There are official definitions of what a robot is, issued by ISO and by IEEE. It would be better to refer to them rather than to a paper.

References

Institute of Electronics and Electrical Engineers: IEEE standard ontologies for robotics and automation. IEEE Std 1872-2015 pp. 1–60 (2015). DOI 10.1109/IEEESTD.2015.7084073

International Standards Organization: ISO 8373:2012 – Robots and robotic devices — Vocabulary. ISO (2012)

RESPONSE: We appreciate this suggestion. The robot definition that we used in the previous manuscript was taken from the IEEE website (<https://robots.ieee.org/learn/what-is-a-robot/>), and we are grateful that you have pointed us to the more formal definitions. In line with your suggestion, we replaced the previous definition with the following one: Institute of Electronics and Electrical Engineers: IEEE standard ontologies for robotics and automation. IEEE Std 1872-2015 pp. 1–60 (2015). DOI 10.1109/IEEESTD.2015.7084073. The revised section is available on **p.3**: “We adopt a general definition proposed by the Institute of Electrical and Electronics Engineers (IEEE²²), according to which robots are devices that can act in the physical world to accomplish different tasks and are made of mechanical and

electronic parts. These devices can be autonomous or, in some cases, subordinated to humans or software agents that act on behalf of humans. They can also form groups (i.e., robotic systems) in which they cooperate to accomplish collective goals (e.g., car manufacturing).”

We preferred the IEEE definition over the one proposed by the ISO because it is slightly more comprehensive and may be more informative for the readers.

COMMENT 8:

164

It would be better to focus on significant differences: sonars and infrared sensors both measure a simple distance, while a camera or a depth camera or a laser range finder would better give an idea of the range of possibilities in sensing, when compared to the simple single distance sensor such a sonar.

RESPONSE: Thank you for this suggestion. We replaced the robot definition we used in the original manuscript with the one you suggested in Comment 7 (Institute of Electronics and Electrical Engineers: IEEE standard ontologies for robotics and automation. IEEE Std 1872-2015 pp. 1–60 (2015). DOI 10.1109/IEEESTD.2015.7084073). Therefore, the text to which you refer in this comment was deleted because it was part of the previous definition we used.

COMMENT 9:

167

No robot with the characteristic b) (which means autonomy at least at a certain degree) are performing surgery. Surgical robots are completely tel-guidated and have no autonomy: all the responsibility is up to the surgeon. As an example of a sophisticated robot you may mention humanoids able to run and jump, or autonomous vehicles.

RESPONSE: Thank you for pointing this out. We replaced the robot definition we used in the original manuscript with the one you suggested in Comment 7 (Institute of Electronics and Electrical Engineers: IEEE standard ontologies for robotics and automation. IEEE Std 1872-2015 pp. 1–60 (2015). DOI 10.1109/IEEESTD.2015.7084073). Therefore, the text to which you refer in this comment was deleted because it was part of the previous definition we used.

COMMENT 10:

169

It seems that feelings investigated are not about robots, but about the idea of robot that people may

have developed from media, often not from direct experience. Direct experience may change a lot these feelings, as roboticists know from exhibitions of their products to general public. This should be put in evidence.

RESPONSE: Based on our understanding, this comment points out that, in the original manuscript, in the section in which we summarized affective responses toward robots uncovered by previous research, we mentioned only articles that studied images or descriptions of robots as stimuli but failed to mention articles focusing on physical robots. Therefore, to address your comment, we included the following references that examined affective responses to physical robots in the paragraph where we summarize affective responses toward robots in the revised manuscript (**pp.3-4**), and in the expanded version of this section in Supplementary Information (**p.6**):

Bonarini, A., Clasadonte, F., Garzotto, F., Gelsomini, M., & Romero, M. (2016, December). Playful interaction with Teo, a mobile robot for children with neurodevelopmental disorders. In *Proceedings of the 7th International Conference on Software Development and Technologies for Enhancing Accessibility and Fighting Info-exclusion* (pp. 223-231).

Nomura, T., Kanda, T., Suzuki, T., & Kato, K. (2008). Prediction of human behavior in human--robot interaction using psychological scales for anxiety and negative attitudes toward robots. *IEEE transactions on robotics*, 24(2), 442-451.

Darling, K., Nandy, P., & Breazeal, C. (2015, August). Empathic concern and the effect of stories in human-robot interaction. In *2015 24th IEEE international symposium on robot and human interactive communication (RO-MAN)* (pp. 770-775). IEEE.

Broadbent, E., MacDonald, B., Jago, L., Juergens, M., & Mazharullah, O. (2007, October). Human reactions to good and bad robots. In *2007 IEEE/RSJ International Conference on Intelligent Robots and Systems* (pp. 3703-3708). IEEE.

Seo, S. H., Geiskovitch, D., Nakane, M., King, C., & Young, J. E. (2015, March). Poor thing! Would you feel sorry for a simulated robot? A comparison of empathy toward a physical and a simulated robot. In *Proceedings of the Tenth Annual ACM/IEEE International Conference on Human-Robot Interaction* (pp. 125-132).

Moreover, we also want to point out that our comprehensive response to the general issue you raise in this comment (i.e., studying attitudes toward the concept of robot, rather than toward physical robots) is available in our answer to your Comment 2, where this issue is first raised. In this context, in the revised manuscript on **p.10** we acknowledge the role that direct experience plays in relation to psychological responses to robots, in line with your statement in the present comment: “Nonetheless, previous research showed that direct interaction with robots impacts people’s experiences^{27,171,172}.” However, we also explain why using depictions of robots rather than their physical counterparts as stimuli was the

only plausible option in our research (see the revised manuscript, **p.5**): “Across the three phases, we used robots that were either imagined (Phase One) or presented digitally on the screen and described (Phases Two and Three) as stimuli to avoid nongeneralizable findings that apply to few robots^{108,109}. Indeed, this approach allowed us to sample robots across all areas of human activity, which would not have been possible had we focused on physical robots, many of which are inaccessible for in-person research due to their size, cost, limited production, or potential use as weapons (e.g., industrial robots, military robots).”

COMMENT 11:

175

As recently put in evidence, uncanny valley effect comes from bad implementations of robots more than from the concept itself, as proposed by Mori more than 50 years ago.

References

The uncanny valley is wrong - https://www.youtube.com/watch?v=LKJBND_IRdI

Bartneck, C., Kanda, T., Ishiguro, H., & Hagita, N. (2009, September). My robotic doppelgänger-A critical look at the uncanny valley. In RO-MAN 2009-The 18th IEEE international symposium on robot and human interactive communication (pp. 269-276). IEEE.

Cheetham Marcus, Suter Pascal, Jäncke Lutz (2011) The Human Likeness Dimension of the “Uncanny Valley Hypothesis”: Behavioral and Functional MRI Findings, *Frontiers in Human Neuroscience*, 5, <https://www.frontiersin.org/articles/10.3389/fnhum.2011.00126>

RESPONSE: Thank you for pointing this out and sharing the resources. To address the issue you raise, in the revised manuscript we changed the text you are referring to (“Other important negative feelings are linked to “uncanny valley”—i.e., the notion that robots closely resembling humans but without a realistic human-like appearance evoke eeriness or creepiness”) into “Individuals can also find robots creepy if they are designed to be human-like but look unnatural and inconsistent with human appearance²⁸.” (**p.3**) This statement is now empirically correct, and it does not mention the uncanny valley or indicate that creepiness or eeriness is experienced when robots closely resemble humans but fail to fully attain the human look. We also deleted any mention of or reference to the uncanny valley from other parts of the article where we previously mentioned it. Finally, in Supplementary Information (**p.137**), we added the following section where we critically discuss the uncanny valley in the context of the present research:

“There are various insights that can be inferred from the PNC model regarding different psychological processes and their relationships. For example, feelings of creepiness or eeriness regarding a robot have traditionally been explained using the construct of the uncanny valley, according to which robots that closely resemble humans but fail to achieve a realistic human-like appearance evoke these

feelings^{3,201,202}. However, this construct has been criticized, and various studies have failed to support it^{203,204}. In line with these studies, the present research did not yield evidence that would support the uncanny valley as an explanation of creepiness. Indeed, in our Study 4 (Supplementary Table 6), creepiness was classified in the same domain as characteristics that convey the non-human nature of robots, such as “not human” or “emotionless”, but also in the same domain as characteristics that convey the possibility that robots are destroying our basic humanness and our society (e.g., “societal issues,” “robots contribute to human degeneration”). These insights are inconsistent with the uncanny valley in two ways. First, they indicate that robots may be perceived as creepy when they have non-human characteristics rather than when they resemble humans but fail to fully achieve a realistic human-like appearance. Second, they indicate that creepiness is not linked only to robots’ appearance but also to the consequences that robots may have for humans and wider society. This is only one example of how the PNC model can be used to critically examine different constructs related to robots and inspire various ideas and observations to deepen our understanding of the psychological processes these constructs tackle.”

COMMENT 12:

343 and 348

Links should be mentioned in print, not just as a link.

RESPONSE: Thank you for flagging this. We now state full links in print whenever we use links across the article (p.9, p.11, p.12, p.16).

COMMENT 13:

356

See comment on line 162: this is not true, since the normative organisms have defined them. Of course, anybody may have personal opinions, and the writer is also critical about the official definitions, but they are a reference for all norms related to robots.

RESPONSE: We are happy to implement this suggestion. In the revised manuscript, we changed the text you are referring to (“there is no consensus on how to define a robot”) into “describing them (i.e., robots) as an overarching category can be less straightforward^{19–21}.” (p.3) This statement is now factually correct because it does not indicate that a consensus definition of robots does not exist, but that defining robots as an overarching category is not always straightforward, which is reflected in previous research (e.g., Lo, 2017).

COMMENT 14:

357

This goal has some relevance, although it is an analysis of the expectations about robots that may be influenced by personal experiences and attitudes, as seen also in the cited literature, and not an analysis of the experience with robots.

RESPONSE: Thank you for sharing this insight. The statement from the original version of the manuscript you are referring to (“Second, to eventually capture representative and generalizable psychological processes in response to robots, it is important to go beyond experts and understand how everyday people perceive and define robots”) was changed in the revised manuscript to “The first step in this endeavor was devising a general definition of robots in Study 1, since robot definitions are typically proposed by experts^{6,21,22,110}, and it is less known whether they reflect how people more broadly perceive robots.” (p.6) We hope that this revised statement addresses your comment because it avoids implying that personal experiences and attitudes influence an analysis of the experience with robots.

COMMENT 15:

390

The sample is strongly biased on US and UK people, as from table 2. It should be mentioned from the beginning that these are results from a population of these two countries. Not much is said about confidence with technology and robots, but if they are people recruited through platform, their confidence with technology may be quite good. This is also a strong bias.

RESPONSE: We agree with your comment. To address it, we acknowledged that the present research is based on the UK and US samples already in the abstract (p.2): “Across seven studies that tested 9274 UK and US participants recruited via online panels...” Furthermore, compared to the original manuscript, we more comprehensively discussed this limitation in the Discussion section of the revised manuscript (p.10) to examine its implications for generalizability: “There are several limitations to this research. First, our participants were from Western countries (US and UK). Since our research proposed and investigated a construct (i.e., psychological processes regarding robots) from scratch, our priority was to establish its foundations. Combining the investigation of cultural differences with this agenda using equally meticulous methods would have exceeded the scope of a single article. Nevertheless, as measurement invariance analyses showed that the PNC model applies to individuals regardless of their income, age, education, use of robots at work, political orientation, ethnic identity, and relationship status, it is plausible that the model would generalize to countries different from the UK or US on these population characteristics. Conducting an in-depth examination of this question will be a crucial step as the research topic progresses.”

A more comprehensive version of this discussion is available in Supplementary Information (p.139) as follows:

“Second, our participants were from Western countries (US and UK). Since our research proposed and investigated a construct (i.e., psychological processes regarding robots) from scratch, our priority was to establish its foundations. Combining the investigation of cultural differences with this agenda using equally meticulous methods would have exceeded the scope of a single article. Nevertheless, as measurement invariance analyses showed that the PNC model applies to individuals regardless of their income, age, educational attainment, use of robots at work, political orientation, ethnic identity, employment status, and relationship status (Supplementary Tables 8-9), it is plausible that the model would generalize to countries different from the UK or US on these population characteristics. In other words, whether countries are rich or not, have liberal or conservative governments, have populations that differ in educational attainment and ethnic diversity, etc., might not matter for the structure of the PNC model. Overall, investigating whether these evidence-based inferences are accurate and generally examining the cultural generalizability of the PNC model will be a long-term project that may span a range of different studies and publications. Ideally, the first step would be sampling countries that are representative of the entire possible cultural space. For example, if Hofstede dimensions²¹⁹ are used to define cultural space, then countries across the spectrum of each of these dimensions (i.e., power distance, individualism vs. collectivism, masculinity vs. femininity, uncertainty avoidance, long vs. short-term orientation, and indulgence vs. restraint) would need to be selected. Second, for each of these countries, the construct of psychological processes regarding robots would ideally be tested using the rigorous methods employed in the present research, by first developing the robot definition and domains from participants’ responses, then mapping a comprehensive content space of psychological processes regarding robots, followed by organizing these processes into a taxonomy, and finally determining their individual difference predictors and the mechanisms behind these predictors. Then, the countries would be compared to each other to understand whether any differences exist. Finally, if any differences are detected, it would be necessary to understand which variables explain them. For example, did these differences arise due to cultural variables (e.g., individualism vs. collectivism²¹⁹) or some other influences (e.g., social, economic, environmental, political, etc.)? Conducting this type of in-depth examination of the generalizability of psychological processes regarding robots will be a crucial step as the research topic progresses.”

To address the part of your comment about recruitment through platform and confidence with technology, we stated already in the abstract that we recruited participants via online panels (**p.2**): “...9274 UK and US participants recruited via online panels...” Furthermore, we discussed and justified this potential limitation in the Discussion section (**p.10**): “Third, we recruited online participants who are inherently more confident with technology. Whereas this might have influenced the findings, alternative modes of recruitment (e.g., lab) would have yielded smaller and less representative samples^{166–170}. Furthermore, because measurement invariance analysis showed that our taxonomy was equivalent for people who do (vs. do not) use robots at work and all machine learning models we fit controlled for previous frequency of interaction with robots (Supplementary Tables 9-11), it is less plausible that technological proficiency biased the findings.”

A more comprehensive version of this discussion is available in Supplementary Information (p.140) as follows:

“Fourth, we recruited online participants who are inherently more confident with technology. Whereas this might have influenced the findings, alternative modes of recruitment (e.g., participant pool of the university research lab) were suboptimal because they would have resulted in smaller and less representative samples (e.g., university students and staff members plus some volunteers who live in the proximity of the university), which would have by default compromised the power and generalizability of the findings^{77–80,224}. Moreover, it is not a given that these modes of recruitment would have resulted in participants who have lower confidence with technology, considering that research participation is often advertised online and participants such as students tend to use technology for their studies. However, it is important to point out that various analyses we conducted make it less plausible that technological proficiency biased the findings. For example, measurement invariance analyses showed that our taxonomy was equivalent for people who do (vs. do not) use robots at work (Supplementary Table 9). Moreover, one of the covariates we used in the machine learning models was people’s previous frequency of interaction with robots (Supplementary Tables 10-11).”

COMMENT 16:

453

Possibly, it is more common to have experience with robots at home (e.g., cleaners, robotic pets) than at work.

RESPONSE: Although we could not find exact data to make this comparison, your speculation on people having more experience with certain types of robots at home than at work is probably right. We measured people’s use of robots at work to inform our future studies but also because we assumed that work is the most likely place where people may be able to encounter specialized robots they typically do not own because they may be too expensive, too large, etc. In that sense, we found this variable to be useful as a proxy for identifying people who may have access to some robots that other people typically encounter only via media or other similar routes. We also touch upon the topic of where people may encounter robots in Supplementary Information (p.140), as part of the section in which we discuss the choice of stimuli used in the present research: “For instance, among the 28 domains of human activity examined in this research (Supplementary Table 5), individuals may have direct exposure to robots in only a few of these domains (e.g., household, education, public services). Even within these domains, people may own only one or a few robots and directly interact with them (e.g., robotic pets, social robots, robotic vacuum cleaners). Conversely, it is plausible that people encounter most other robots and form opinions about them via social media or news websites in the form of visuals and descriptions.”

COMMENT 17:

484

Links should be mentioned in print, not just as a link.

RESPONSE: Thank you for flagging this. We now state full links in print whenever we use links across the article (**p.9, p.11, p.12, p.16**).

COMMENT 18:

526

This apparent contradiction hides an error of overgeneralization: a robot can have many different appearances and abilities. Possibly a robotic pet (robotic pets are the highest number of robots in our houses, increasing by millions every Xmas) might be more accepted than a robotic fighting drone, a Nao more accepted than Boston Dynamics' robots doing parkour.

References

Bonarini, A. (2016) Can my robotic home cleaner be happy? Issues about emotional expression in non-bio-inspired robots. *Adaptive Behavior*, 24(5), 335-349.

Bonarini A., Besio S. (2022), *Robot play for all: Developing Toys and Games for Disability*. Springer Cham, CH.

RESPONSE: We assume that the error of overgeneralization you are pointing out refers to the following statement from the section in which we more comprehensively described how different clusters of robot characteristics informed the definition we developed: "Part 4 was informed by Clusters 2 and 4 because it conveyed that people can perceive robots positively (Cluster 2), but they can also attribute negative characteristics to robots, generally because of their non-human nature (Cluster 4)." In the revised manuscript, this section was moved to Supplementary Information due to space constraints. However, we expanded the section as follows to address your comment (see Supplementary Information, **p.79**): "Part 4 was informed by Clusters 2 and 4 because it conveyed that people can perceive robots positively (Cluster 2), but they can also attribute negative characteristics to robots, generally because of their non-human nature (Cluster 4). This implies that robots can be perceived negatively in cases where people attribute a lack of humanness to them, but it does not mean that robots who are very different from humans are always seen negatively. Indeed, robots can have many different appearances and abilities, and robots who are less (vs. more) like humans might sometimes be more accepted by users^{178,179}." We also cited the references you mention (Bonarini, 2016; Bonarini & Besio 2022) at the end of the final sentence of this section.

COMMENT 19:

In general, Table 4 seems a discursive description of the ISO and IEEE definitions, so a confirmation that people may describe robots as the experts did, as then reported at the end of the paper as a feature of this research. Contrasts between "common sense" and "experts" should not be mentioned in the paper since the authors found that they aren't present. It is good that they could support that experts are not living in an ivory tower, and are not too far from laymen.

RESPONSE: We agree with this insight. To address your comment, we added the following paragraph in the Discussion section (**p.9**): "Starting with Phase One, we first focus on the robot definition (Table 2) and then discuss the domains (Table 3). Our definition shares several elements with the widely used IEEE definition²² since both conceptualize robots as devices or entities that can perform different tasks (Part 1, Table 2), emphasize that robots can have different degrees of autonomy (Part 2, Table 2), and include robots' composition (Part 5, Table 2). Our definition also contains two elements that are not mentioned by the IEEE—durability (Part 3, Table 2) and positive/negative attributes (Part 4, Table 2)—whereas the IEEE definition has one element we did not cover—robots' capability to form robotic systems. Overall, although our definition is somewhat more nuanced, both definitions are remarkably aligned, indicating that experts and lay individuals perceive robots similarly."

In this section, we discuss only the IEEE definition, given that this is the main definition that we use as the reference point in the article (**p.3**) and due to space constraints. However, in Supplementary Information (**pp.135-136**), we also discuss the ISO definition in the expanded version of the paragraph above:

"The robot definition we developed (Supplementary Table 4) shares several elements with the widely used IEEE definition⁸. They both conceptualize robots as devices or entities that can perform different tasks (Part 1, Supplementary Table 4), and they emphasize that robots can have different degrees of autonomy (Part 2, Supplementary Table 4). Moreover, they refer to robots' composition (Part 5, Supplementary Table 4). Our definition contains two elements that are not mentioned by the IEEE—durability (Part 3, Supplementary Table 4) and positive/negative attributes (Part 4, Supplementary Table 4)—whereas the IEEE definition has one element we did not cover—robots' capability to form robotic systems. Another common definition, proposed by the International Organization for Standardization (ISO²⁰⁰), is relatively brief and contains two elements that are also part of our definition: robots as devices that can perform tasks (Part 1, Supplementary Table 4) and have different degrees of autonomy (Part 2, Supplementary Table 4). Overall, although our definition is somewhat more nuanced, it is remarkably aligned with the two official definitions, which indicates that experts and lay individuals perceive robots similarly. Importantly, the two elements that all three definitions share refer to robots as devices or entities that perform different tasks (Part 1, Supplementary Table 4) and can vary in their degree of autonomy (Part 2, Supplementary Table 4). It is therefore possible to speculate that these two characteristics—performing tasks and a degree of autonomy—might be fundamental characteristics of robots because they have emerged in both expert and non-expert settings."

COMMENT 20:

678

It is relevant to have these questions listed, since they are strongly related to the main findings of Study 3.

RESPONSE: To address your comment, we re-read the relevant section in the previous version of the manuscript, and we realized that our description regarding how we assessed participants' psychological processes in Study 3 was somewhat vague. Therefore, in the revised manuscript, we have included a more precise description that accurately reflects the questions we asked them (**p.13**): "To assess participants' psychological processes, we then asked them to list and describe feelings they experienced (for affective responses), thoughts they had (for cognitive responses), and actions they engaged in (for behavioral responses) when they interacted with any robots they could think of from each domain, or to write about feelings, thoughts, and actions they could conceive in case they had never interacted with these robots." Moreover, as indicated in the Data Availability statement on **p.16**, all materials we used in the present research are publicly available via the Open Science Framework (OSF) using the following link: https://osf.io/2ntdy/?view_only=2cacc7b1cf2141cf8c343f3ee28dab1d

COMMENT 21:

724

The way the robot was presented should be reported in the paper: a picture, a description, a functional/operational description, all of them? The selection and the presentation of the robots strongly influence the answers. An explicit reference to supplementary material should be put here. Alexa (one of the "robots" included in the list) is by no way a robot, but an AI application. It does not move.

RESPONSE: We fully agree it is important that the way robots are presented should be clear to readers. To improve on this, we added the following description to the Study 4 procedure where the presentation of the robots you are referring to is used for the first time (**p.13**): "After answering the consent form, participants were randomly allocated to a domain (Table 3) and received a specific example of a robot from that domain that included an image and description approximately eight lines long (see Supplementary Figures 1-29). For the Sex domain, two robot examples were created (one male and one female), and participants assigned to this domain were randomly allocated to one." This paragraph now conveys the most important information about the stimuli we used (i.e., the robots were presented using an image and description roughly eight lines long). We were not able to include images and descriptions of the robots in the manuscript itself because the format of the journal is not suitable for this due to the number of robots we used. However, the presentation of all robots is included in Supplementary Information (see Supplementary Figures 1-29 on **pp.19-48** for Studies 4-7, and

Supplementary Figures 30-58 on **pp.49-77** for Studies 5-7), and this is now clearly stated in the Procedure sections for Studies 4-7 in the revised manuscript (**pp.13-15**). We thank you for this question and allowing us to clarify.

Regarding your comment on Alexa, the main reason why we included this device among the robot examples that we used as stimuli (Supplementary Information, **pp.28-29**) was because our research generally revolved around participants' perceptions, conceptualizations, and experiences of robots, and Alexa was spontaneously mentioned by participants as an example of a robot in 35 instances in Study 3 that we conducted to identify a comprehensive range of psychological processes regarding robots (raw analysis with participants' responses can be accessed in the file "Study 3 - Iterative Analysis File" that can be found in the folder "Study 3 -> Iterative Categorization" using the following link: https://osf.io/nejvm?view_only=79b6e42e24cb2a977927712bdcd2).

Considering that participants perceived Alexa as a robot, and that our work focused on understanding psychological processes regarding robots from participants' perspective, we found it appropriate to include Alexa among the stimuli. Based on other research, it seems common for people to use the word robot in reference to Alexa. For example, in a study by Liu and Yao (2023, p.4), participants said the following when talking to or referring to Alexa: "Why are you going to the movies you are a robot" or "Alexa is a robot." Moreover, in a study by Fortunati, Edwards, Edwards, Manganelli, and de Luca (2022, p.6) a participant made the following observation regarding Alexa: "males treat it more as a person than a robot and females use it more like a robot than a human." In that regard, our research is consistent with other studies when it comes to Alexa being perceived as a robot.

Importantly, given that we used two sets of stimuli (see Supplementary Information, **pp.19-77**), with one set of stimuli that did not include Alexa, and that measurement invariance analyses (see Table 6 on **pp.41-42** in the manuscript) showed that the taxonomy we developed had equivalent factor structure, loadings, and intercepts for both of these two stimuli sets, evidence indicates that participants experience Alexa like other stimuli that depict robots. In other words, it does not seem that Alexa confounded the findings regarding psychological processes in response to robots.

We want to emphasize that we agree with you that Alexa does not move and therefore may not be fully aligned with one of the elements of the IEEE robot definition you mention in Comment 7 (i.e., "acting in the physical world in order to accomplish one or more tasks"). However, since Alexa can play music and even control physical tasks such as switching on/off a lightbulb or adjusting the temperature on a thermostat, it is possible that people perceive it as a robot because of how they intuitively interpret its functionalities in relation to their conceptualization and definition of robots. For example, in relation to our definition that was developed from participants' responses (see Table 2 on **p.31** in the revised manuscript), people may see Alexa as a robot because it follows commands (Part 2), performs certain tasks (e.g., playing music, switching on a smart lightbulb; Part 1), can work tirelessly (Part 3), and runs on software while having the appearance of a non-living object (Part 5). Although we did not expose participants to the IEEE definition mentioned above, it is also possible they might see Alexa as a robot in

the context of this definition because, from their perspective, Alexa can act in the physical world (e.g., it can operate light bulbs and various other devices in smart homes), it can act on behalf of humans, and it is made of mechanical and electronic parts.

Based on psychological research and theorizing, it would not be surprising if lay participants indeed interpreted robot definitions more loosely and perceived devices that are not stringently aligned with these definitions as robots. For example, according to construal level theory of psychological distance (Trope & Liberman, 2010), a person can be psychologically close to or distant from an event, object, person, etc. Being psychologically distant from an object is associated with a more abstract way of thinking about it, referred to as high construal level, which results in using broader categories when making sense of the respective object. Therefore, given that lay individuals are probably psychologically more distant from robots because they do not work with them every day, they may perceive them in a more abstract way and thus use broader and more overarching categories when deciding what a robot is or is not. In that context, they would broadly interpret playing music or switching off a smart lightbulb as acting in the physical world. Investigating this further may be an interesting topic for future research.

Overall, to address your comment, on **pp.28-29** in Supplementary Information where we present Alexa as one of the stimuli, we added the following footnote to explain why we use it amongst other robots:

“Although Alexa can change its physical surroundings by switching on and off or controlling smart home devices such as lightbulbs, it does not move and may therefore not be fully aligned with one of the elements of the IEEE robot definition⁸ (i.e., “acting in the physical world in order to accomplish one or more tasks”). Nevertheless, the main reason why we included this device among the stimuli was because our research generally revolved around participants’ perceptions, conceptualizations, and experiences of robots, and Alexa was spontaneously mentioned by participants as an example of a robot in 35 instances in Study 3 (raw analysis with participants’ responses can be accessed in the file “Study 3 - Iterative Analysis File” that can be found in the folder “Study 3 -> Iterative Categorization” using the following link: https://osf.io/nejvm?view_only=79b6eeee42e24cb2a977927712bdcdd2).

Considering that participants saw Alexa as a robot, and that our research tackled psychological processes regarding robots from participants’ perspective, we found it appropriate to use the image and description of Alexa as one of the stimuli. Based on other research, it seems common for people to use the word robot in reference to Alexa. For example, in a study by Liu and Yao¹⁷⁶ (p.4), participants said the following when talking to or referring to Alexa: “Why are you going to the movies you are a robot” or “Alexa is a robot.” Moreover, in a study by Fortunati, Edwards, Edwards, Manganelli, and de Luca¹⁷⁷ (p.6) a participant made the following observation regarding Alexa: “males treat it more as a person than a robot and females use it more like a robot than a human.” In that regard, our research is consistent with other studies when it comes to Alexa being perceived as a robot.

Importantly, given that we used two sets of stimuli (Supplementary Figures 1-58), the second one of which did not include Alexa, and that measurement invariance analyses (Supplementary Table 8) showed

that the taxonomy we developed had equivalent factor structure, loadings, and intercepts for these two stimuli sets, evidence indicates that participants experience Alexa like other stimuli that depict robots. In other words, it does not seem that Alexa confounded the findings regarding psychological processes in response to robots.”

COMMENT 22:

1223

In the light of the mentioned references (see comments about line 175, above) this is not surprising at all. Moreover, attitudes towards humans themselves may affect opinions about humanoids, as it appears also from the findings reported later in the paper.

RESPONSE: We assume that this comment refers to the following statement from the previous version of the manuscript: “the most surprising finding was that anthropomorphism positively predicted both positive and negative psychological processes.” To address your comment, we have deleted this statement from the revised manuscript.

COMMENT 23:

1250

What is the reference to (SM, pp.100-109)? Supplementary material may be mentioned explicitly.

RESPONSE: In the previous manuscript, we used the abbreviation SM in reference to Supplementary Materials. However, in line with your comment, we agree that this negatively affected the clarity of the manuscript. Therefore, in the revised manuscript, we always use a full phrase when referring to content from Supplementary Information (e.g., Supplementary Methods, Supplementary Figures, Supplementary Results), and we do not use any abbreviations in this respect. This is also in line with the formatting requirements of the journal. The file that was previously called Supplementary Materials is now labelled Supplementary Information because the journal requires this phrasing.

COMMENT 24:

1256 and 1284

Links should be mentioned in print, not just as a link.

RESPONSE: Thank you for flagging this. We now state full links in print whenever we use links across the article (**p.9, p.11, p.12, p.16**).

COMMENT 25:

1452

The authors has to add a couple of domains not mentioned, but present in their experience and possibly other domains are missing, maybe because related to a niche or not enough known by people in the sample, or not associated to robots.

RESPONSE: You are right that, despite using a very meticulous procedure to identify the domains (i.e., qualitative analyses of participants' responses in combination with a literature review), there is always a chance that we missed some domain that is niche or that is just emerging. To acknowledge this, we have added the following sentence in the revised manuscript (p.9): "It is important to emphasize that, despite the meticulous procedure we used to develop the list, it is always possible we failed to identify some more niche domains."

COMMENT 26:

1459

This is not "what robot are", but "what the idea of robot is", quite different from reality, and still it is not clear whether the expectations will be matched. For instance, most of the autonomous robots are expected to survive at least a working day without being attached to a power source, and research about this is struggling since years without any significant results. Many "intelligent" robots needs to be attached to a reliable internet link and are not autonomous.

RESPONSE: We assume that this comment refers to the following statement from the previous version of the manuscript: "On a theoretical level, it allows researchers to understand how people perceive what robots are (Tables 3 and 4) and where to find them (Table 5)." To address your comment, we have removed this statement from the revised manuscript, and it is not present either in the manuscript or Supplementary Information.

COMMENT 27:

1499

Given the time passed from the IEEE work and the significant advancement of technology, we may expect that this list will grow in the next years.

RESPONSE: We are happy to implement this insight. In the revised manuscript, we have included the following sentence to acknowledge it (p.9): "Additionally, the number of domains will likely continue increasing as technology advances."

COMMENT 28:

1627

Declaring a limitation does not mean that it does not exist, and it should put in evidence from the beginning and prevent to present results as general. A careful phrasing should be considered throughout the paper.

RESPONSE: To our understanding, this comment refers to the limitation we discussed in the previous version of the manuscript that concerns using Western samples (UK and US; in the revised manuscript, the limitation, including its implications for generalizability, is discussed in the first highlighted paragraph on **p.10**, whereas an extended version of this discussion is available in Supplementary Information, **p.139**). In line with your suggestion, we have now acknowledged that the present research is based on these samples already in the abstract (**p.2**): "Across seven studies that tested 9274 UK and US participants recruited via online panels..." This should allow the readers to understand from the start to which populations our results may apply, and it is a common practice in abstracts from Nature Human Behaviour (for a more comprehensive overview of the changes we made in this regard, please see our response to your Comment 15). To help the readers get a deeper understanding of the samples we used, we expanded Table 1 (**pp.29-30**) by adding information about employment status and use of robots at work, and in Supplementary Information (**p.142**) we added even more extensive tables that include information about participants' region in the US or UK, a more comprehensive breakdown of age categories, etc.

Overall, we would like to thank you once again for investing effort into reading our manuscript and giving us many useful comments. We hope that our responses reflect that we have taken your suggestions seriously and used them to make the manuscript stronger.

Decision Letter, first revision:

19th July 2023

Dear Dr Krpan,

Thank you once again for your revised manuscript, entitled "The Positive-Negative-Competence (PNC) Model of Psychological Responses to Robots," and for your patience during the re-review process.

Your manuscript has now been evaluated by Reviewer 2 from the original round of review, as well as a new Reviewer (Reviewer 3) with expertise in human-robot interaction. All reviewer feedback is included at the end of this letter. Although the reviewers found your manuscript to have improved

during revision, they also raise some important outstanding concerns. We remain very interested in the possibility of publishing your study in Nature Human Behaviour, but would like to consider your response to these outstanding concerns in the form of a revised manuscript before we make a decision on publication.

1. Please modify the framing in your manuscript to fit what your experiments can actually show, and discuss in detail the key limitations that Reviewer 2 points out. In doing so, please also follow Reviewer 3's advice and ensure that it is clear to the reader how the different experiments relate to each other.

2. Reviewer 3 asks that you acknowledge some additional previous literature in your manuscript, and discuss similarities and differences with your current work. Please revise your manuscript accordingly.

In sum, we invite you to revise your manuscript taking into account all reviewer and editor comments. We are committed to providing a fair and constructive peer-review process. Do not hesitate to contact us if there are specific requests from the reviewers that you believe are technically impossible or unlikely to yield a meaningful outcome.

We hope to receive your revised manuscript within 4-8 weeks. I would be grateful if you could contact us as soon as possible if you foresee difficulties with meeting this target resubmission date.

- Include a "Response to the editors and reviewers" document detailing, point-by-point, how you addressed each editor and referee comment. If no action was taken to address a point, you must provide a compelling argument. This response will be used by the editors and reviewers to evaluate your revision.
- Highlight all changes made to your manuscript or provide us with a version that tracks changes.

[REDACTED]

We look forward to seeing the revised manuscript and thank you for the opportunity to review your work. Please do not hesitate to contact me if you have any questions or would like to discuss these revisions further.

Sincerely,

Samantha Antusch

Samantha Antusch, PhD
Senior Editor
Nature Human Behaviour

Reviewer expertise:

Reviewer #2: machine learning ; human-robot interaction

Reviewer #3: human-robot interaction

REVIEWER COMMENTS:

Reviewer #2:

Remarks to the Author:

Thank you for your answers, which make more clear to me your approach, and for the modifications you implemented in the paper.

As far as I understand, what you are presenting is not about proposing a "model that synthesizes people's psychological reactions to robots and identifies the factors that shape them" (as stated in the paper), but, instead, proposing a "model that synthesizes the psychological reactions to robots of the population from which you have extracted the sample and identifies the factors that shape them". Could we we say that you consider the "robot construct" for this population, based on images, descriptions of robots generated by the investigators (how?), and (non investigated) background of the subjects?

From the paper, the characteristics of the population are not defined, nor are inclusion criteria for the subjects. There are no ways to understand whether the sample is a "good" representation of a "general" population; cardinality of a sample, as you may well know, is not a way to support the fact that a sample is a good sample.

So, first the characteristic of the population should be defined, then the criteria for the selection of subject should be given.

Given this, the specificity of your results should be put in evidence avoiding any possible interpretation of them as concerning robots as entities in general, whose construct may be built on different experiences, which may completely modify your results.

This may also give a justification for your choice of presenting so many robot images and descriptions, and for that kind of classification, incomplete and with overlapping categories, typical of informal and non-informed classification processes.

Some assumptions made in the paper are personal opinions of the investigators. For instance the number of people interacting with robots different from industrial robots is much higher that the number of people interacting with industrial robots. So, assuming that a question about this is a proxy

for "confidence with robots" is not correct. Even in your investigation you have seen that people consider even Alexa and similar stuff as robots, and millions of toy robots reach our homes every Christmas. Direct experience with them (good or bad) influence your research.

In general, there may be activities that cannot be done, or it would be too expensive to do, but this cannot be a justification for performing activities in substitution that have different characteristics. So, the justification that making subjects experiencing the interaction with so many actual robots would have been unfeasible cannot be considered as valid to justify the presentation of images and descriptions instead. There is evidence in literature that the actual interaction changes a lot opinions about robots.

Moreover, the mentioned fact that this approach is typical of certain literature does not mean that it is scientifically correct. Psychology is a hard discipline, and research should be performed with a scientific method. A non-scientific approach leads to what is common to see in this area: papers contradicting each other, often due to unjustified and unsupported assumptions.

Given this, your research is valid and interesting to investigate this specific "robot construct" as defined above and by no means there should appear in the paper anything that may lead to the doubt that these results are valid for the "general population", in particular in the title, abstract, and conclusions.

Reviewer #3:

Remarks to the Author:

The authors address a significant topic given the probable increasing role of robots in our society. The proposed Positive-Negative-Competence (PNC) model categorizes psychological reactions to robots by following what seems a very thorough process. The paper sets out an ambitious agenda that consists of conducting and presenting a series of seven different studies that are cleverly sequentially arranged in three different stages. The approach taken in all these studies appears complete and methodologically sound.

Originality, References and Citations:

The paper has referenced a wide range of research. However, I do feel like that important literature from the field of Human-Robot Interaction is left out of the introduction, discussion and/or supplemental material. Here follows 3 examples:

- Carpinella, C. M., Wyman, A. B., Perez, M. A., & Stroessner, S. J. (2017, March). The robotic social attributes scale (RoSAS) development and validation. In Proceedings of the 2017 ACM/IEEE International Conference on human-robot interaction (pp. 254-262).

-Bartneck, C., Kulić, D., Croft, E., & Zoghbi, S. (2009). Measurement instruments for the anthropomorphism, animacy, likeability, perceived intelligence, and perceived safety of robots. International journal of social robotics, 1, 71-81.

-Heerink, M., Kröse, B., Evers, V., & Wielinga, B. (2010). Assessing acceptance of assistive social agent technology by older adults: the almere model.

Not acknowledging the similarities and dissimilarities between this work and the main contributions from the HRI field makes it harder to assess the originality of the paper. I do feel that each study addresses a relevant problem but the several focuses of the paper might have led the authors to not fully delve into the appropriate related work and in clarifying the why and novelty of each study.

Clarity:

The paper is well written but suffers from clarity problems as it is hard to keep up with each individual study. I feel like a figure/diagram that would connect the different studies would help the reader navigate the paper better and more clearly understand its multiple contributions. A website that contains all the supplemental material in a more organized hierarchical fashion might also contribute to further clarity and be relevant for extending the relevance of the work.

A small mistake can be found in lines 270 and 272 where anthropomorphism appears to be listed twice unnecessarily.

Humans vs Robots and Robot Domains

The robot domains in table 3 appear to be categorized with different granularity levels. As examples, category 2 is "social and companionship", category 10 is "Leisure, recreation, and travel" and category 11 is "Culture/entertainment, gaming, toys, and other amusement". Leisure and recreation with robots seem perhaps to be a subdomain of category 2 and category 11 seem to be a subdomain of 10. Some parts of these examples such as gaming, toys and travel seem to be even more specific. While this categorization is still useful, I think that a deeper dive in this part of the work would have yielded even greater contributions. I would also like the authors to discuss whether, if the technology allows for, robots with a similar embodiment to a human will be able to perform and simulate all human activities. If that is the case, this work could soon become more outdated as all human activities will become possible robot activities and + all the many superhuman tasks that can be provided by different embodiments or superhuman strength/speed/reasoning etc.

Overall, the sheer amount of work presented in this paper already deserves attention from several research fields. If iterated further in other publications, it can indeed lead to broad implications for how we understand human psychological reactions to robots, which could impact numerous domains such as the ones listed in the paper. While I do think the paper is publishable as is, I do recommend the authors to work on better positioning their work in the lens of human-robot interaction and to attempt to make the work of the reader easier by providing a smoother navigation through the 7 different studies with one of the suggestions above.

Author Rebuttal, first revision:

REVIEWER 2'S COMMENTS

COMMENT 1: Thank you for your answers, which make more clear to me your approach, and for the modifications you implemented in the paper.

RESPONSE: We would like to thank you for reading our revised manuscript in depth and providing us with additional comments that we believe further strengthen it. We carefully considered each of your comments, and below we explain the changes we made to address them.

Before we proceed with responding to the comments, we outline several formatting details that should be helpful in navigating the revisions we made. In the revised manuscript and the Supplementary Information file, all changes that we made in response to your and Reviewer 3's comments are highlighted in yellow. Furthermore, in both the revised manuscript and Supplementary Information, references appear as superscript Arabic numerals. Therefore, whenever we copy-paste a section from the manuscript or Supplementary Information in response to your comment, the references are

presented in this format and can be matched to the sources in the corresponding reference lists. Overall, we are grateful for the time and effort you dedicated to reading our manuscript and for the valuable advice you gave us.

COMMENT 2: As far as I understand, what you are presenting is not about proposing a "model that synthesizes people's psychological reactions to robots and identifies the factors that shape them" (as stated in the paper), but, instead, proposing a "model that synthesizes the psychological reactions to robots of the population from which you have extracted the sample and identifies the factors that shape them". Could we say that you consider the "robot construct" for this population, based on images, descriptions of robots generated by the investigators (how?), and (non investigated) background of the subjects?

From the paper, the characteristics of the population are not defined, nor are inclusion criteria for the subjects. There are no ways to understand whether the sample is a "good" representation of a "general" population; cardinality of a sample, as you may well know, is not a way to support the fact that a sample is a good sample.

So, first the characteristic of the population should be defined, then the criteria for the selection of subject should be given.

Given this, the specificity of your results should be put in evidence avoiding any possible interpretation of them as concerning robots as entities in general, whose construct may be built on different experiences, which may completely modify your results.

This may also give a justification for your choice of presenting so many robot images and descriptions, and for that kind of classification, incomplete and with overlapping categories, typical of informal and non-informed classification processes.

RESPONSE: We appreciate the insights you provide in this comment. Based on our interpretation, this comment posits that our model of psychological processes applies to a population of robots from which we sampled the stimuli used in the present research (i.e., in your own words: "model that synthesizes the psychological reactions to robots of the population from which you have extracted the sample"). Your main observation is that the characteristics of this population of robots are not well defined in the manuscript, and therefore there is no way of understanding whether our stimuli are representative of this general population of robots (i.e., in your words: "From the paper, the characteristics of the population are not defined... There are no ways to understand whether the sample is a "good" representation of a "general" population). For that reason, you conclude that we should not frame our results as being representative of a general population of robots and that we should frame our findings more specifically in relation to the stimuli sample we used (i.e., in your words: "the specificity of your results should be put in evidence avoiding any possible interpretation of them as concerning robots as entities in general"). You further state that framing the manuscript in such a way would also provide justification for the procedure we used to select the stimuli (i.e., in your own words: "This may also give a justification for your choice of presenting so many robot images and descriptions"). In the remainder of our response, we first discuss this comment and then outline the changes we made to address it.

Overall, we agree with your observation that we do not precisely define in the manuscript the characteristics of the robot population from which we sampled the stimuli in a way this would for example be possible for a human population of participants. As Wells and Windschitl (1999, p.1123) acknowledge: “it is often difficult or impossible to define the stimulus population.” For a human population of a specific country, census data that cover various characteristics such as population size, age, gender, geographical region, ethnicity, education etc. exist, and it is therefore straightforward to use some of these variables to define the population of interest and recruit a sample of participants.

For robots, we attempted to find similar data that would allow us to define a general population of robots to use in our research, but we fell short of this goal for several reasons. First, whereas the numbers of robots belonging to certain categories (e.g., industrial) are being tracked, this is not clear for all robot categories, and it is also not clear which specific robots exist and how many, considering that technology is rapidly advancing, and new varieties of robots are continuously being produced. Second, and more importantly, whereas for humans there are clear variables that can be used to define a population (e.g., age, gender, region, etc.), for robots this is less straightforward, at least when it comes to defining their population for the purpose of psychological research. For example, average age of the entire population of robots is not known, and even if it were known it may not be a good characteristic to define this population. However, what would be the good characteristics to define the population? Are these characteristics being measured? Can they be used for sampling the stimuli?

We spent a lot of effort to investigate these questions when we were starting our research project, and we did not find a good solution to define the general population of robots in a way this would be possible for the human population. Our methodological procedure was an optimal strategy we developed to attempt to circumvent this problem and identify the general population in an alternative way. Our rationale can be summarized as follows. We reasoned that the general population of robots can be defined as “all robots that currently exist”. Given that identifying all such robots was implausible, we considered that developing an exhaustive list of the domains into which robots could be classified would comprise a summary of the general population. Using this strategy did not require us to know each specific robot that existed because the domains themselves would contain groups of robots that are sufficiently similar.

Phase One of our research put this approach into practice by producing an exhaustive list of 28 robot domains across all areas of human activity (see Table 2, **pp.33-34**). The idea was that identifying a robot for each of these domains would comprise a summary of the general population of robots and would therefore be representative of this population. To ensure that the specific robots chosen per each domain did not make a difference, in Study 5 we used two different robot examples per each domain (Supplementary Information, **pp.19-77**) and performed measurement invariance analyses to show that the structure of our taxonomy (i.e., factor structure, factor loadings, and item intercepts) did not change for these two stimulus sets. As can be seen from Table 5 (**p.43**) in the revised manuscript, measurement invariance analyses indeed convincingly demonstrated that the structure of the taxonomy remained consistent regardless of the robot examples.

Overall, we believe that the arguments outlined above, and the measurement invariance analyses reported, support the notion that our stimuli were representative of the general population of robots. However, regardless of our rationale, we agree with your comment and acknowledge that we cannot formally show evidence for this because we cannot define the general population in a quantifiable manner that would resemble human populations, which is a common problem that applies to stimulus sets in psychological research (Wells and Windschitl, 1999). Therefore, we made several changes to address your comment, with which we agree.

First, we deleted from the manuscript and Supplementary Information any expressions which would indicate that our stimuli are representative of the general population of robots, and instead of “representative” and “generalizable” we used expressions such as “comprehensive” or “exhaustive” which more accurately describe our stimulus sample (Supplementary Information, **pp.19-77**) and the procedures we used to develop it (e.g., for an overview of how we developed the robot domains into which the stimuli are organised, see sections “Phase One: Mapping a Comprehensive Content Space of Robots From All Domains of Human Activity” on **pp.5-6 and 11-12**). For example, we changed the expression “a representative set of stimuli” into “a comprehensive stimulus sample” (**p.2**). Such changes were also made on **p.5** (i.e., changing “representative content space” into “comprehensive content space” and removing the expression “representative of the stimuli population” previously on the same page), **p.9** (i.e., changing “generalizable sample of robots” into “comprehensive sample of robots”), **p.10** (i.e., removing the expression “representative across diverse robots” previously on this page), and **p.34** (i.e., changing “representative content space” into “comprehensive content space” in Table 2 note).

Second, in line with your suggestion to frame our findings more specifically in relation to the stimuli sample we used (in your words: “the specificity of your results should be put in evidence avoiding any possible interpretation of them as concerning robots as entities in general”), we changed how we present this across the paper:

- In the abstract (**p.2**), we changed “in response to a representative set of stimuli depicting robots from all domains of human activity (e.g., education, hospitality, industry)” into “in response to a comprehensive stimulus sample depicting robots from 28 domains of human activity (e.g., education, hospitality, industry).” In the abstract (**p.2**), we also changed “all psychological processes regarding robots” into “all psychological processes in response to the stimulus sample”, and we changed “psychological functioning in relation to robots” into “psychological functioning regarding representations of robots” to emphasize that our findings apply to the specific stimuli and robot construct we investigated rather than to a general population of robots.
- Importantly, in the introduction (**p.5**), we added a paragraph that includes the following text: “Despite the wide variety of our stimulus sample, it is unclear to what degree this sample is representative of the general population of robots because (a) there are no established recommendations on what variables would need to be measured to accurately define this population, (b) the type of data used to quantify general characteristics of human populations is not available for robots, and (c) the field of robotics is rapidly evolving. Therefore, in the context of our research, we use the term “robot/s” in reference to our specific stimulus sample, and we do not imply that our insights extend to the general population (i.e., all physical robots).” This points out the limitation that, despite the comprehensiveness of our

stimuli, we cannot understand at present to what degree they are representative of the general population of robots, and notifies the reader that our research and its findings should be interpreted in the light of the stimulus sample we use.

- In the discussion section, we revised the paragraph (p.10) in which we discussed using depictions of robots rather than actual robots as a limitation by stating "...the stimuli were not physical robots but their depictions... Therefore, based on the present findings, it is not known whether our taxonomy applies to the physical counterparts of the robots our stimuli depict."

Overall, these revisions now accurately portray our research and its findings and frame them more specifically in relation to the stimuli we used.

For the end, we just want to note that our understanding is that your Comment 2 refers to the population of robots from which we sampled the stimuli rather than to our participant populations because your language is strongly indicative of this and because the criticism you express does not apply to our participant populations. Namely, you start the comment by referring to "robots of the population from which we have extracted the sample" and to "the robot construct for this population," and you conclude by stating that we should avoid interpreting our results "concerning robots as entities in general." Moreover, the criticism you raise about the population not being clearly defined does not apply to our research participants because a) Table 1 (pp.31-32) in the manuscript contains participant characteristics suitable for the main article, b) Supplementary Tables 1 and 2 (p.143 in Supplementary Information) contain a comprehensive breakdown of participant characteristics including the exact criteria used for representative sampling for gender, age, and region, c) this is clearly stated in the manuscript on p.11 ("Table 1 summarizes key participant information. In Studies 4-6, participants were recruited to be reasonably representative of the UK/US populations for age, gender, and geographical region, and in Study 1 (Sample 1) for gender only. More comprehensive breakdowns of participant information and the criteria used for representative sampling are available in Supplementary Tables 1-2."), and d) in the second paragraph on p.11 we describe the inclusion criteria in terms of checks participants had to pass for their data to be included in analysis. Therefore, based on all this, we do not have any reason to believe your comment concerns our participant populations, and we are flagging this just to show that we considered your comment from many angles to address it.

COMMENT 3: Some assumptions made in the paper are personal opinions of the investigators. For instance the number of people interacting with robots different from industrial robots is much higher than the number of people interacting with industrial robots. So, assuming that a question about this is a proxy for "confidence with robots" is not correct. Even in your investigation you have seen that people consider even Alexa and similar stuff as robots, and millions of toy robots reach our homes every Christmas. Direct experience with them (good or bad) influence your research.

RESPONSE: Thank you for flagging this. We agree that referring to the question about robots at work as being a proxy for "confidence with robot" may not be optimal considering that we wanted to identify participants who used robots at work because work is the most likely place where people may be able to

encounter specialized robots they typically do not own (e.g., due to their cost, size, limited production, etc.). Therefore, we removed the reference to “confidence with robots” in relation to the variable about the use of robots at work from the paragraph in question, and the paragraph is now as follows (**p.10**):

“Third, we recruited online participants who are inherently more confident with technology. Whereas this might have influenced the findings, alternative modes of recruitment (e.g., lab) would have yielded smaller and less representative participant samples^{171–175}. Furthermore, to reduce the chance of technological proficiency biasing the findings, all ML models controlled for a variable indicative of technological proficiency (i.e., previous frequency of interaction with robots; Supplementary Tables 10-11).”

Moreover, we also revised an expanded version of this paragraph available in Supplementary Information by removing the reference to “confidence with robots” in relation to the variable about the use of robots at work (see Supplementary Information, final paragraph on **p.141**).

COMMENT 4: In general, there may be activities that cannot be done, or it would be too expensive to do, but this cannot be a justification for performing activities in substitution that have different characteristics. So, the justification that making subjects experiencing the interaction with so many actual robots would have been unfeasible cannot be considered as valid to justify the presentation of images and descriptions instead. There is evidence in literature that the actual interaction changes a lot opinions about robots.

Moreover, the mentioned fact that this approach is typical of certain literature does not mean that it is scientifically correct. Psychology is a hard discipline, and research should be performed with a scientific method. A non-scientific approach leads to what is common to see in this area: papers contradicting each other, often due to unjustified and unsupported assumptions.

RESPONSE: Thank you for this comment. To address it, we removed from the revised manuscript the content you are critical of that refers to justifying using a particular approach just because it is widely used or because another approach is not possible (in your own words: “there may be activities that cannot be done, or it would be too expensive to do, but this cannot be a justification for performing activities in substitution that have different characteristics” and “the mentioned fact that this approach is typical of certain literature does not mean that it is scientifically correct”). First, we deleted the following sentence and the paragraph that contained it (previously on **p.5**): “Indeed, this approach allowed us to sample robots across all areas of human activity, which would not have been possible had we focused on physical robots, many of which are inaccessible for in-person research due to their size, cost, limited production, or potential use as weapons (e.g., industrial robots, military robots).” Second, we deleted the following text (previously on **p.10**): “(These stimuli) are widely used.”

We removed this content from the manuscript to address your comment, but also to avoid any misinterpretation, because now we see that it can be interpreted differently than we intended, especially the statement that referred to justifying using a particular approach just because another

approach is not possible. Therefore, in our response to your comment, beyond outlining the changes we made above, we also want to more clearly describe our thinking behind initially including this content.

All stimuli ultimately have their limitations. Various theories and paradigms in psychology indicate that people's formation of attitudes and impressions about an object is influenced by numerous factors, including direct interaction with this object, information we read about the object, its depictions, news about it, etc. (e.g., Smith & Queller, 2002; Smortchkova, Dołrega, & Schlicht, 2020). Therefore, a perfect stimulus set would comprise all these factors, which is not possible, and researchers inherently must select the stimuli that are most appropriate for what they want to achieve in their research. In the case of robots, using physical robots as stimuli is of course preferable, and studies using physical robots as stimuli have produced immense insights on how people interact with robots, perceive them, etc. In that respect, we hold your research in high regard.

However, for the type of research we conducted, our focus was on many diverse robots because taxonomies are inherently about classifying and organising a large amount of information. Therefore, we had to select the stimuli that were compatible with this focus and use them in the most scientific way possible. Focusing on physical robots as stimuli was simply not an option because many of the robots we planned to test would not be available for in person research for various reasons (e.g., cost, size, limited production, potential use as weapons, not being commercially available). Therefore, the choice was between either finding an alternative approach to testing actual robots and making it scientifically valid, or not conducting the research.

In this context, we think that your statement (i.e., "there may be activities that cannot be done, or it would be too expensive to do, but this cannot be a justification for performing activities in substitution that have different characteristics") is fair and valid when directed at research projects that use stimuli A simply because they could not use stimuli B but do not make their methodological approach regarding stimulus selection systematic, robust, and scientifically valid. We understand that we attracted this criticism because the statements that are now deleted but that the previous manuscript contained were overly simplistic and created the impression that we support such practices (i.e., using stimuli A just because stimuli B cannot be obtained but without applying scientific rigour).

To avoid any misunderstandings, we want to emphasize that we absolutely do not support such practices, and that our methodological approach regarding stimuli was comprehensive and rigorous, which you also acknowledged in your comments on the first version of our manuscript (in your own words: "The methodological apparatus is impressive and may be a guidance for further research"). In fact, the main goal of the entire Phase One of our research was to inform our stimulus selection by first asking participants to generate robot characteristics (Study 1: Sample 1) to develop a robot definition (Study 1: Sample 2) and identify the list of all domains of human activity where robots as aligned with this definition can be found (Study 2). Ultimately, we chose to depict robots from these domains by using images and descriptions rather than alternative stimuli such as videos to minimize the number of potential confounding effects associated with the stimuli (this is now explained in Supplementary Information, in paragraph 1 on **p.140**). For example, whereas different videos covering the robots in question could have different durations and contain texts of different lengths and narration styles,

images allowed us to avoid the duration as a potential confound, and to equalize the length and styles of robot descriptions, thus also avoiding these elements as confounds.

Overall, we thank you for your comment, and we hope that our response helps in understanding additional background behind our methodological approach, in addition to deleting from the revised manuscript the content you were critical of in this comment.

COMMENT 5: Given this, your research is valid and interesting to investigate this specific "robot construct" as defined above and by no means there should appear in the paper anything that may lead to the doubt that these results are valid for the "general population", in particular in the title, abstract, and conclusions.

RESPONSE: We appreciate your comment and acknowledgment that our research is valid and interesting to investigate the robot construct you are referring to. To address the comment and make sure that we do not indicate in particular in the title, abstract, and conclusions that our findings apply to a general population of robots, we made several changes.

First, we changed the title (**p.1**) from "The Positive-Negative-Competence (PNC) Model of Psychological Responses to Robots" to "The Positive-Negative-Competence (PNC) Model of Psychological Responses to Representations of Robots." The limit length for titles in Nature Human Behaviour is 100 characters (including spaces), which means that only brief titles are possible, and we could not make the title any more specific than this. However, the title now conveys that our findings refer to representations of robots, which accurately describes our stimuli, rather than to actual physical robots.

Second, we revised the abstract as follows (**p.2**): "Robots are becoming an increasingly prominent part of society. Despite their growing importance, there exists no overarching model that synthesizes people's psychological reactions to robots and identifies what factors shape them. To address this, we created a taxonomy of affective, cognitive, and behavioral processes in response to a comprehensive stimulus sample depicting robots from 28 domains of human activity (e.g., education, hospitality, industry) and examined its individual difference predictors. Across seven studies that tested 9274 UK and US participants recruited via online panels, we used a data-driven approach combining qualitative and quantitative techniques to develop the Positive-Negative-Competence (PNC) model, which categorizes all psychological processes in response to the stimulus sample into three dimensions: positive, negative, and competence-related. We also established the main individual difference predictors of these dimensions and examined the mechanisms for each predictor. Overall, this research provides an in-depth understanding of psychological functioning regarding representations of robots."

From the abstract, it is now clear that (a) our taxonomy specifically applies to depictions of robots from 28 domains of human activity, and the word "representative" is not mentioned, (b) the taxonomy was developed from responses of specific participants (9274 UK and US participants recruited via online panels), (c) the PNC model applies to the specific stimuli described above, and (d) the main insights of our research are valid for representations of robots rather than actual physical robots. Therefore, the

abstract does not present the findings in relation to some general population of robots, but in relation to the specific stimuli and methodological approach we used. Please do note that the first and second sentence of the abstract describe the motivation behind our research, which is what Nature Human Behavior recommends including in the abstract, and that is why they refer to “robots” and do not mention the specific stimuli we used. In the remainder of the abstract, it is then described how this motivation is reflected in our specific methodological approach.

Third, in the introduction section (**p.5**), we added the following paragraph:

“All in all, to achieve our research objectives, as stimuli we used representations (i.e., images and descriptions) of robots (Supplementary Figures) from 28 exhaustive domains of human activity that robots operate in (Table 2). This comprehensive approach allowed us to minimize the chance that our findings are driven by idiosyncrasies of a sample that is small in size and/or variety of robot types, which could compromise replicability^{111,112}. Despite the wide variety of our stimulus sample, it is unclear to what degree this sample is representative of the general population of robots because (a) there are no established recommendations on what variables would need to be measured to accurately define this population, (b) the type of data used to quantify general characteristics of human populations is not available for robots, and (c) the field of robotics is rapidly evolving. Therefore, in the context of our research, we use the term “robot/s” in reference to our specific stimulus sample, and we do not imply that our insights extend to the general population (i.e., all physical robots).”

This important paragraph precisely defines our stimuli (i.e., images and descriptions of robots from 28 exhaustive domains of human activity), it points out the comprehensive methodological approach we used but acknowledges that despite this it should not be assumed that the stimuli are representative of the “general population” of robots because we did not find a way of validating this, and finally it clearly states that whenever we use the word “robot/s” in relation to our research, we are referring to our specific stimuli. Overall, this paragraph ensures that our readers clearly understand the specific stimuli and context to which our research applies.

Finally, in the discussion section, we revised the paragraph (see **p.10**) that discusses the limitation of not using physical robots as stimuli, and we included this paragraph as the first and most important limitation (it was the fifth limitation in the previous version of the article):

“There are several limitations to this research. First, the stimuli were not physical robots but their depictions. These stimuli hold ecological validity since people often interact with robots indirectly (e.g., via social media or various websites), and many psychological processes may therefore be shaped in this manner. Nonetheless, previous research showed that direct interaction with robots impacts people’s experiences^{27,169,170}. Therefore, based on the present findings, it is not known whether our taxonomy applies to the physical counterparts of the robots our stimuli depict, and investigating this is currently unachievable because many of these robots are inaccessible for in-person research due to their size, cost, limited production, or potential use as weapons (e.g., industrial robots, military robots). However, this research may be possible in the future if such robots become more accessible.”

This paragraph now clearly states that “it is not known whether our taxonomy applies to the physical counterparts of the robots our stimuli depict”, which was not clearly stated in the previous version. Therefore, the paragraph makes it explicit that we can only claim that our findings apply to the stimuli we tested, and it explains why at present it is not possible to empirically investigate whether the taxonomy we developed would be consistent with physical robots. An extended version of this paragraph is also available in Supplementary Information (see first two paragraphs on **p.140**).

Overall, we hope that these revisions address the concerns you express in Comment 5, and we also want to point out that our response to Comment 2 additionally addresses these concerns because it describes the rationale behind our stimulus selection approach and expresses that we deleted from the manuscript and Supplementary Information any expressions which would indicate that our stimuli are representative of the general population of robots, and instead of “representative” we used expressions such as “comprehensive” or “exhaustive”.

We thank you once again for investing the effort into reading our manuscript and giving us another round of excellent, very useful feedback. We have taken your suggestions seriously, and we hope that our responses reflect this.

REVIEWER 3’S COMMENTS

COMMENT 1: The authors address a significant topic given the probable increasing role of robots in our society. The proposed Positive-Negative-Competence (PNC) model categorizes psychological reactions to robots by following what seems a very thorough process. The paper sets out an ambitious agenda that consists of conducting and presenting a series of seven different studies that are cleverly sequentially arranged in three different stages. The approach taken in all these studies appears complete and methodologically sound.

RESPONSE:

Thank you for reading our revised manuscript and acknowledging its strengths while giving us suggestions to improve it. We are particularly grateful for the ideas on how to make the manuscript clearer and more aligned with previous research. Below we explain the changes we made to address your comments.

Before we proceed with responding to the comments, we outline several formatting details that should be helpful in navigating the revisions we made. In the revised manuscript and the Supplementary Information file, all changes that we made in response to your and Reviewer 2’s comments are highlighted in yellow. Furthermore, in both the revised manuscript and Supplementary Information, references appear as superscript Arabic numerals. Therefore, whenever we copy-paste a section from the manuscript or Supplementary Information in response to your comment, the references are presented in this format and can be matched to the sources in the corresponding reference lists. Overall, thank you for dedicating your time to our manuscript and for your valuable insight and suggestions.

COMMENT 2: Originality, References and Citations:

The paper has referenced a wide range of research. However, I do feel like that important literature from the field of Human-Robot Interaction is left out of the introduction, discussion and/or supplemental material. Here follows 3 examples:

- Carpinella, C. M., Wyman, A. B., Perez, M. A., & Stroessner, S. J. (2017, March). The robotic social attributes scale (RoSAS) development and validation. In Proceedings of the 2017 ACM/IEEE International Conference on human-robot interaction (pp. 254-262).

-Bartneck, C., Kulić, D., Croft, E., & Zoghbi, S. (2009). Measurement instruments for the anthropomorphism, animacy, likeability, perceived intelligence, and perceived safety of robots. *International journal of social robotics*, 1, 71-81.

-Heerink, M., Kröse, B., Evers, V., & Wielinga, B. (2010). Assessing acceptance of assistive social agent technology by older adults: the almere model.

Not acknowledging the similarities and dissimilarities between this work and the main contributions from the HRI field makes it harder to assess the originality of the paper. I do feel that each study addresses a relevant problem but the several focuses of the paper might have led the authors to not fully delve into the appropriate related work and in clarifying the why and novelty of each study.

RESPONSE: Thank you for providing us with these useful references that helped us to further improve the integration of our research with previous relevant work. Considering that one of the references you mention—Heerink, Kröse, Evers, and Wielinga (2010)—extends the technology acceptance model (TAM), similar to the unified theory of acceptance and use of technology on which it directly draws, we found it most appropriate to include this reference in the introduction where we refer to TAM in the context of inductive integration.

In particular, in the relevant paragraph (i.e., see the fifth full paragraph on **p.4**), we explain that our research lends itself to a data-driven approach because, although various theories and models of human-technology relationships and interactions exist, these models typically have a more specific purpose than our PNC model and do not aim to encapsulate the entirety of psychological functioning regarding robots by identifying, organizing, and predicting psychological processes that robots trigger. To address your comment, we expanded the following sentence from this paragraph where we illustrate the relevant models we are referring to (**p.4**): “To illustrate, the technology acceptance model (TAM⁸⁴⁻⁸⁶) and its extensions, the unified theory of acceptance and use of technology (UTAUT⁸⁷⁻⁸⁹) and the Almere model⁹⁰, examine the factors that make people accept technology (e.g., perceived usefulness, ease of use, social influence), whereas the media equation⁹¹⁻⁹³ examines if people interact with media (e.g., computers) similar to how they interact with other humans.”

Therefore, this addition recognizes several theories and models that are highly relevant in the field of human-robot interaction and acknowledges that, whereas our research and these theories/models all deal with human-technology relationships, our research has a more general purpose (i.e., to encapsulate the entirety of psychological functioning regarding robots by identifying an exhaustive list of

psychological processes and organizing them), whereas these models typically have more specific purposes (e.g., to understand factors that make people accept technology).

Concerning the remaining references that you mention (i.e., Bartneck, Kulić, Croft, & Zoghbi, 2009; Carpinella, Wyman, Perez, & Stroessner, 2017), we found it most appropriate to include them in the discussion section and create a new paragraph in which we use them in explaining the main contribution of the PNC model. In particular, we think that our research, these articles, and other articles that study human-robot relationships and interactions are related in the sense that they all study some kind of psychological reactions to robots. For example, for Bartneck, Kulić, Croft, and Zoghbi (2009), these are anthropomorphism, animacy, likeability, perceived intelligence, and perceived safety of robots, whereas for Carpinella, Wyman, Perez, and Stroessner (2017) these are various social attributes. However, the goal of our research was not to focus on these or some other specific psychological reactions, but to identify all reactions that may comprise human psychological functioning and investigate them under the umbrella of psychological processes. In that regard, the PNC model can be seen as an integrative framework that links and organizes an exhaustive list of psychological processes, both those that researchers have already studied separately and the less common ones that our participants generated.

To express this in the revised manuscript and address your comment by adding the relevant references you mention, we included the following paragraph on **p.9**:

“One of the main contributions of our research is showing that these seemingly highly diverse psychological processes fall under three dimensions: positive (P), negative (N), and competence (C; Tables 3 and 4). In general, previous research on human-robot relationships and interactions generally focused on studying and measuring specific psychological reactions to robots (e.g., safety, anthropomorphism, animacy, intelligence, likeability, various social attributes^{159,160}) but did not attempt to identify all these reactions and investigate them under an all-encompassing construct of psychological processes. In that regard, the PNC model can be seen as an integrative framework that links and organizes an exhaustive list of psychological processes, both those that researchers have already studied separately and the less common ones that our participants generated. We believe that our model moves the field forward not only through this integration but also by enabling researchers to systematically study psychological processes regarding robots by (a) using the PNC as a guide to inform the design of future research on these processes and (b) employing the PRR scale to measure them.”

An extended version of this paragraph is available in Supplementary Information (**pp.137-138**).

COMMENT 3: Clarity:

The paper is well written but suffers from clarity problems as it is hard to keep up with each individual study. I feel like a figure/diagram that would connect the different studies would help the reader navigate the paper better and more clearly understand its multiple contributions. A website that contains all the supplemental material in a more organized hierarchical fashion might also contribute to further clarity and be relevant for extending the relevance of the work.

RESPONSE: We agree that, due to complexity of our research, navigating different studies and understanding how they are connected can be challenging, and we particularly welcomed your suggestions on how to improve clarity. We therefore made several changes to address your comment.

First, we created a figure that summarizes our research by presenting the goals of each research phase, summarizing how these goals are achieved via the corresponding studies, outlining the link between the successive phases, describing the studies, and outlining the analyses conducted for each study. We think that, in line with your suggestion, this figure brings the entire manuscript and its studies together because it provides an integrative summary of our research, and after reading the figure our audience should be able to more easily navigate the specific studies because of having the mental picture of the research as a whole.

The figure caption is available in the revised manuscript as follows (p.30): **“Figure 1: Overview of the Present Research.** “Phase” states the goals of each research phase, summarizes how the goals were achieved, and outlines the link between the successive phases. “Study” and “Description” indicate the number of each study and its goal, whereas “Analysis” specifies the statistical analyses that were used in each study.”

The figure itself was uploaded separately from the manuscript file, and we are also copy-pasting it on the next page of this document for your convenience.

In addition to creating the figure you recommended, to address your Comment 3 we also combined previous Tables 2 and 3 into one Table 2 (see pp.33-34 in the revised manuscript). Our rationale behind this step was as follows. Previous Tables 2 and 3 presented the robot definition developed in Study 1 and the domains of human activity in which these robots play a role developed in Study 2, respectively. Since the robot definition was used to develop the domains, and since the definition and domains together convey the essence of robots as described by our participants (i.e., what robots are and where they can be found), we thought it would be more holistic to present this information in a single table so that the readers can see it in one place. In other words, we thought that this improves the clarity of findings obtained in Phase One.

Furthermore, to address your Comment 3, we improved the headings in the Methods section (pp.11-14). In the previous manuscript, this section had only basic headings (i.e., Phase One; Phase Two; Phase Three) without any description of the phases. In the revised manuscript, this section now has more descriptive headings that correspond to the headings of Phases in Figure 1 so they can be linked to the Figure and more easily navigated. In particular, the following headings were included in the revised manuscript:

“Phase One: Mapping a Comprehensive Content Space of Robots From All Domains of Human Activity” (p.11)

“Phase Two: Developing the Taxonomy (i.e., Key Dimensions) of Psychological Processes” (p.12)

“Phase Three: Determining Main Individual Difference Predictors and Their Mechanisms” (p.14)

Whereas the Results section from the revised manuscript already had descriptive headings that correspond to the new Figure 1, we made some minor phrasing changes in that section to improve clarity (see the highlighted text on pp.5-8).

Finally, to address your Comment 3, we also considered organizing supplemental material in a more hierarchical fashion and uploading it on a repository website (i.e., OSF). Although we have created this document and you can access it using the link in the parentheses (https://osf.io/96csz?view_only=2cacc7b1cf2141cf8c343f3ee28dab1d), we are concerned that it might potentially confuse the readers for the following reasons. Official Supplementary Information that the journal requires needs to be organized using only the following sections (in line with how we organized the official file): Supplementary Figures, Supplementary Tables, Supplementary Notes, Supplementary Methods, Supplementary Results, Supplementary Discussion, and/or Supplementary References. If we link another supplemental material file with a different organisation to the article, the readers might get confused by trying to compare the documents and make sense of the information that is presented differently in the files. This is why our preference would be not to use the alternative supplemental material file, especially because the other revisions you suggested significantly improved the clarity of the article, but we are happy to include this file if you deem it appropriate.

COMMENT 4: A small mistake can be found in lines 270 and 272 where anthropomorphism appears to be listed twice unnecessarily.

RESPONSE: Thank you for pointing this out. In the text you are referring to, we wanted to express that anthropomorphism was identified as one of the key individual difference predictors of both the positive and negative dimension of the PNC model. However, we now realize that this was not explained with sufficient clarity, and to address your comment we therefore revised the text as follows (p.8):

“Several variables met these criteria and were therefore deemed the main individual difference predictors of the PNC dimensions. For the positive dimension, these were general risk propensity (GRP¹⁴⁹), anthropomorphism (IDAQ¹⁵⁰), and parental expectations (FMPS_PE¹⁵¹); for the negative dimension, they were trait negative affect (PANAS_TNA¹⁵²), psychopathy (SD3_P¹⁵³), anthropomorphism (IDAQ¹⁵⁰), and expressive suppression (ERQ_ES¹⁵⁴); and for the competence dimension, they were approach temperament (ATQ_AP¹⁵⁵) and security-societal (PVQ5X_SS¹⁵⁶).”

COMMENT 5: Humans vs Robots and Robot Domains

The robot domains in table 3 appear to be categorized with different granularity levels. As examples, category 2 is “social and companionship”, category 10 is “Leisure, recreation, and travel” and category 11 is “Culture/entertainment, gaming, toys, and other amusement”. Leisure and recreation with robots seem perhaps to be a subdomain of category 2 and category 11 seem to be a subdomain of 10. Some

parts of these examples such as gaming, toys and travel seem to be even more specific. While this categorization is still useful, I think that a deeper dive in this part of the work would have yielded even greater contributions.

RESPONSE: We think that your appraisal of the categories we developed is fair, and we appreciate that you find the categorisation useful. As we explain in the note to Table 2 that outlines the robot domains (p.34): “Our aim was to develop domains that are narrow rather than broad, which means that some overlap between them may be present. This approach was aligned with our objective to establish a comprehensive content space of all robots to decrease the probability of using a biased stimulus sample^{111,112} when developing the taxonomy of psychological responses to representations of robots. For that reason, it was more optimal to lean toward having too many rather than too few domains to reduce the chance of failing to cover the content space of all robots in detail and omitting important types of robots.”

In that context, as you indicate, we agree that some domains could potentially be subdomains of other domains, or that they may be mutually inclusive. However, by treating them separately we wanted to ensure that we do not miss any important robot types in our research. As Willig (2018) pointed out, qualitative analyses are inevitably informed by the researcher’s perspective or approach to interpretation, and the categories we have created were informed by our objective to generate a comprehensive sample of robots across all areas of human activity (this limitation is discussed in Supplementary Information, paragraph 3 on pp.140).

In that respect, whereas from our perspective the categories have achieved our research aims, we fully agree with you that “a deeper dive in this part of the work” could have yielded interesting and important contributions. Importantly, we do not perceive the present paper as the end process, and we think that the data our participants generated can still be used for that purpose. Indeed, in the section Data Availability (p.16 in the revised manuscript), it can be seen that we shared our data (including participants’ responses from which we generated the categories) using the following link: https://osf.io/2ntdy/?view_only=2cacc7b1cf2141cf8c343f3ee28dab1d We hope that other researchers will be able to use these data to generate novel insights through quantitative analyses, or that our categories as reported and discussed in the paper will inspire novel research projects.

COMMENT 6: I would also like the authors to discuss whether, if the technology allows for, robots with a similar embodiment to a human will be able to perform and simulate all human activities. If that is the case, this work could soon become more outdated as all human activities will become possible robot activities and + all the many superhuman tasks that can be provided by different embodiments or superhuman strength/speed/reasoning etc.

RESPONSE: Thank you for suggesting this idea. We agree this is an interesting future possibility with implications for the PNC model we developed and therefore should be discussed in the manuscript. To answer your comment, we added the following paragraph in the discussion section when discussing limitations to address the issue you raise (p.10):

“Finally, rapid technological development might make robots with a similar embodiment to humans able to perform and simulate all human activities, therefore significantly changing how people perceive robots. However, since our comparison of the PNC model and SCM^{161,162} (note: SCM refers to the stereotype content model that we discuss on p.9) indicates that people form impressions of robots and humans in a similar manner, it is unlikely that robots becoming more like humans will have a significant impact on the structure of our model. Even if it does, the PNC can be updated via the same methodological procedures we used.”

We also added the same paragraph to Supplementary Information (p.142).

COMMENT 7: Overall, the sheer amount of work presented in this paper already deserves attention from several research fields. If iterated further in other publications, it can indeed lead to broad implications for how we understand human psychological reactions to robots, which could impact numerous domains such as the ones listed in the paper. While I do think the paper is publishable as is, I do recommend the authors to work on better positioning their work in the lens of human-robot interaction and to attempt to make the work of the reader easier by providing a smoother navigation through the 7 different studies with one of the suggestions above.

RESPONSE: We thank you once again for the useful suggestion and recognizing the qualities of our work. We hope that our revisions address the concerns you raised, and we find that your advice significantly improved the manuscript.

Decision Letter, second revision:

14th August 2023

Dear Dr. Krpan,

Thank you for your patience as we've prepared the guidelines for final submission of your Nature Human Behaviour manuscript, "The Positive-Negative-Competence (PNC) Model of Psychological Responses to Representations of Robots" (NATHUMBEHAV-22102805B). Please carefully follow the step-by-step instructions provided in the attached file, and add a response in each row of the table to indicate the changes that you have made. Please also check and comment on any additional marked-up edits we have proposed within the text. Ensuring that each point is addressed will help to ensure that your revised manuscript can be swiftly handed over to our production team.

We would hope to receive your revised paper, with all of the requested files and forms within two-three weeks. Please get in contact with us if you anticipate delays.

Nature Human Behaviour offers a Transparent Peer Review option for new original research manuscripts submitted after December 1st, 2019. As part of this initiative, we encourage our authors to support increased transparency into the peer review process by agreeing to have the reviewer comments, author rebuttal letters, and editorial decision letters published as a Supplementary item. When you submit your final files please clearly state in your cover letter whether or not you would like to participate in this initiative. Please note that failure to state your preference will result in delays in accepting your manuscript for publication.

In recognition of the time and expertise our reviewers provide to Nature Human Behaviour's editorial process, we would like to formally acknowledge their contribution to the external peer review of your manuscript entitled "The Positive-Negative-Competence (PNC) Model of Psychological Responses to Representations of Robots". For those reviewers who give their assent, we will be publishing their names alongside the published article.

Cover suggestions

As you prepare your final files we encourage you to consider whether you have any images or illustrations that may be appropriate for use on the cover of Nature Human Behaviour.

ORCID

Non-corresponding authors do not have to link their ORCIDs but are encouraged to do so. Please note that it will not be possible to add/modify ORCIDs at proof. Thus, please let your co-authors know that if they wish to have their ORCID added to the paper they must follow the procedure described in the following link prior to acceptance:

Nature Human Behaviour has now transitioned to a unified Rights Collection system which will allow our Author Services team to quickly and easily collect the rights and permissions required to publish your work. Approximately 10 days after your paper is formally accepted, you will receive an email in providing you with a link to complete the grant of rights. If your paper is eligible for Open Access, our Author Services team will also be in touch regarding any additional information that may be required to arrange payment for your article.

Please note that *Nature Human Behaviour* is a Transformative Journal (TJ). Authors may publish their research with us through the traditional subscription access route or make their paper immediately open access through payment of an article-processing charge (APC). Authors will not be required to make a final decision about access to their article until it has been accepted. Find out more about Transformative Journals

[REDACTED]

Best regards,
Alex McKay

Editorial Assistant
Nature Human Behaviour

On behalf of

Samantha Antusch

Samantha Antusch, PhD
Senior Editor
Nature Human Behaviour

Final Decision Letter:

Dear Dr Krpan,

We are pleased to inform you that your Article "The Positive-Negative-Competence (PNC) Model of Psychological Responses to Representations of Robots", has now been accepted for publication in Nature Human Behaviour.

Please note that *Nature Human Behaviour* is a Transformative Journal (TJ). Authors may publish their research with us through the traditional subscription access route or make their paper immediately open access through payment of an article-processing charge (APC). Authors will not be required to make a final decision about access to their article until it has been accepted. Find out more about Transformative Journals

With best regards,

Samantha Antusch

Samantha Antusch, PhD
Senior Editor
Nature Human Behaviour